The fossil Osmundales (Royal Ferns)—a phylogenetic network analysis, revised taxonomy, and evolutionary classification of anatomically preserved trunks and rhizomes

Bomfleur Benjamin 1 2 bbomfleur@uni-muenster.de
http://orcid.org/0000-0003-0674-3553 Grimm Guido W. 3 4
http://orcid.org/0000-0001-6723-239X McLoughlin Stephen 2
1 Institut für Geologie und Paläontologie, Westfälische Wilhelms-Universität Münster , Münster , Germany
2 Department of Palaeobiology, Swedish Museum of Natural History , Stockholm , Sweden
3 Department für Paläontologie, Universität Wien , Wien , Austria
4 Orléans , France
DiMichele William
Electronic publication date: 2017 Jul 11
Publication date: 2017
Volume: 5
Electronic Location ID: e3433
Received 2016 Dec 17; Accepted 2017 May 17
Copyright: © 2017 Bomfleur et al.
Copyright year: 2017
Copyright holder: Bomfleur et al.
License: This is an open access article distributed under the terms of the Creative Commons Attribution License, which permits unrestricted use, distribution, reproduction and adaptation in any medium and for any purpose provided that it is properly attributed. For attribution, the original author(s), title, publication source (PeerJ) and either DOI or URL of the article must be cited.
License URL: https://creativecommons.org/licenses/by/4.0/

Keywords: Evolutionary classification, Osmundaceae, Phylogeny, Filicopsida, Leptosporangiatae, Network analysis, Taxonomy, Anatomy, Ferns, Guaireaceae

Funding: Swedish Research Council (VR grants) 2014–5232 and 2014–5234 Austrian Science Fund FWF M1751-B16 This work was supported by the Swedish Research Council (VR grants 2014–5232 to B. Bomfleur and 2014–5234 to S. McLoughlin) and the Austrian Science Fund FWF (M1751-B16 to G. W. Grimm). The funders had no role in study design, data collection and analysis, decision to publish, or preparation of the manuscript.

==============================
The Osmundales (Royal Fern order) originated in the late Paleozoic and is the most ancient surviving lineage of leptosporangiate ferns. In contrast to its low diversity today (less than 20 species in six genera), it has the richest fossil record of any extant group of ferns. The structurally preserved trunks and rhizomes alone are referable to more than 100 fossil species that are classified in up to 20 genera, four subfamilies, and two families. This diverse fossil record constitutes an exceptional source of information on the evolutionary history of the group from the Permian to the present. However, inconsistent terminology, varying formats of description, and the general lack of a uniform taxonomic concept renders this wealth of information poorly accessible. To this end, we provide a comprehensive review of the diversity of structural features of osmundalean axes under a standardized, descriptive terminology. A novel morphological character matrix with 45 anatomical characters scored for 15 extant species and for 114 fossil operational units (species or specimens) is analysed using networks in order to establish systematic relationships among fossil and extant Osmundales rooted in axis anatomy. The results lead us to propose an evolutionary classification for fossil Osmundales and a revised, standardized taxonomy for all taxa down to the rank of (sub)genus. We introduce several nomenclatural novelties: (1) a new subfamily Itopsidemoideae (Guaireaceae) is established to contain Itopsidema, Donwelliacaulis, and Tiania; (2) the thamnopteroid genera Zalesskya, Iegosigopteris, and Petcheropteris are all considered synonymous with Thamnopteris; (3) 12 species of Millerocaulis and Ashicaulis are assigned to modern genera (tribe Osmundeae); (4) the hitherto enigmatic Aurealcaulis is identified as an extinct subgenus of Plenasium; and (5) the poorly known Osmundites tuhajkulensis is assigned to Millerocaulis. In addition, we consider Millerocaulis stipabonettiorum a possible member of Palaeosmunda and Millerocaulis estipularis as probably constituting the earliest representative of the (Todea-)Leptopteris lineage (subtribe Todeinae) of modern Osmundoideae.

Introduction

The Royal Ferns (Osmundales) form the most ancient surviving lineage of leptosporangiate ferns. The order comprises about 20 extant species originally placed in three or four genera with four (three) subgenera. The recently published, first comprehensive classification of the pteridophyte phylogeny group (PPG) recognizes all former subgenera of Osmunda as distinct genera (PPG I, 2016). Osmundales has arguably the richest and most informative fossil record of any extant group of ferns (Arnold, 1964; Miller, 1971; Tidwell & Ash, 1994). Detached fertile foliage (e.g., Anomopteris Brongn., 1828, Todites Seward, 1900, Cladotheca T.Halle, 1911, Osmundopsis T.M.Harris, 1931, Cacumen Cantrill & J.A.Webb, 1987, and Osmunda L., 1753, and possibly Damudopteris D.D.Pant & P.K.Khare, 1974 and Dichotomopteris Maithy, 1974), together with morphologically similar sterile fronds (e.g., Cladophlebis Brongn., 1849) and dispersed spores (e.g., Osmundacidites Couper, 1953, Todisporites Couper, 1958), are probably the most common organs of Osmundales in the fossil record but, in many cases, these remains are difficult to discriminate with confidence from other fern orders. Moreover, foliage adpressions and spores typically lack the character resolution necessary for confident attribution to families and genera within Osmundales (Escapa & Cúneo, 2012). Anatomically preserved trunks and rhizomes of Osmundales occur in the fossil record since the Permian (matching the stratigraphic range of fossil foliage and spores), and show an extraordinary taxic and structural diversity (Arnold, 1964; Miller, 1971; Tidwell & Ash, 1994; Tian, Wang & Jiang, 2008; Wang et al., 2014b; Fig. S1). The most comprehensive and detailed synthesis of the fossil record of osmundalean rhizomes remains Miller’s (1971) “Evolution of the fern family Osmundaceae based on anatomical studies.” Miller compiled, analysed, and evaluated an unparalleled amount of data on the axis structure of most extant and all fossil Osmundaceae known at that time, and provided systematic descriptions and taxonomic treatments for all included taxa in a standardized format; he also carefully designed and analysed character matrices in order to reconstruct the phylogenetic history of the group from the Permian to the present. However, since that time, well over 50 additional species, 12 new genera, four new subfamilies, and one new family of Osmundales have been established based on new finds or alternative interpretations of permineralized axes alone (see reviews of, e.g., Tidwell & Ash, 1994; Tian, Wang & Jiang, 2008; Wang et al., 2014b; Fig. S1). Hence, our knowledge about the diverse structural architectures in osmundalean axes has advanced considerably, providing a sound basis for a re-evaluation of the significance of anatomical features for taxonomic delimitation and systematic classification. Unfortunately, not all subsequent authors adopted Miller’s standardized terminology and format of description; consequently, studies of the anatomy of fossil Osmundales are riddled with inconsistent terminology and taxonomy, which renders any attempt at an exhaustive analysis problematic.

Here, we present and discuss the results of a novel systematic–phylogenetic analysis of 129 osmundalean axes, including all currently known fossil records (accepted species plus synonyms and additional separately described specimens) plus 14 extant representatives. We briefly review the diversity of anatomical features of osmundalean axes using a revised, uniform, descriptive terminology, and introduce a new morphological character matrix consisting of 45 architectural characters of the plant axis. The matrix serves two main purposes: the standardized description and identification of specimens and the characterization of natural groups. The latter incorporates results of customized phylogenetic network analyses of this matrix. Based on these results, we propose an anatomy-based classification for fossil Osmundales that is as natural as possible, with a revised and standardized taxonomy listing diagnostic axis characters for all taxa down to the rank of (sub)genus. For practicality, we also accept several explicitly paraphyletic taxa as valid taxonomic units. Appendices provide the formal taxonomic treatments of nomenclatural novelties (Appendix A), a glossary of terms and abbreviations (Appendix B), a polytomous identification key (Appendix C), and a tutorial and example analysis (File S5) to aid the taxonomic placement of future discoveries of fossil Osmundales axes.

Axis anatomy in Osmundales—a critical reappraisal

Osmundalean axes range from small, creeping or shortly erect rhizomes to huge, arborescent trunks, and they display a remarkable diversity in anatomical structure. Features that are common to all osmundalean axes—extant and extinct—include (1) radial symmetry; (2) helical phyllotaxis; (3) a central cylindrical stele; (4) a single peripheral metaxylem siphon surrounding the stem core; and (5) a single, primarily C-shaped (i.e., adaxially concave), entire vascular bundle in the base of the stipe (Fig. 1). Most osmundalean stems have a prominent cortex (except species of Osmundacaulis) and are surrounded by a prominent mantle of roots and persistent stipe bases (except some Guaireaceae).

Figure 1 Diagrammatic representation of a cutout stem portion of a Millerocaulis-type osmundoidalean rhizome (Osmundoideae, Osmundaceae) showing a selection of general anatomical features.

Petiolar parenchyma in yellow-green; outer and inner surface of sclerenchymatic outer cortex in light grey; xylem in brown; parenchyma of pith and inner cortices left transparent and most roots omitted for clarity reasons. Labelled features are as follows: a = stelar xylem segment; b = internal embayment in metaxylem siphon indicative of an incipient leaf gap formation; c = departing leaf trace; d = leaf gap forming complete perforation of the xylem cylinder immediately upon departure of leaf trace; e = peripheral incision into stelar xylem cylinder above departed leaf trace; f = leaf trace in inner stem cortex; g = leaf trace in outer stem cortex; h = stipe vascular bundle; i = root trace upon departure from stem. Anatomical features of cortex and mantle: j = outer sclerenchymatic stem cortex; k = stipe sclerenchyma cylinder (“sclerenchyma ring” in TS); l = stipular wings forming part of the mantle. A PDF version of this image without labels is provided in the Supplemental Information.

Stem core

In Guaireaceae and in most Osmundaceae, the stem core consists primarily of a parenchymatous pith, in some cases with varying amounts of additional sclerenchyma and, more rarely, tracheids or “medullary traces”; in thamnopteroid Osmundaceae, the stem core is composed of tracheids that may be short, elongated, or enlarged and “parenchyma-like” (Fig. 2). Thus, three basic types have traditionally been described to occur in Osmundales: “protosteles” (characterizing subfamily Thamnopteroideae) and more or less modified “siphonosteles” and “dictyosteles” (characterizing subfamily Osmundoideae and family Guaireaceae, respectively).

Figure 2 Diagram showing different tissue compositions of selected types of osmundalean stem cores as seen in cross-section through the stem core, together with the respective character scoring used in the matrix (for definition of characters and of character states see text).

A gradual transition can be drawn from only tracheids (Thamnopteris gracilis) to parenchyma to fully sclerified stem cores (extant Todea barbara). Parenchyma in pale yellow; sclerenchyma in light brown; tracheids/xylem in dark brown.

However, this classification falls short of recognizing that the stele of thamnopteroid Osmundaceae is neither a true protostele (like those of Grammatopteris and Rastropteris, considered by some authors to be ancient relatives of Osmundales) nor a true siphonostele, but rather an intermediate between the two (Fig. 2): it lacks a true parenchymatous pith yet has a distinct peripheral metaxylem siphon that can be entire or perforated just like those of other Osmundales. We consider this shared presence of a distinct peripheral metaxylem siphon in all Osmundales as more significant than the variable tissue composition of the stem core: in the latter, a gradual transition can be drawn from long tracheids (Thamnopteris gracilis) to short parenchyma-like tracheids (most Thamnopteroideae) to short tracheids with interspersed parenchyma (Thamnopteris kidstonii) to parenchyma with interspersed tracheids (e.g., Millerocaulis lutziae, Millerocaulis kolbei, and Claytosmunda beardmorensis) to parenchyma (and secondary sclerenchyma; most Osmundoideae) to fully sclerified stem cores (e.g., Todea) (Fig. 2). Thus, the stele type of thamnopteroid Osmundaceae corresponds structurally and functionally to a siphonostele, regardless of the varying proportions of tracheids or parenchyma it contains. Current terminology and classification of stele architectures does not cover such a type (see Schmid, 1982), and we contend that a descriptive terminology should be used instead of categorical classification in the systematic description of these stele types: “Stem core consisting of […]” as opposed to, e.g., “Stem protostelic.”

Stele

The stele of all Osmundales is characterized by a distinct, peripheral metaxylem siphon (Figs. 1 and 3). Whereas in a few species of Thamnopteroideae this siphon may appear completely entire and uninterrupted throughout its length (e.g., Thamnopteris kidstonii), the departing leaf traces usually leave more or less prominent notches or gaps in the stele periphery (Fig. 3). Furthermore, a protoxylem cluster in the stele may become associated with a parenchyma pocket that increases in size distally and that may eventually break through toward the pith before the leaf trace departs the stele, thus creating a deep notch or embayment along the internal margin of the stele; in these cases, a given stem transverse section (TS) may show one or more inverse U-shaped (parts of) xylem segments with the openings directed toward the pith (e.g., Chasmatopteris principalis; Osmundoideae; Osmundacaulis spp.; Fig. 3).

Figure 3 Diagram showing selected examples of the wide variety of modifications of the stelar xylem siphon in osmundalean stems as seen in cross-section through the stele, together with the respective character scoring used in the matrix (for definition of characters and of character states see text).

Traditionally, the stems of Thamnopteroideae have been described as having a gapless and imperforate stele consistent with their classification as being protostelic (see, e.g., Wang et al., 2014b). However, in most thamnopteroids, the departure of a leaf trace does leave a noticeable gap in the form of at least a shallow peripheral incision in the external surface of the metaxylem siphon (e.g., Thamnopteris schlechtendalii). In C. principalis, this incision may even perforate the xylem siphon completely; thus, except for its stem core consisting presumably of enlarged tracheids rather than parenchyma, the stele of Chasmatopteris equates to the “dictyoxylic siphonostele” found in most other Osmundoideae.

The degree of perforation of the metaxylem siphon is commonly used as a diagnostic feature in taxonomic treatments of Osmundales (Miller, 1971; Tidwell & Ash, 1994; Tian, Wang & Jiang, 2008; Wang et al., 2014b), but authors disagree about the significance of this character (see Vera, 2008; Herbst, 2015). In Osmundales, the stele perforation embraces a continuous spectrum, ranging from a simple, straight, imperforate tube (e.g., Thamnopteris kidstonii) to a highly complex, reticulate network of stelar xylem segments (e.g., Osmundacaulis, Lunea; Fig. 3; Table S1). Whereas classification of end members in this spectrum is rather straightforward, the range of variation represented—especially in the diverse array of Mesozoic osmundoid rhizomes—makes a categorical classification difficult (Vera, 2008). This is further complicated by the fact that the development of complete perforations in the xylem cylinder depends on the ontogenetic stage of the individual specimen and on the level at which it is sectioned (Vera, 2008; see also Seward & Ford, 1903; Wardlaw, 1946, 1947; Hewitson, 1962; Gould, 1970; Cantrill, 1997; Herbst, 2001, 2006; Fig. 4). In addition, the presence of complete leaf-gap perforations may be obscured by taphonomic effects (Kidston & Gwynne-Vaughan, 1907: 760), or perforations may be so short or transect the stele at such an acute angle that no single transverse section of the xylem siphon will show a complete perforation (Sharma, 1973; Vera, 2008; see also Fig. 4).

Figure 4 Diagram illustrating different modes of leaf-trace formation in selected types (A–H) of osmundalean stems, each showing a stem radial section through the centre of the developing trace and the resulting aspects of four successive cross-sections.

Metaxylem in brown; protoxylem strands in blue.

We agree with Vera (2008) that the degree of stele perforation should not be emphasized as a character for generic distinction within Osmundoideae. Considering Osmundales as a whole, highly perforate steles certainly evolved more than once, but are nonetheless characteristic of particular lineages within the Osmundaceae (e.g., Osmundacaulis, Plenasium subgenus Aurealcaulis in Osmundoideae) and Guaireaceae (Guairea in Guaireoideae). This applies also to the opposite extreme: (nearly) imperforate steles are characteristic of Thamnopteroideae in Osmundaceae and of Itopsidemoideae in Guaireaceae. Finally, we argue that an unusually high degree of perforation can also be characteristic of a particular species or even genus within a group with otherwise imperforate or only moderately perforated steles (consider, e.g., the highly perforated stele of Millerocaulis kolbei or the perforate stele of Chasmatopteris in Thamnopteroideae). Altogether, the significance of the degree of stele perforation depends on the context of the individual fossil and its putative relatives.

The peripheral xylem siphon usually measures from 10 to 20, rarely up to 25 tracheid cells in radial thickness; exceptions to this rule are Guairea, Donwelliacaulis, and Osmundacaulis, in which the siphon is particularly thick and ranges from about 30 to, in extreme cases (Donwelliacaulis, Osmundacaulis pruchnickii), more than 70 cells in radial thickness (Table S1). In nearly all Osmundales, the siphon consists of metaxylem tracheids with more or less evenly distributed protoxylem poles in mesarch or subexarch position; in many taxa, developing leaf trace protoxylem clusters become associated distally with pockets of parenchyma. In this respect, Itopsidema, Donwelliacaulis, and Tiania (Guaireaceae: Itopsidemoideae) are unique in that the peripheral xylem siphon consists of a spongy admixture of xylem tracheids with diffusely interspersed “nests” of parenchyma that are not immediately related to protoxylem clusters of early-developing leaf traces (dotted xylem signature in Fig. 3).

Endodermis and phloem are usually only external to the stele. In some taxa, they may occur also internally, and even connect through leaf gaps and completely ensheath individual xylem segments in a given stem transverse section. In these latter types, cortical tissues may then invaginate through the leaf gaps into the pith (or vice versa). Such particularly complex stele types have been referred to as “dissected-siphonostelic” (in cases where endodermal layers connect) and “dictyostelic” (in cases where phloem layers connect) (see Miller, 1971). This differentiation, however, is difficult to use for the classification of extant and extinct Osmundales; in many fossils, imperfect preservation makes it difficult to distinguish between phloem, endodermis and cortical parenchyma and, hence, to identify siphonostelic, dissected-siphonostelic, and dictyostelic conditions (Herbst, 2015). Furthermore, anatomical studies of more than 150 rhizomes of Osmundastrum cinnamomeum have shown that occurrence and configuration of phloem and endodermis are so variable that practically all three of these stele types can occur in just this single extant species; proper recognition of the stele type would then depend on the individual plant and on the position at which it is sectioned (Faull, 1910; Hewitson, 1962; Miller, 1967; Tidwell & Ash, 1994). In most cases, special modifications have been observed to occur below incipient rhizome bifurcations (see Hewitson, 1962; Miller, 1971).

Stem cortex

The structure of the stem cortex (Fig. 5) is an important feature for higher-level taxonomic classification in Osmundales. In Guaireaceae, the cortex is composed primarily of parenchyma with interspersed sclerotic nests or secretory ducts, and is not differentiated into distinct layers. In Osmundaceae, by contrast, it is differentiated into an inner, primarily parenchymatous layer and an outer sclerenchymatous layer (Fig. 5). The composition of the outer cortex is usually homogeneous, but has distinct cylinders of particularly thick-walled fibres surrounding the leaf traces in Leptopteris and Todea (Miller, 1971). The structure of the stem cortex also determines the makeup of the stipe cortex (see there).

Figure 5 Diagram illustrating selected examples for different layering and composition of cortical tissues in osmundalean stems as seen in stem cross-section, together with the respective character scoring used in the matrix (for definition of characters and of character states see text).

Parenchyma in pale yellow; sclerenchyma in light brown; thicker walled fibre patches as black stipples or rings.

Leaf traces

Leaf traces in Osmundales develop from protoxylem poles that appear initially in subexarch to mesarch position in the peripheral metaxylem siphon of the stele (Figs. 4 and 6). At a specific point in the upward and outward course of the leaf trace through the stem, the protoxylem becomes “exposed” at the adaxial surface of the trace (thereby making the trace endarch), and begins to divide as the trace widens and its curvature increases. Once outside the stem, the trace forms a strongly adaxially curved, endarch, entire vascular bundle with incurved or recurved tips (Figs. 6 and 7). The number of initial protoxylem poles per leaf trace and the points of exposure and first division of the protoxylem in the stem represent important diagnostic criteria (Fig. 6).

Figure 6 Diagram illustrating selected examples for different modes of leaf-trace formation and development in osmundalean stems each in the form of four simplified aspects of successive cross-sections (at the level of the stele, upon departure from the stele, mid-way through the cortex, and upon departure from the stem), together with the respective character scoring used in the matrix (for detailed explanation and for definition of characters and of character states see text).

Metaxylem in brown; protoxylem strands in blue; parenchyma in pale yellow.

Figure 7 Diagram illustrating selected anatomical features of the stipes of Osmundales together with the respective character scoring used in the matrix (for detailed explanation and for definition of characters and of character states see text).

Metaxylem (characters 26 and 27) in dark brown; parenchyma in pale yellow; sclerenchyma in brown; scattered isolated sclereids (characters 32 and 40) indicated as brown stipples; patches or clusters of particularly thick-walled fibres in black.

Miller (1971) described and illustrated three modes of leaf trace and gap formation in Osmundaceae, which have been adopted in many comparisons and included in morphological matrices. In reference to earlier descriptions (Kidston & Gwynne-Vaughan, 1910; Miller, 1967), two of these examples were labelled “delayed gap” and “immediate gap,” respectively, depending on whether the gap breaks through to the pith upon or after the departure of the trace from the stele; the third example was termed “Plenasium-type gap,” characterized primarily by the fact that each leaf trace is formed from two independent protoxylem poles from two adjacent xylem segments. Tidwell & Ash (1994) illustrated three additional, complex types of leaf trace formation occurring in Osmundacaulis species, two being termed “semi-plenasoid” (Tidwell & Jones, 1987) and one “plenasoid”; these types are superficially similar to those of Plenasium, but differ in that all originate from just a single protoxylem pole and not from two independent poles as in Plenasium.

However, these aforementioned types should not be considered as fixed categories, but rather regarded as selected examples of the wide range of leaf trace modes that are now known to occur in osmundalean axes (Figs. 4 and 6). Again, we propose to use descriptive (point-by-point) rather than generalized categorical—and to a large degree interpretative—terminology to accommodate this structural diversity.

Important features for characterizing the development of the leaf trace include (1) the number of protoxylem poles from which a leaf trace is formed; (2) the position in which the protoxylem first appears in the stele; (3) the point at which the protoxylem becomes exposed at the adaxial surface of the trace, i.e., at which the trace becomes endarch; (4) the point at which the protoxylem undergoes first (and subsequent) divisions; and (5) changes in the overall shape of the leaf trace in its course through the stem. Consequently, we score leaf trace development in our matrix in a series of eight characters that describe the number and position(s) of protoxylem poles in the stele, upon departure of the trace, in the central part of the cortex, and upon departure from the stem (Fig. 6).

Stipe bases

Stipe bases of all Osmundales contain a single vascular bundle that is strongly curved adaxially, i.e., with the opening directed toward the stem. In Osmundaceae, the adaxial margins of the bundle can be straight or incurved, giving the bundle a characteristic horseshoe-shape in transverse section; in Guaireaceae, the margins of the bundle are typically recurved or constricted, giving the bundle an inverse omega-shape or a flask-shape in transverse section (Fig. 7).

The anatomy of the stipe derives from the tissues that the leaf trace passes in its course through the stem. The structure of the stipe cortex thus mirrors that of the stem cortex. Consequently, the stipes of Osmundaceae have an inner parenchymatous groundmass that is surrounded by a robust sclerenchyma cylinder, and these resistant stipe bases form a thick persistent mantle around the stem. Furthermore, the stipes in Osmundaceae typically have a pair of basal stipular wings. The stipes of Guaireaceae, by contrast, are primarily parenchymatous like the stem cortex; presumably owing to the lack of mechanical resistance, they do not form a persistent mantle and are rarely preserved. Based on the occurrence of simple petioles lacking stipular wings in Lunea jonesii, it has been suggested that guaireaceaean stipes lack stipular wings in general (Tidwell, 1991). The stipe bases of Itopsidema vancleavii and of the poorly known Bathypteris rhomboidea bear large, multicellular spines (Daugherty, 1960; see Miller, 1971).

Of special importance for the classification of Osmundaceae is the occurrence, distribution, and distalward differentiation of sclerenchyma at particular levels and in particular tissue regions of the stipe. Characteristic sclerenchyma configurations can be found in the concavity of the vascular bundle, the inner cortex surrounding the bundle, the sclerenchyma cylinder (i.e., the stipe outer cortex), and the stipular wings (Fig. 7). In the primarily parenchymatous groundmass of the (inner) cortex and the stipular wings, sclerenchyma may occur in the form of isolated scattered cells (commonly referred to as “fibres”) or of larger masses whose positions and shapes may change from the base of the stipe distally (Fig. 7). Typical patterns include, for instance: solid or interrupted bands of sclerenchyma lining the adaxial or the abaxial side of the trace concavity; one central mass or two lateral masses of sclerenchyma inside the trace concavity, or one or several elongate patches of sclerenchyma in the centre of each stipular wing (Fig. 7).

Of further importance is the occurrence and development of masses of particularly thick-walled fibres within the sclerenchymatic outer cortex of the petiole, i.e., in the “sclerenchyma ring” or “sclerenchyma cylinder” of the stipe (Fig. 7). In all modern Osmundaceae (genera with extant representatives, i.e., Todea, Leptopteris, Claytosmunda, Osmundastrum, Osmunda, and Plenasium), the composition of this sclerenchyma cylinder is heterogeneous and shows distinctive configurations of patches of thick-walled sclerenchyma cells. Differentiation of the cylinder typically begins with the development of an arch or crescent of thicker-walled cells lining the abaxial periphery of the ring in cross-section. This abaxial arch then begins to differentiate further into characteristic patterns that are diagnostic of particular genera, subgenera, and species (Fig. 7): in extant Todea, Leptopteris, and Plenasium, the arch differentiates distally into a complete ring of thicker-walled cells at the outer periphery of the sclerenchyma ring; in Osmundastrum, it differentiates into two lateral masses and one abaxial mass; in Claytosmunda, it differentiates into just two lateral masses; and in extant Osmunda, it develops initially into two lateral masses that, further along the stipe, conjoin along the adaxial side of the ring to form an adaxial arch. Additional patterns may occur in other fossil representatives of modern Osmundoideae, although most of the Mesozoic fossil rhizomes of modern Osmundoideae typically have configurations that are similar to those represented in the extant Osmundastrum and Claytosmunda.

Roots

Roots in all Osmundales contain a single, diarch vascular bundle with two opposite protoxylem poles in exarch position. In a few Permian Osmundaceae (i.e., Chasmatopteris, Palaeosmunda, some Thamnopteris), the root vascular bundles may also (rarely) be triarch (see Gould, 1970). In most cases, the root cortex consists of parenchyma surrounded by a sclerenchymatic sheath. In Osmundaceae, roots generally arise from the stele or laterally from departing leaf traces, whereas they may also arise abaxially from leaf traces further out in the cortex in Guaireaceae. Outside the stem, the roots in many arborescent taxa accumulate to form a dense root mantle (e.g., most Guaireaceae; see, e.g., Tidwell & Ash, 1994), or may gradually replace softer tissues in the mantle of stipe bases (in Osmundaceae; see, e.g., Miller, 1971).

Materials and Methods

A novel morphological character matrix

In order to resolve systematic relationships among extant and fossil Osmundales based on axis anatomy and to aid description and taxonomic assessment of forthcoming material, we developed a novel morphological character matrix (Files S1 and S2). The basic organization of this matrix follows that of a previous study (Bomfleur, Grimm & McLoughlin, 2015), which in turn was based on that of Miller’s (1971) analyses. Our previous matrix consisted of 34 operational units and 33 binary or ternary characters; it was designed specifically for placing extinct members of modern Osmundaceae genera into a phylogenetic reconstruction incorporating molecular data from extant species. The new matrix presented below (129 operational units and 45 unweighted binary or ternary characters; Files S1 and S2) is, by contrast, designed to better accommodate the morphological disparity in extinct Osmundales as a whole. Its main purpose is to provide a standard framework for the description, identification, and classification of “natural” taxa, which may provide a more solid basis for reconstructing the evolution of the order. An annotated version of the matrix (NEXUS-formatted) is included in the Supplemental Data Archive (SDA; Grimm, Bomfleur & McLoughlin (2017) available from Dryad Digital Repository http://dx.doi.org/10.5061/dryad.270gs) and included in Supplemental Information linked to this paper.

Annotated character list

Character 1: Stem core: tracheids: (0) absent; (1) accessory; (2) forming main tissue.

Character 2: Stem core tracheids (separated and modified from character I.A of Miller (1971)): (0) long (up to 10 or more times longer than broad); (1) short (one to three times, rarely up to five times longer than broad).

Character 3: Stem core: parenchyma: (0) absent; (1) accessory; (2) forming main tissue.

Character 4: Stem core: sclerenchyma: (0) absent; (1) present, scattered; (2) variable, up to entirely sclerified (character II.B of Miller (1971) and character 8 of Bomfleur, Grimm & McLoughlin (2015)).

Character 5: Stele: (0) solid or diffusely heterogeneous; (1) heterogeneous with a distinct peripheral xylem siphon (Modified from character I.A of Miller (1971) and from character 4 of Bomfleur, Grimm & McLoughlin (2015)).

Character 6: Thickness of stelar xylem siphon: (0) <30 tracheids thick; (1) >30 tracheids thick (Modified from character II.C of Miller (1971) and from character 9 of Bomfleur, Grimm & McLoughlin (2015)).

Character 7: Leaf gaps (i.e., external notches or incisions into the metaxylem siphon above departing leaf traces): (0) absent; (1) shallow (shallower than leaf trace thickness); (2) prominent (deeper than leaf trace thickness) (Replaces characters 4 and 5 of Bomfleur, Grimm & McLoughlin (2015)).

Character 8: Degree of stele perforation: (0) (nearly) imperforate: ≤3 complete perforations; (1) moderately to densely perforated: 4–15 complete perforations, number of complete perforations per millimeter stele perimeter (PmmS) 0.11–2.45; (2) highly perforated: ≥16 complete perforations and PmmS ≥ 0.30 (Modified from character III.R of Miller (1971); replaces characters 4, 5, and 10 of Bomfleur, Grimm & McLoughlin (2015)).

This categorization is based on the results of k-median clustering analyses undertaken with Cluster 3.0 (de Hoon et al., 2004; see Bomfleur, Grimm & McLoughlin (2015) for configuration details) and k = 3 (see File S1) based on (1) the absolute number of complete perforations and (2) the relative number of complete perforations per mm of stele perimeter (PmmS) expressed in the formula, PmmS=N(CP)max(d(Stele)*π)(Formula 1)

in which N(CP)max is the maximum number of complete perforations in a given transverse section of the stelar xylem cylinder; d(Stele) the stele diameter (see Table S1).

The PmmS value was included to check for potential cylinder thickness influence on the number of perforations. For instance, xylem cylinders of Guairea have a moderate maximum number of complete perforations (12–18) but very large steles. Consequently, they appear much less perforated than, e.g., Millerocaulis herbstii and Osmunda shimokawaensis with the same maximum number of complete perforations (18), which are much more densely spaced (PmmS = 1.42 compared to 0.11–0.13 in Guairea). Final values range between entirely imperforate to highly perforate with 75 complete perforations (Plenasium dakotense) and 2.45 perforations per mm stele perimeter (Millerocaulis amarjolensis). Inferred cut-off values of the clusters were ≤4, 5–15, and ≥16 for complete perforations and ≤0.56 and ≥0.6 (k = 2) or ≤0.32, 0.36–0.95, and ≥0.99 (k = 3), respectively.

Most operational units in the first cluster have no or only few complete perforations and accordingly low PmmS values (≤0.19); those were scored as “(nearly) imperforate” (Table S1). The two operational units with four complete perforations have much higher PmmS values (0.64, 0.75), comparable to values found frequently in operational units with five or more perforations, hence, these were scored as “moderately perforate.” One of the 35 operational units with ≥16 complete perforations (scored as “highly perforate”) has an extremely low PmmS value (Guairea braziliensis, 18 complete perforations; PmmS = 0.11), hence this species was scored as “moderately perforate” by analogy with its congeners with similar PmmS but fewer complete perforations (12–14). Next lowest PmmS values in the group of operational units scored as “highly perforate” are found in three Osmundacaulis species (PmmS = 0.30–0.32; value gradually increases in other species of the same genus). The 5–15 complete perforations in the remainder (49 operational units) can be widely or densely spaced. We categorized these as “moderately to densely perforate.”

Character 9: Internal embayments in xylem siphon (indicative of an incipient complete leaf-gap formation): (0) none; (1) fewer than peripheral incisions; (2) as many as or more than peripheral incisions (This descriptive character records the relative degree to which a complete perforation extends down- and inward into the stele, resulting in the occurrences of inverse U-shaped transverse sections of (parts of) xylem segments).

Character 10: Parenchyma interspersed in stelar xylem, creating “spongy” structure: (0) absent; (1) present (New character that was included to recognize the different tissue composition in the steles of Itopsidema, Donwelliacaulis, and Tiania).

Character 11: Dissected condition, i.e., with external and internal endodermal layers connecting through leaf gaps: (0) absent or very rare; (1) usually present (Modified from character II.D of Miller (1971) and character 7 of Bomfleur, Grimm & McLoughlin (2015)).

Character 12: Dictyostelic condition, i.e., with external and internal phloem connecting through leaf gaps: (0) absent or very rare; (1) usually present. (Modified from character II.D of Miller (1971) and from character 6 of Bomfleur, Grimm & McLoughlin (2015)).

Character 13: Relative thickness of cortex (RTC): (0) <1.2; (1) [1.2; 2.8]; (2) >2.8. The relative thickness of the cortex was quantified using Formula 2. As in the case of Character 8, a k-median clustering with three classes (k = 3) was used to define thresholds for each character state. RTC=d(Stem)max−d(Stele)maxd(Stele)max(Formula 2)

in which d(Stem)max is the maximum stem diameter; d(Stele)max the maximum stele diameter.

Final values for RTC ranged between 0.14 in Osmundacaulis zululandensis and 13.62 in B. rhomboidea (Table S1).

Character 14: Third, entirely parenchymatous inner cortex layer: (0) absent; (1) present (New character accounting for the distinct third cortex layer in outgroup taxa).

Character 15: Composition of the inner cortex (the inner one of two main layers): (0) parenchyma only; (1) parenchyma with scattered sclerenchyma cells; (2) parenchyma with a patch of fibres adaxial to each departing leaf trace (Character II.G of Miller, 1971)

Character 16: Sclerenchymatous outer cortical layer: (0) absent; (1) thinner than parenchymatous layer; (2) thicker than parenchymatous layer (Modified from characters I.B and II.F of Miller (1971) and from character 16 of Bomfleur, Grimm & McLoughlin (2015)).

Character 17: Sclerenchymatous outer cortical layer: (0) homogeneous; (1) heterogeneous with a distinct cylinder of thicker-walled sclereids surrounding each leaf trace (New character accounting for the heterogeneous composition of the outer cortex in Todea and Leptopteris; see Hewitson, 1962; Chandler, 1965; Miller, 1967, 1971; compare character 1243 of Jud, Rothwell & Stockey, 2008).

Character 18: Number of initial protoxylem poles per leaf trace: (0) one; (1) two; (2) more than two (Character 16 of Wang et al. (2014b) and character 11 of Bomfleur, Grimm & McLoughlin (2015)).

Character 19: Leaf trace protoxylem initiation in stele: (0) subexarch in peripheral bulge; (1) mesarch (New character accounting for the different position of protoxylem in the stems of Thamnopteroideae; see Miller, 1971).

Character 20: Number of leaf trace protoxylem strands upon departure from stele: (0) one; (1) two; (2) more than two (Modified, descriptive coding of characters I.F, II.H, and III.E of Miller (1971)).

Character 21: Position of leaf trace protoxylem strands upon departure from stele: (0) lateral; (1) mesarch; (2) endarch (Modified from character I.D of Miller (1971)).

Character 22: Shape of leaf trace immediately after departure from stele: (0) oblong or slightly curved adaxially; (1) strongly curved adaxially; (2) two individual segments (New character based on observations by Chandler (1965) and Miller (1971)).

Character 23: Number of leaf trace protoxylem strands in central part of cortex: (0) one; (1) two; (2) more than two (Modified, descriptive coding of characters I.F, II.H, and III.E of Miller (1971)).

Character 24: Position of leaf trace protoxylem strands in central part of cortex: (0) mesarch; (1) endarch (Expanded from modified character I.D of Miller (1971)).

Character 25: Number of leaf trace protoxylem strands upon departure from stem: (0) one; (1) two; (2) more than two (Modified, descriptive coding of characters I.F, II.H, and III.E of Miller (1971)).

Character 26: Shape of stipe bundle immediately after departure from stem: (0) oblong or only slightly curved adaxially; (1) strongly curved adaxially.

Character 27: Lateral tips of stipe bundle immediately after departure from stem: (0) straight or incurved; (1) recurved.

Character 28: Sclerenchyma lining stipe bundle abaxially: (0) absent; (1) interrupted; (2) continuous.

Character 29: Sclerenchyma in stipe bundle concavity: (0) absent; (1) free mass; (2) lining band.

Character 30: Sclerenchyma in stipe bundle concavity, special states: (0) solid; (1) interrupted/scattered; (2) bifurcating.

Character 31: Appearance of sclerenchyma in stipe bundle concavity: (0) in stipe only; (1) extending proximally into stem cortex; (2) extending proximally into stele.

Character 32: Scattered sclerenchyma in inner cortex of stipe: (0) absent; (1) present.

Character 33: Stipe cortex with distinct outer sclerenchyma cylinder: (0) absent; (1) present.

Character 34: Sclerenchyma cylinder of stipe base: (0) homogeneous or diffusely heterogeneous; (1) heterogeneous, having a distinct abaxial arch of thick-walled fibres in basal cross-sections.

Character 35: Abaxial arch developing distally into a complete ring: (0) absent; (1) present.

Character 36: Abaxial arch differentiating distally into two lateral masses: (0) absent; (1) present.

Character 37: Abaxial arch (or two lateral masses) differentiating distally into two lateral masses and one abaxial mass: (0) absent; (1) present.

Character 38: Two lateral masses developing distally into an adaxial arch: (0) absent; (1) present.

Character 39: Stipular extensions: (0) none; (1) wings; (2) spines.

Character 40: Scattered sclerenchyma fibres in stipular wings: (0) absent; (1) present.

Character 41: Distinct sclerenchyma masses in stipular wings: (0) absent; (1) one mass in each wing; (2) two or more masses in each wing.

Character 42: Shape or arrangement of distinct sclerenchyma masses in cross-sections of stipular wings: (0) irregular; (1) elongate along lateral extent of stipular wing.

Character 43: Point of root emergence: (0) always from stele; (1) from leaf trace upon emission from stele; (2) from leaf trace further out in cortex.

Character 44: Number of roots per leaf trace: (0) one, sporadically two; (1) two, sporadically one.

Character 45: Predominant orientation of roots in mantle cross-section as an indicator of arborescence: (0) primarily vertical; (1) primarily radial.

Operational units

The more differentiated “operational units” (typically taxa) are coded into a character matrix, the better-substantiated the inferences will be. In this respect, the increasing use of the genus as the basic category in evolutionary studies is problematic because genera are often arbitrarily circumscribed, non-equivalent concepts that do not exist in reality (Hendricks et al., 2014). Arbitrary blending of the varied morphological information from a pool of species or specimens into a single “generic-level composite” eliminates potentially informative data from the analysis, just like subjective exclusion of operational units does. The optimal solution would be to use not just species, but specimens as the basic operational units in a morphological character matrix (see, e.g., Upchurch, Tomida & Barrett, 2004; Tschopp, Mateus & Benson, 2015). However, an all-specimen-level analysis of fossil Osmundales is impossible at present because species descriptions that are based on more than one specimen usually do not list specimen data separately but summarize all morphological information available in the form of ranges of values, ratios, and dimensions.

Nevertheless, we aim to include as much in-group information in our analysis as possible. Therefore, the matrix includes not just all currently accepted species, but also additional operational units that contain independently coded information either from synonymous species or from separate, individually documented fossil records (Table S1; Files S1 and S2). As a result, (1) Plenasium dowkeri (Carruth.) Bomfleur, G.W.Grimm & McLoughlin sensu Miller is separated into two operational units: Plenasium dowkeri (Carruth.) Bomfleur, G.W.Grimm & McLoughlin sensu Chandler (see Chandler, 1965) and Plenasium chandleri (Arnold) Bomfleur, G.W.Grimm & McLoughlin (see Arnold, 1952); (2) Guairea carnieri (J.Schust.) R.Herbst is separated into two operational units Guairea carnieri (J.Schust.) R.Herbst sensu Miller (see Schuster, 1911; Miller, 1971) and its junior synonym Osmundacaulis braziliensis (H.N.Andrews) C.N.Mill. (see Andrews, 1950; Miller, 1971); (3) Millerocaulis dunlopii (Kidst. & Gwynne-Vaughan) Tidwell is separated into Millerocaulis dunlopii (Kidst. & Gwynne-Vaughan) Tidwell sensu Kidston & Gwynne-Vaughan (1907) and its junior synonym Osmundites aucklandicus P.Marshall (see Marshall, 1926); (4) Osmundacaulis hoskingii R.E.Gould is separated into Osmundacaulis hoskingii var. hoskingii R.E.Gould and Osmundacaulis hoskingii var. tabulatus R.E.Gould (see Gould, 1973); (5) Millerocaulis australis (E.I.Vera) E.I.Vera is separated into two operational units, one being based on the original type specimen (Vera, 2007) and the other being based on the later report of an additional specimen with slightly different features (Vera, 2010); (6) Osmundastrum precinnamomeum (C.N.Mill.) Bomfleur, G.W.Grimm & McLoughlin, which some authors consider to be a junior synonym of Osmundastrum cinnamomeum (L.) C.Presl (see Serbet & Rothwell, 1999; Matsumoto & Nishida, 2003), is coded as a separate operational unit; and (7) Osmundastrum cinnamomeum is further separated into four operational units based on information from extant material (see Miller, 1971) and from three individual fossil occurrences of this species from the Neogene of Japan (Matsumoto & Nishida, 2003) and from the Cretaceous and Neogene of North America (Serbet & Rothwell, 1999; Miller, 1967).

The selection of extant species is adopted from Miller (1971); the coding of these species is based on Miller’s (1971) character lists but has, whenever possible, been supplemented with additional information from earlier literature on the anatomical structure of Osmundaceae (Faull, 1901; Seward & Ford, 1903; Bower, 1911, 1926; Gwynne-Vaughan, 1911, 1914; Hewitson, 1962; Miller, 1967). Sources for revised data are provided in the form of comments either in the final matrix (Files S1 and S2) or in the spreadsheet listing morphological features (Table S1).

Outgroup operational units have been selected based on earlier suggestions that Grammatopteris and Rastropteris belong to a group of filicalean ferns closely allied to Osmundales (see Sahni, 1932; Miller, 1971; Galtier et al., 2001; Rößler & Galtier, 2002). In contrast to the most recent phylogenetic analysis (Wang et al., 2014b), we retain Grammatopteris as an outgroup operational unit because it does not show sufficient diagnostic characteristics to warrant assignment to Osmundales. Furthermore, we included the fossil Grammatocaulis donponii as an additional outgroup operational unit because of its close similarity to Grammatopteris (Tidwell & Rozefelds, 1990).

In order to mitigate noise and poor definition of pairwise distances from uninformative operational units with too much missing data, we excluded those operational units with more than 60% missing characters (an arbitrary cut-off value) from the analyses. This applies to Anomorrhoea fischeri, Claytosmunda nathorstii, Osmunda kidstonii, Osmundacaulis janae, and Todea papuana (File S3). The status of Anomorrhoea fischeri is problematic because too many diagnostic features essential for systematic classification are missing (see Miller, 1971). Claytosmunda nathorstii, Osmunda kidstonii, and Osmundacaulis janae are only known from petiole sections (see Miller, 1971; Tidwell & Pigg, 1993) that, however, permit unambiguous generic assignment. No detailed information has been published on the anatomical structure of Todea papuana; the species remains in our matrix solely because it yielded molecular information used in an earlier version of the matrix (Bomfleur, Grimm & McLoughlin, 2015; Grimm et al., 2015).

Data acquisition and character coding

The character coding for the individual operational units is based on data from published literature. Wherever possible, the coding was compiled on the basis of the original protolog of the particular taxon. Data adopted from diagnoses and descriptions were critically checked with information provided by accompanying illustrations. In case information was lacking from the original text, coding of the operational unit was completed as far as possible using accompanying illustrations or additional information from later references; in cases of conflict between figured material and text descriptions, data were corrected according to the illustrations. Common problems that we encountered concern, for instance, erroneous dimensions of stem features resulting from the inconsistent use of the terms “stele,” “stem” (i.e., stele plus cortex), and “trunk” or “rhizome” (i.e., stele plus cortex plus mantle). Another source of error is the inconsistency in counting stelar xylem segments, depending on whether they are counted as separate already when they are deeply divided or only when they are entirely isolated from another (“Hewitson’s method”). In any case, corrected data are indicated as such in the form of annotations either in the spreadsheet compilation of morphological features (Table S1) or in the final matrix file (Files S1 and S2).

Phylogenetic analysis

Use of phylogenetic networks

Classification of fossils should be based on morphology rather than theoretical concepts. The study of evolutionary processes, however, requires phylogenetically meaningful (as opposed to merely phenetic) “natural” taxa, and the definition of such “natural” groups requires an implicit phylogenetic framework via the recognition of evolutionary lineages of common ancestry (Darwin, 1859; Haeckel, 1866; Simpson, 1945; Fig. 8A). In the absence of extant representatives of most Osmundales lineages, and hence, molecular data for these groups, the framework can only be based on morphological traits—characters of uncertain evolutionary relevance (Scotland, Olmstead & Bennett, 2003; Wiens, 2004) but with substantial impact on phylogenetic inferences (Hillis & Wiens, 2000; Wiens, 2004; Müller & Reisz, 2006; Cobbett, Wilkinson & Wills, 2007).

Figure 8 Diagram illustrating different naming concepts (classifications) for extinct and extant representatives of a phylogenetic lineage.

(A) A hypothetical phylogeny of three extant genera (taxa A, B, and C) and their fossil relatives including actual ancestors and extinct sister lineages. (B) A traditional phenetic classification, in which all species that share a similar morphology (morphospace) are classified together regardless of their phylogenetic relationship; note that the resulting genera can thus be para- or polyphyletic. (C) An incomprehensive cladistic classification accepting only holophyletic taxa; note that one fossil has to be accommodated in its own genus and that two other fossils (members of the lineage ancestral of genera A–B and genera A–D, respectively) must be excluded (“gen. indet.”) to avoid violation of the principle. (D) Comprehensive cladistic classification accepting only holophyletic taxa; the problem of unclassifiable fossils is avoided by including all fossils and extant species of the modern lineage in one large genus and by introducing lower taxon ranks to distinguish holophyletic (sub)groups. (E) A phenetic–cladistic “hybrid” classification common in palaeontology in which selected fossils are assigned to extant genera, whereas others are assigned to artificial taxa (“parataxa,” “fossil taxa,” formerly “form taxa” or “morphotaxa”) in order to maintain holophyly of extant taxa; note that an arbitrary and inconsistent date-line separates the realms of modern taxonomy and fossil taxonomy, and that fossils can effectively build up paraphyletic or polyphyletic taxa. (F) Evolutionary classification in which paraphyletic taxa are accepted as valid taxonomic units; note that the resulting classification takes phylogenetic history into account and at the same time produces only monophyletic taxa that can be diagnosed by a distinct morphology; potential new fossils can be easily incorporated and nomenclatural stability is maintained; taxa are informative; and inflationary numbers of single-specimen taxa or of taxonomic ranks are avoided.

Signals from morphological matrices are generally complex, especially if fossil (presumably primitive) and modern (presumably derived) taxa from different time periods are pooled (Cobbett, Wilkinson & Wills, 2007; Denk & Grimm, 2009). With more than 100 operational units but fewer than 50 characters, our matrix dimensions are far from optimal. We refrained from filtering homoplastic characters and those that are generally variable within potential lineages because even those might still prove useful for discrimination at and above species level. Phylogenetically unsorted, homoplastic signals (incompatible with the true tree) can outcompete phylogenetically informative (compatible with the true tree) signals, eventually resulting in erroneous tree inferences, especially if parsimony is used as the optimality criterion (Scotland & Steel, 2015). Incongruent signals, i.e., in which different sets of characters prefer contrasting topologies, are common and pose an additional problem for phylogenetic inference. On the other hand, taxa sharing a common origin are typically more similar to each other than those not sharing a common origin (e.g. Felsenstein, 2004).

Altogether, we tried to keep the phylogenetic analyses as simple and straightforward as possible: we used a matrix of unsorted and unweighted characters to infer neighbour-net splits graphs (Bryant & Moulton, 2002, 2004) based on simple (Hamming) pairwise distances. Neighbour-nets are designed to better handle incompatible signals, and are more sensitive with respect to actual ancestor–descendant relationships than are dichotomous trees (Spencer et al., 2004; Denk & Grimm, 2009). The distance between two tips in a neighbour-net reflects the actual distance value, which is not necessarily the case in dichotomous trees (Bryant & Moulton, 2004; Huson & Bryant, 2006).

Analytical procedure

Neighbour-nets were inferred from pairwise Hamming (mean) distance matrices, and were visualized using SplitsTree v. 4.13 (Huson & Bryant, 2006). Distances were computed with PAUP* v. 4b10 (Swofford, 2002) using the default settings on the complete set of operational units and on selected taxon subsets (lineages; see File S3 for details). As pairwise distances become increasingly unrepresentative with the proportion of missing data, operational units with more than 60% undefined characters (Anomorrhoea fischeri, Claytosmunda nathorstii, Osmunda kidstonii, Osmundacaulis janae, Todea papuana) were excluded from the analyses. For each set analysed, we further excluded all invariant (“constant”) characters before calculating the pairwise distances so that the resulting distance matrices and neighbour-net splits graphs are based only on those characters that are variable (and thus effective) within the focal group; hence, a 0.1 distance between two operational units indicates that they differ in 10% of those characters that are variable within the selected set of operational units. This procedure allowed us to focus on particular aspects of morphological diversity in osmundalean lineages. Furthermore, it helped interpret bootstrap support values as approximations (in the ideal case) of the proportion of characters supporting a given phylogenetic split (i.e., a clade in a rooted phylogenetic tree) (File S3).

The distance quality (“tree-likeness”) of the signals from the entire matrix and from taxon subsets was estimated using the matrix Delta Value (mDV) and the range of individual Delta Values (iDV) within (Holland et al., 2002; Auch et al., 2006; Göker & Grimm, 2008; File S3). Delta values summarize the weight proportion between the two possible, competing edge bundles for quartets of operational units (“four-taxon subsets”), either for the entire matrix (in case of the mDV) or for all quartets including a selected recombinant operational unit (in case of iDV) (see Holland et al., 2002, for the formula). A Delta Value ∼0 indicates high distance quality and thus a highly treelike signal; a Delta Value ∼1 indicates poor distance quality with highly ambiguous signals, with the resulting tree approaching the form of a star with equally long terminal branches for all taxa.

To estimate robustness of support for inferred (or hypothesized) relationships, we used non-parametric bootstrapping (BS; Felsenstein, 1985) under three optimality criteria: maximum likelihood (ML), least-square (LS), and parsimony. For ML–BS, we used the fast BS implemented in RAxML v. 8.2 (Stamatakis, Hoover & Rougemont, 2008; Stamatakis, 2014; option -x); for ML optimization, we used Lewis’ (2001) Mk substitution model for categorical characters, allowing for between-site variation modeled as a Gamma distribution. For LS–BS, we used the BioNJ modification (Gascuel, 1997) of the neighbor-joining (NJ) algorithm (Saitou & Nei, 1987) implemented in PAUP* (Search = NJ/BioNJ). BS under parsimony was performed using PAUP* following Müller (2005): the “MulTrees” option was deactivated and the heuristic search was set to a single tree per BS replicate, which was optimized using the default TBR (tree-bisection-and-reconnection) branch swapping (Search = Heuristic, NRep = 1, AddSeq = Furthest). BS supports (split frequencies) are based on 10,000 pseudoreplicates for ML, LS/NJ, and parsimony. The ambiguous signals in the BS pseudoreplicate samples can be studied using bootstrap consensus networks (Schliep et al., 2017; “bipartition networks,” Grimm et al., 2006), a special form of consensus network (Holland & Moulton, 2003) in which the edge lengths are proportional to the frequency of the corresponding phylogenetic split (taxon bipartition) in the underlying BS pseudoreplicate sample (option “count” in SplitsTree). For clarity, all splits with a frequency of <10% were filtered.

All (raw) files for analyses (input, batch, and results files) are provided in the SDA (Supplemental Data Archive; Grimm, Bomfleur & McLoughlin (2017), available from the Dryad Digital Repository http://dx.doi.org/10.5061/dryad.270gs).

Paraphyletic groups as valid taxonomic units

Classification was historically based on form regardless of evolutionary relationships (see Fig. 8B). By contrast, cladistics-based systematics—today the prevalent school of thought in systematic biology—accepts as valid taxonomic units only holophyletic groups (i.e., “monophyletic” in the sense of Hennig, 1950, and not of Haeckel, 1866; see Ashlock, 1971; Mayr & Bock, 2002; Hörandl, 2006, 2007). Whether a group is holophyletic is either observed in the form of at least one unique and derived (“synapomorphic”) trait shared by all its members (Hennig, 1950) or inferred from a cladogram or phylogram, in which holophyly—the inclusive common ancestry of a group of taxa—is defined by the fact that two or more taxa comprise a single complete subtree (clade) in a rooted phylogenetic tree (Farris, 1983). In the case of morphological matrices, the cladogram used as the basis for classification is usually inferred using parsimony as the optimality criterion, presumably for no other reason than to be consistent with traditional assertions by early cladists (Felsenstein, 2001: 466).

Albeit theoretically appealing owing to its simplicity and its sense of objectivity and reproducibility, the ideal of a classification that accepts only holophyletic groups has severe disadvantages in practice (Mayr & Bock, 2002; Hörandl, 2006, 2007; Van Wyk, 2007; see Figs. 8C–8E). Just a single newly scored trait in a morphological matrix may change a clade in a phylogenetic tree into a grade or vice versa, thwarting the aim of nomenclatural and taxonomic stability (compare the results of Rothwell, Crepet & Stockey, 2009, with those of Crepet & Stevenson, 2010); imbalanced scoring of traits, homoplasy, and long-branch attraction can result in the recovery of artificial clades that are polyphyletic in reality (Scotland & Steel, 2015); new evidence may reveal that a synapomorphy is a symplesiomorphy or even a convergently evolved trait; and insufficient numbers of derived diagnostic features in ancestral members of an evolutionary lineage may obstruct the recovery of a clade, which also applies to extinct sister lineages of extant groups. Conversely, extinct groups might have independently evolved derived features that occur today only in a single surviving lineage; hence, aut- or synapomorphies of modern taxa may be convergences over time (Fig. 8A). Thus, comprehensive morphological matrices that include extinct and extant organisms provide a combination of signals that is generally non-treelike, i.e., too complex to enable the inference of a single phylogenetic tree as an accurate representation of the systematic relationships of its taxa. Branches instead become unstable, support is generally low, and topologies are drastically affected by character coding (compare, e.g., comprehensive morphology-based seed-plant phylogenies obtained by Hilton & Bateman, 2006; Friis et al., 2007; Rothwell, Crepet & Stockey, 2009; Crepet & Stevenson, 2010). These problems render any strict cladistic-phylogenetic classification inherently unstable in the case of fossil plants or plant groups with fossil and extant members. Even if the evolutionary pathways were known in detail (Fig. 8A), a cladistic classification, applied consistently above the species level, remains problematic. If each genus would be inclusive for extinct and extant members of a subtree that can be diagnosed by one or a few unique characters, or characteristically conserved suites of characters, some fossils (such as ancestors of more than a single extant genus) would need to remain unnamed (Fig. 8C). Alternatively, all species/specimens including the earliest representative of the lineage would need to be included in one large and morphologically disparate genus, to the effect that phylogenetic-unambiguously resolved extant genera would need to be renamed and “down-graded” to subgenera (Fig. 8D). As a consequence, many classifications of lineages with fossil and extant representatives are chimeras (Fig. 8E) with the modern taxa classified according to molecular-data-derived (i.e., assumedly holophyletic) clades and the remaining fossil specimens being assigned variably to modern lineages or to more- or less artificial “form-genera,” “morphotaxa,” or “fossil taxa,” for which holophyly need not be established.

An important step toward resolving these problems may be to accept (potentially) paraphyletic taxa as valid taxonomic units following the concept of “evolutionary classification” (see, e.g., Mayr & Bock, 2002; Hörandl, 2006; Van Wyk, 2007: Fig. 8F). For instance, following detailed comparative analysis, it should be valid to assign an ancestor of a group of extant genera to the paraphyletic-per-definition stem genus of these genera if it shows the relevant (lack of) diagnostic characters. Paraphyletic or not, all taxa in our systematic concept are considered natural, monophyletic taxa that share a single common origin (monophyly in the original Haeckelian sense) and that correspond to evolutionary lineages—ancestral and modern ones—within Osmundales.

Taxonomic resolution

The degree of taxonomic resolution and the number of ranks applied below the rank of order vary in our classification depending on the degree of detail to which one taxon can be discriminated from others (see Fig. S1). Highest taxonomic resolution with up to six ranks (Family, Subfamily, Tribe, Subtribe, Genus, and Subgenus) is achieved only for modern members of Osmundaceae, i.e., those genera with (also) extant species. Extinct, but comparatively well-known lineages are moderately well-resolved in three or four ranks. Naturally, the most poorly resolved taxa with only one or two ranks resolved are those with uncertain affinities, such as Shuichengella, Osmundacaulis, or Bathypteris.

Nomenclatural remarks

The electronic version of this article in portable document format (PDF) will represent a published work according to the International Code of Nomenclature (ICN) for algae, fungi, and plants, and hence the new names contained in the electronic version are effectively published under that Code from the electronic edition alone. The online version of this work is archived and available from the following digital repositories: PeerJ, PubMed Central, and CLOCKSS.

Results and Discussion

Numerical phylogenetic framework

The all-inclusive matrix (S00, all Osmundales + outgroup) provides partly incompatible signals accompanied by substantial intertaxonomic diversity. The maximum pairwise distance found in the Osmundales matrix is 0.81; the matrix Delta Value (mDV) is relatively high with 0.38 (File S3), but is within the range expected for morphological matrices including mainly fossil taxa (Guido W. Grimm, 2005 onwards, personal observation; Table S2; see Göker & Grimm, 2008). In general, a tree derived from a matrix with a mDV > 0.2 (Table S2) can be assumed to represent the signal in the underlying dataset only inadequately, occasionally including clades that are incompatible with molecular data (Denk & Grimm, 2005; Denk, Grimm & Hemleben, 2005; Manos et al., 2007; Friis et al., 2009, see also Sareela et al. 2007). In this most comprehensive all-taxa matrix (matrix S00), signals from all operational units are more or less ambiguous; iDV range between 0.31 (Osmundacaulis skidegatensis) and 0.43 (Millerocaulis tuhajkulensis), and the iDVs are largely decoupled from the proportion of missing data (File S3). The resulting neighbour-nets reflect the non-treelikeness of the signal in showing pronounced box-like structures associated with generally low bootstrap (BS) support for all known potential and alternative relationships (File S3). Our matrix has been optimized for taxonomic assessment. Thus, it includes many homoplastic traits and few generally sorted (informative) individual characters, i.e., characters compatible with splits reflecting potential phylogenetic lineages. Bootstrapping of such matrices produces many pseudoreplicate matrices in which the few sorted, phylogenetically informative characters become replaced by uninformative ones that provide only diffuse signals. Nevertheless, the BS supports reflect a consistent signal for certain relationships. Overall, the matrix signal is most decisive when using ML or LS/NJ, whereas the overall support is lowest under parsimony. Apart from this, there are very few apparent conflicts between the different optimization criteria. The diffusion effect from homoplastic signals is mitigated when the analysis is narrowed to a particular taxon subset (File S3); for example, support for a clade including all modern Osmundastrum rhizomes, including Cretaceous and younger fossils of Osmundastrum cinnamomeum, is overall low in the all-inclusive matrix (BSML/NJ/MP = 25/30/10), but increases to BSML/NJ/MP = 56/65/18 for the least-inclusive matrix (Osmundinae) including members of this lineage (File S3).

The high amount of incompatible signals limits the utility of our matrix for inferring explicit trees as phylogenetic scenarios, but it can to some degree be accommodated using neighbour-nets instead. Taxa that belong to the same (putative) phylogenetic lineage (detailed below), defined by notably similar character suites and sharing traits that are not or rarely found outside the (putative) lineage, are typically grouped in the neighbour-nets. The lineage-corresponding edges are usually equivalent to the best-supported alternative(s) found in the BS pseudoreplicates. Furthermore, by adding a new taxon to the matrix (or one of the taxon subsets), its systematic affinities can be readily established (see example provided in File S5).

Revised and annotated classification

There is general agreement that Osmundales is holophyletic (see, e.g., Yatabe, Nishida & Murakami, 1999; Smith et al., 2006; Schuettpelz & Pryer, 2007; PPG I, 2016; Testo & Sundue, 2016). Our subdivision of Osmundales into two main families—Osmundaceae and Guaireaceae—follows most previous studies since Guairea and Guaireaceae were established (Herbst, 1981; see, e.g., Tidwell & Ash, 1994; Tian, Wang & Jiang, 2008; Fig. S1). Those Guaireaceae taxa that possess perforated (dissected) siphonosteles or dictyosteles (Guairea, Lunea, and Zhongmingella) are assigned to subfamily Guaireoideae, whereas those that possess gapless steles that are composed of a spongy mix of xylem and parenchyma are assigned to a new subfamily, Itopsidemoideae subfam. nov. Osmundacaulis is included in Osmundaceae (see Tidwell & Ash, 1994) instead of Guaireaceae (see Wang et al., 2014b) because of the occurrence of a two-layered cortex, the C-shaped stipe bundles with incurved tips, and the presence of stipular wings. Also Shuichengella is assigned to Osmundaceae due to its similarly two-layered cortex. Owing to the peculiar cortex organization with a much thinner sclerenchymatic outer layer and to the otherwise unclear relationships to the remaining taxa in Osmundaceae, we refrain from subfamily assignment of Osmundacaulis and Shuichengella at present. Apart from the poorly known Bathypteris, whose position within Osmundaceae remains uncertain, all other remaining taxa in Osmundaceae are resolved in two subfamilies. Our analyses support Thamnopteroideae as a clearly delimitable subfamily (Miller, 1971; Tidwell & Ash, 1994; see Fig. S1). Furthermore, Zalesskya, Iegosigopteris, and Petcheropteris are so similar to Thamnopteris (see Zalessky, 1935; Miller, 1971) that we consider them synonymous. The remaining Osmundaceae genera, whose stem centre is primarily parenchymatic and whose xylem siphon shows conspicuous leaf gaps, are assigned to subfamily Osmundoideae in accordance with previous studies (see also Miller, 1971; Tidwell & Ash, 1994; Tian, Wang & Jiang, 2008). Extinct genera of Osmundoideae include Palaeosmunda (perhaps including Millerocaulis stipabonettiorum) and Millerocaulis, which is characterized by a plesiomorphic character suite including a homogeneous petiolar sclerenchyma ring without distinct fibre patches. The differentiation of this petiolar sclerenchyma ring into distinct arches and patches of thick-walled fibres is here considered diagnostic for the tribe Osmundeae, which comprises all six Osmundaceae genera with extant representatives (see PPG I, 2016): Todea and Leptopteris in subtribe Todeinae and Claytosmunda, Osmundastrum, Osmunda, and Plenasium in subtribe Osmundinae. One major taxonomic novelty resulting from our analysis is that Aurealcaulis, whose systematic position has hitherto remained enigmatic (Tidwell & Parker, 1987; Tidwell & Medlyn, 1991; Tidwell & Pigg, 1993; Tidwell & Ash, 1994; Tian, Wang & Jiang, 2008; Wang et al., 2014b), is recognized as an extinct subgenus of Plenasium. In addition, several fossil species that were previously included in Millerocaulis or Ashicaulis are recognized as extinct species of subtribe Osmundinae (congruent with the former Osmunda sensu lato), most of which belonging to Claytosmunda and some to Osmundastrum. Osmundacaulis estipularis from the Cretaceous of India (Sharma, Bohra & Singh, 1979) is assigned to subtribe Todeinae, and may represent a previously unrecognized fossil member of Leptopteris.

Detailed discussion and explanations of our classification are given below in the comments section for the particular taxa.

Nomenclatural novelties

Our analysis warrants the following taxonomic changes for members of fossil Osmundales as presently understood: (1) institution of the new subfamily Itopsidemoideae (Guaireaceae) subfam. nov. to contain Itopsidema vancleaveii Daugherty, Donwelliacaulis chlouberii S.R.Ash, and Tiania yunnanense (Bao-Lin Tian & Jiang-Lin Chang ex Shi-Jun Wang, J.Hilton, Galtier et al.) Shi-Jun Wang, J.Hilton, Galtier et al.; (2) institution of two new subtribes within tribus Osmundeae, i.e., Todeinae subtribus nov. and Osmundinae subtribus nov.; (3) synonymy of Zalesskya Kidst. & Gwynne-Vaughan, Petcheropteris Zalessky, and Iegosigopteris Zalessky with Thamnopteris Brongn. with the resulting new combinations: Thamnopteris diploxylon (Kidst. & Gwynne-Vaughan) comb. nov., Thamnopteris gracilis (Eichw.) comb. nov., Thamnopteris javorskii (Zalessky) comb. nov., Thamnopteris splendida (Zalessky) comb. nov., Thamnopteris uralica (Zalessky) comb. nov.; (4) recognition of a new species of Millerocaulis that was previously assigned to Osmundites, i.e., Millerocaulis tuhajkulensis (Gorskii ex Pryn.) comb. nov.; (5) several new combinations in Osmundeae (“modern” Osmundoideae) of species that were previously assigned to Ashicaulis and Millerocaulis, including Claytosmunda beardmorensis (J.M.Schopf) comb. nov., Claytosmunda chengii nom. nov. (replacement name for a new combination based on Ashicaulis claytoniites Y.M.Cheng), Claytosmunda johnstonii (Tidwell, Munzing & M.R.Banks) comb. nov., Claytosmunda liaoningensis (Wu Zhang & Shao-Lin Zheng) comb. nov., Claytosmunda plumites (N.Tian & Y.D.Wang) comb. nov., Claytosmunda preosmunda (Y.M.Cheng, Yu F.Wang & C.S.Li) comb. nov., Claytosmunda sinica (Y.M.Cheng & C.S.Li) comb. nov., Claytosmunda tekelili (E.I.Vera) comb. nov., Claytosmunda wangii (N.Tian & Y.D.Wang) comb. nov., Claytosmunda embreei (Stockey & S.Y.Sm.) comb. nov., Osmundastrum indentatum (R.S.Hill, S.M.Forsyth & F.Green) comb. nov., and Osmunda kidstonii (Stopes) comb. nov.; (6) new combinations in Osmundinae resulting merely from the recent elevation of former subgenera of Osmunda to separate genera (PPG I, 2016), i.e., Osmundastrum pulchellum (Bomfleur, G.W.Grimm, McLoughlin) comb. nov., Plenasium arnoldii (C.N.Mill.) comb. nov., Plenasium chandleri (Arnold) comb. nov., and Plenasium dowkeri (Carruth.) comb. nov.; and (7) recognition of Aurealcaulis as a subgenus of Plenasium with the resulting new species combinations, Plenasium bransonii (Tidwell & Medlyn) comb. nov., Plenasium burgii (Tidwell & J.E.Skog) comb. nov., Plenasium crossii (Tidwell & L.R.Parker) comb. nov., Plenasium dakotense (Tidwell & J.E.Skog) comb. nov., Plenasium moorei (Tidwell & Medlyn) comb. nov., and Plenasium nebraskense (Tidwell & J.E.Skog) comb. nov. Formal taxonomic treatment of these nomenclatural novelties is provided in Appendix A.

Note on orthography of taxon names

Several taxon names contained orthographical or typographical errors that were corrected according to Articles 60 and 62 of the ICN for Algae, Fungi, and Plants (Melbourne Code, 2011). Corrected errors include, e.g., terminations of adjectival epithets not in accordance with the gender of the genus (e.g., Millerocaulis indicus replacing Millerocaulis indica; see Articles 23.5 and 32.2 of the Melbourne Code, 2012); terminations of honorific substantival epithets not in accordance with the sex or number of the person(s) honored, e.g., lutziae replacing lutzii (in honor of Dr Alicia M. Lutz; see Note 4 on Art. 60 of the Melbourne Code, 2012) or stipabonettiorum replacing stipabonettii (in honor or Drs Stipanicic and Bonetti; see Article 60.12 of the Melbourne Code, 2012); or formation of regular compounds not in accordance with classical usage, e.g., bromeliifolium replacing bromeliaefolium (see Recommendation 60G of the Melbourne Code, 2012).

General remarks

In the following treatment, the diagnostic axis characters accumulate with increasing resolution of taxonomic rank, i.e., features considered diagnostic of the family are included again also in the diagnosis for the subfamily, and those are together again repeated in the diagnosis of the genus, and so forth. We follow this practice to account for parallelism owing to potentially homoplastic but, nonetheless, informative morphological features: individual features or feature combinations diagnostic of a taxon at a given rank within a given target group may independently also be diagnostic of another distantly related taxon in another target group. Following Miller (1971), we set those features of a diagnosis in italic font whose combination serves best to differentiate this particular taxon from any other taxon at the same rank in the same target group (rank-specific discriminating characters within the target group). By doing so, potentially homoplastic, but nonetheless informative features (e.g., high degree of stele perforation) can serve as differentiating diagnostic characters within a target group of closely related taxa, even though these features may occur independently also within a more distantly related group.

Order Osmundales Link 1833

(Fig. 9)

Figure 9 Planar network (neighbour-net) for all operational units of order Osmundales except those with >60% characters missing.

Note that the currently accepted family and subfamily concept is well supported with the exception of the Osmundacaulis lineage. The six-letter labels are contractions of the taxon name of an operational unit formed from the first three letters of the genus name in bold followed by the first three letters of the specific epithet; Oum, Osmundastrum. A fully labelled raw version of this graph is provided in Fig. S3.

Diagnostic axis characters: Stems radially symmetrical. Stele with a more or less modified, distinct peripheral metaxylem siphon, with protoxylem initially mesarch or subexarch. Phyllotaxis a tight spiral. Vascular bundle in the stipe base entire, strongly curved adaxially (e.g., C-, omega-, or horseshoe-shaped), with endarch protoxylem.

Status: Holophyletic, extant (relictual) with fossil representatives.

Known geochronologic range: Late Permian to present.

Comments: The holophyly of all Osmundales, extant and extinct, is universally accepted (see, e.g., Yatabe, Nishida & Murakami, 1999; Smith et al., 2006; Schuettpelz & Pryer, 2007). For the family and subfamily subdivision indicated in Fig. 9 see relevant descriptions and remarks below.

1 (†) (†) Family Guaireaceae R.Herbst, 1981

(Fig. 10)

Figure 10 Planar network (neighbor-net) for all operational units of family Guaireaceae.

The six-letter labels are contractions of the taxon name of an operational unit formed from the first three letters of the genus name in bold followed by the first three letters of the specific epithet. A fully labelled raw version of this graph is provided in Supplemental Information.

Diagnostic axis characters: Stem and stipe cortex primarily parenchymatous and not differentiated into distinct layers; where known, stipes lacking stipular wings and stipe bundle with recurved tips (i.e., more or less omega-shaped); roots commonly arising from abaxial side of leaf trace within the stem cortex.

Status: Putatively holophyletic, extinct.

Known geochronologic range: Late Permian to Early Jurassic.

Comments: Our subdivision of Osmundales into two main families—Osmundaceae and Guaireaceae—follows most previous studies since Guairea and Guaireaceae were established (Herbst, 1981; see, e.g., Tidwell & Ash, 1994). The main difference from the most recent proposed classification (Wang et al., 2014b) is that we exclude Osmundacaulis and Shuichengella from Guaireaceae and re-assign them to Osmundaceae. The corresponding clade in that study is most likely a long-branch artefact. The neighbour-net illustrates the morphological disparity within the earliest (Permian) Guaireaceae, but also highlights the relative scarcity of data for members of the family. In the all-inclusive data set, neither family received measurable support (BSML/NJ/MP < 10), nor the alternative proposed by Wang et al. (2014b) of a family Guaireaceae including Osmundacaulis. Nonetheless, the character suites found in the earliest Guiareaceae and Osmundaceae indicate that they represent potential sister lineages within the Osmundales and that neither one evolved from the other. The position of the Guaireaceae root is undetermined. By analogy with Osmundaceae, Itopsidemoideae may be basal within Guaireaceae considering the imperforate steles of all its members.

1.1 (†) Subfamily Guaireoideae Z.M.Li, 1993

Diagnostic axis characters: Stele perforated; stem and stipe cortex primarily parenchymatous and not differentiated into distinct layers; where known, stipes lacking stipular wings and stipe bundle with strongly recurved tips; roots commonly arising from abaxial side of leaf trace within the stem cortex.

Status: Possibly holophyletic, extinct.

Known geochronologic range: Late Permian to Early Jurassic.

Comments: We recognize those Guaireaceae taxa that possess perforated (dissected) siphonosteles or dictyosteles (Guairea, Lunea, and Zhongmingella) as being sufficiently similar to each other and sufficiently distinct from other Guaireaceae (Fig. 9) that we assign them to one subfamily: Guaireoideae. Although (highly) perforated steles also evolved independently in other Osmundales (in particular, Osmundacaulis; as reflected by BSML/NJ ≥ 20 for an artificial Guairea + Osmundacaulis clade using the all-inclusive matrix), they are typically restricted to distinct sublineages with a probable inclusive common origin. The split between Guaireoideae and Itopsidemoideae is well supported (BSML/NJ/MP ≥ 70) if the taxon set is restricted to only Guaireacae (Fig. 10).

1.1.1 (†) (†) Genus Guairea R.Herbst, 1981

Diagnostic axis characters: Stems forming large arborescent trunks (reaching up to about 10 cm in diameter); stelar xylem siphon very thick (up to >50 tracheids in radial thickness), moderately perforated; endodermis external and internal and connecting through leaf gaps; pith and stem cortex parenchymatous only and not differentiated into distinct layers; ca 20–35 leaf traces in a given stem transverse section; roots commonly arising from abaxial side of leaf trace within the stem cortex.

Status: Holophyletic or paraphyletic, extinct.

Known geochronologic range: Late Permian to Middle Triassic.

Comments: Guairea species are very similar to each other and sufficiently distinct from other Permian Osmundales to be recognized as a genus. The close similarity of the individually scored Osmundites braziliensis, which is the reason for the high BS of the according split under ML and LS/NJ (BSML/NJ ≥ 75), supports its synonymy with Guairea carnieri (see Herbst, 1981). A Guairea-clade would receive (moderately) high support (BSML/NJ/MP ≥ 67). However, the lack of additional Triassic Guaireaceae fossils makes it difficult to assess the relationship of Guairea and Zhongmingella to the Jurassic Lunea (they may be precursors, i.e., forming a paraphyletic group, or ancient holophyletic sister lineages with no ancestor–descendant relationship).

Included species: (†) G. carnierii (J.Schust) R.Herbst, 1981 including Osmundites braziliensis of H.N.Andrews, 1950 (Late Permian: Paraguay, Rio Grande do Sul, Brazil; Middle Triassic: Rio Grande do Sul, Brazil).

(†) G. milleri R.Herbst, 1981 (Late Permian: Paraguay).

References: Andrews (1950), Ash (1994), Bower (1926), Herbst (1981), Herbst, Barboni & Dutra (2012), Kidston & Gwynne-Vaughan (1914), Miller (1971), Schuster (1911), Tidwell (1991) and Tidwell & Ash (1994).

1.1.2 (†) Genus Lunea Tidwell, 1991

Diagnostic characters: Stems rhizomatous to erect; stelar xylem siphon thin (less than ca 20 tracheids in radial thickness), moderately perforated; endodermis external only; pith and cortices of stem and stipes parenchymatous with abundant scattered masses of sclerenchyma fibres; ca 25–35 leaf traces in a given stem transverse section; stipes lacking stipular wings and stipe bundle with strongly recurved tips; roots commonly arising from abaxial side of leaf trace within the stem cortex.

Status: Monotypic, extinct.

Known geochronologic range: Early Jurassic.

Comments: The Guaireaceae-matrix signal regarding the placement of Lunea is ambiguous (Fig. 10). The higher support for a Lunea + Zhongmingella clade compared to the alternative of a Lunea + Guairea clade may be artificial: Lunea is relatively distinct from the (much older) Guairea spp., Zhongmingella even more so, and all are substantially distinct from the members of Itopsidemoideae. There are no traits shared by Lunea and Zhongmingella to the exclusion of Guairea that would support a sister relationship of the former two.

Included species: (†) L. jonesii Tidwell, 1991 (Early Jurassic: Tasmania, Australia).

References: Tidwell (1991), Ash (1994) and Tidwell & Ash (1994).

1.1.3 (†) Genus Zhongmingella S.Jun Wang, J.Hilton, Xiao Y.He et al., 2014

Diagnostic characters: Stems rhizomatous; stelar xylem siphon moderately thick (reaching up to ca 30 tracheids in radial thickness), moderately perforated, dictyostelic with internal and external phloem connecting through leaf gaps; pith and stem cortex parenchymatous with abundant scattered fibre masses; stem cortex not differentiated into distinct layers; more than 40 leaf traces in a given stem transverse section; roots commonly arising from abaxial side of leaf trace within the stem cortex.

Status: Monotypic, extinct.

Known geochronologic range: Late Permian (Changhsingian).

Comments: The pith and cortex of Zhongmingella contain structural elements that have been described variously as groups or clusters of secretory cells or of sclerenchyma (Li, 1983; Wang et al., 2014b). We follow the coding in the matrix of Wang et al. (2014b) and consider these elements to be sclerenchyma masses. Perhaps owing to its peculiar cortex construction with a seemingly distinct innermost zone, Zhongmingella is unlike other members of Osmundales according to phylogenetic reconstructions (Fig. 10).

Included species: (†) Z. plenasioides (Z.M.Li) Shi-Jun Wang, J.Hilton, Xiao-Yuan He et al. 2014 (Late Permian: Guizhou, China).

References: Li (1983) and Wang et al. (2014b).

1.2 (†) Subfamily Itopsidemoideae Bomfleur, G.W.Grimm & McLoughlin, subfam. nov.

Diagnostic characters: Stelar xylem siphon lacking discrete leaf gaps, composed of a spongy admixture of metaxylem and more or less diffusely interspersed patches of parenchyma; stem and stipe cortex primarily parenchymatous and not differentiated into distinct layers; stipes lacking stipular wings and stipe bundle with recurved tips (i.e., more or less omega-shaped); roots commonly arising from abaxial side of leaf trace within the stem cortex.

Status: Holophyletic or paraphyletic, extinct.

Known geochronologic range: Late Permian to Middle Triassic.

Comments: We consider the unique stele composition and structure of three Guaireaceae taxa—Itopsidema, Donwelliacaulis, and Tiania—to be so substantially different from those of the Guaireoideae to warrant the erection of this new subfamily. Itopsidemoideae might be a holophyletic sister lineage of the Guaireoideae—both lineages having appeared about the same time, and the unusual spongy metaxylem tissue representing a unique trait within the Osmundales—or include members from which the Guaireoideae with their much more complex steles have evolved, analogous to the presumed evolution of Osmundoideae from Thamnopteroideae in Osmundaceae. The less complex steles in combination with traits not found in Osmundaceae, account for the relatively high support for Itopsidemoideae vs. all other splits for the all-inclusive (BSML/NJ/MP = 42/47/32) and Guaireaceae (Fig. 10) matrices.

1.2.1 (†) Genus Itopsidema Daugherty, 1960

Diagnostic characters: Pith parenchymatous with interspersed tracheids; stelar xylem siphon thin (reaching ca 20 tracheids in radial thickness), lacking discrete leaf gaps, composed of a spongy admixture of metaxylem and more or less diffusely interspersed patches of parenchyma; stem and stipe cortex primarily parenchymatous and not differentiated into distinct layers; numerous (reaching >100) leaf traces visible in a given stem transverse section; stipe bundle with recurved tips (i.e., inverse omega-shaped or mushroom-shaped); stipe cortex with masses of secretory cells; surface of stem and stipes covered in multicellular spines with interspersed trichomes; roots commonly arising from abaxial side of leaf trace within the stem cortex.

Status: Monotypic, extinct.

Known geochronologic range: Middle Triassic.

Comments: Unfortunately, the diagnosis, description, and documentation of Itopsidema and its type species I. vancleavei (Daugherty, 1960) followed an unconventional format that makes it difficult to compare them with other, probably closely related taxa. Important information missing from the protolog concerns, for instance, the origination and development of leaf trace protoxylem from the stele through the trace and into the stipe. This lack of information makes it impossible at present to determine how, for instance, T. yunnanense (see below) differs structurally from I. vancleavei; the separation of these taxa in the neighbour-net (Fig. 9) is partly a consequence of a poorly defined pairwise distance owing to missing data. Poor data overlap may be one reason that BS analysis provides support for Itopsidema + Tiania (BSML/NJ/MP = 38/36/15) as alternative to Itopsidema + Donwelliacaulis (BSML/NJ/MP = 59/58/63; see below).

Included species: (†) I. vancleavei Daugherty, 1960 (Middle Triassic: Arizona, USA).

References: Daugherty (1960), Hewitson (1962), Miller (1971), Ash (1994) and Tidwell & Ash (1994); see also Wang et al. (2014a).

1.2.2 (†) Genus Donwelliacaulis Ash, 1994

Diagnostic characters: Stems exceeding 25 cm in diameter and forming very large arborescent trunks reaching more than 40 cm in diameter; pith parenchymatous with interspersed tracheid bundles; stelar xylem siphon very thick (reaching >70 tracheids in radial thickness), lacking discrete leaf gaps, composed of a spongy admixture of metaxylem and more or less diffusely interspersed patches of parenchyma; stem cortex primarily parenchymatous with interspersed sclerenchyma masses, not differentiated into distinct layers, containing rather few (ca 20) widely separated leaf traces in a given stem transverse section; stipe bundle inverse omega-shaped with recurved tips.

Status: Monotypic, extinct.

Known geochronologic range: Middle Triassic.

Comments: Hundreds of Donwelliacaulis trunk fragments have been collected from the Holbrook Member of the Moenkopi Formation of east-central Arizona (Ash, 1994), whence also the type and only specimen of Itopsidema derives (Daugherty, 1960). Owing to the small sample size and the incomplete knowledge of important diagnostic characters and about the range of variation in I. vancleavii, it remains unclear in which characters Donwelliacaulis differs structurally from Itopsidema apart from those that might be related to ontogeny and development (e.g., much larger size or fewer leaf traces). Their close relationship is reflected by the BS values for the according split (Fig. 10). We consider the two genera to be at least closely related, and suggest that they might represent different growth stages of the same type of natural plant.

Cells that were originally interpreted as protoxylem are “[…] about 24–36 μm in diameter and with walls about 3–5 μm thick” (Ash, 1994: 6). These dimensions are much too large for true protoxylem. Instead, we interpret those cells as small, early-formed metaxylem tracheids as they occur in the stele periphery of all Osmundaceae (see Ash, 1994: pl. 5, fig. 2; pl. 6, fig. 7). Hence, protoxylem maturation in Donwelliacaulis should no longer be considered exarch as originally proposed (see also remarks for Aurealcaulis).

Included species: (†) D. chlouberii Ash, 1994 (Middle Triassic: Arizona, USA).

References: Ash (1994) and Tidwell & Ash (1994).

1.2.2 (†) Genus Tiania Shi-Jun Wang, J.Hilton, Galtier et al., 2014

Diagnostic characters: Pith primarily parenchymatous; stelar xylem siphon thin (up to ca 10 tracheids in radial thickness), lacking discrete leaf gaps, composed of a spongy admixture of metaxylem and more or less diffusely interspersed patches of parenchyma; stem cortex primarily parenchymatous and not differentiated into distinct layers, containing scattered masses of secretory cells and sclerenchyma; numerous (reaching >100) leaf traces visible in a given stem transverse section; leaf trace protoxylem single and mesarch in stele, dividing into two and becoming endarch before departure from stele; leaf trace with more than four endarch protoxylem strands upon departure from stem; roots commonly arising from abaxial side of leaf trace within the stem cortex.

Status: Problematic (possibly synonymous with Itopsidema).

Known geochronologic range: Late Permian.

Comments: Tiania appears to differ from Itopsidema mainly in preservational aspects and in the accompanying protolog: the former has more fully documented features of the leaf traces but lacks preserved stipes (Wang et al., 2014a), whereas the type material of the latter has a few attached stipe bases but lacks documented features of leaf trace emission (see Daugherty, 1960). Tiania might be a junior synonym of Itopsidema, but formal taxonomic treatment should be based on a re-examination of the type material of Itopsidema and acquisition of more completely preserved axes.

Included species: (†) T. yunnanense (B.L.Tian & J.L.Chang ex Shi-Jun Wang, J.Hilton, Galtier et al.) Shi-Jun Wang, J.Hilton, Galtier et al., 2014a (Late Permian: Yunnan, China).

References: Li & Cui (1995) and Wang et al. (2014a).

2 Family Osmundaceae Martinov 1820

(Fig. 11)

Figure 11 Planar network (neighbour-net) for all operational units of family Osmundaceae.

Note the two well-resolved ‘branches’ for subfamily Thamnopteroideae and for Osmundacaulis; attachment of the isolated Shuichengella near the base of the Osmundacaulis ‘branch’; the box comprising Palaeosmunda plus Millerocaulis stipabonettiorum between Thamnopteroideae and the remaining Osmundoideae; the clustering of modern genera and of subgenera with extant representatives; and the poorly resolved spider-web structure of the remaining Mesozoic fossil Osmundinae and Millerocaulis species. The six-letter labels are contractions of the taxon name of an operational unit formed from the first three letters of the genus name in bold followed by the first three letters of the specific epithet; Oum, Osmundastrum. A fully labelled raw version of this graph is provided in Fig. S4.

Diagnostic axis characters: Cortex of stems and stipes two-layered, differentiated into inner, primarily parenchymatous cylinder and outer sclerenchymatous cylinder. Stipes with a pair of stipular wings; stipe bundle with incurved tips (i.e., more or less horseshoe-shaped). Peripheral xylem siphon typically with leaf gaps (except some Thamnopteroideae).

Status: Natural, possibly paraphyletic with respect to Guaireaceae, extant with fossil representatives.

Known geochronologic range: Late Permian to present.

Comments: Unlike Wang et al. (2014b), we include Osmundacaulis and Shuichengella in Osmundaceae instead of Guaireaceae. We argue that Osmundacaulis may appear affiliated with Guaireaceae, and with Guairea in particular (Fig. 9), because they share several features that are highly homoplastic and are known to vary between closely related taxa or even between individuals of the same species. Such characters include the degree of stele perforation (see Vera, 2008); the occurrence of simple or dissected-siphonostelic or dictyostelic conditions (see Faull, 1901; Hewitson, 1962; Miller, 1971; Serbet & Rothwell, 1999); and the occurrence of tracheids or medullary bundles in the pith (see Gwynne-Vaughan, 1914; Hewitson, 1962). Therefore, we consider the placement of Osmundacaulis closer to Guaireaceae in the trees of Wang et al. (2014b) and in the comprehensive neighbour-nets presented here (Fig. 9) to be artificial and a result of a combination of long-branch attraction and overestimated signals from homoplastic features.

By contrast, Osmundacaulis shares with other Osmundaceae (1) the two-layered cortex of stems and stipes with an inner, primarily parenchymatous cylinder and an outer sclerenchymatous cylinder (as opposed to a homogeneous, un-layered cortex of stems and stipes in Guaireaceae); (2) a C-shaped or horseshoe-shaped vascular bundle with more or less incurved tips in the stipe base (as opposed to omega-shaped or mushroom-shaped bundles with recurved tips); (3) the presence of stipular wings; and (4) the formation of a prominent mantle of many persistent leaf bases and roots (as opposed to a mantle of mainly roots with no or few persistent stipe bases). These features are invariant characteristics distinguishing Osmundaceae (including Osmundacaulis) from Guaireaceae, and represent conserved traits in all extant Osmundaceae. We consider these latter features to have much greater phylogenetic and systematic significance than those used previously, and re-assign Osmundacaulis to Osmundacaeae. Shuichengella is also assigned to Osmundaceae based on the observation that it has a similarly two-layered cortex. The fact that it is not resolved as a close relative in the Osmundales neighbour-net (Fig. 8) but nested instead within Guaireaceae is probably an analytical artefact related to the dearth of preserved diagnostic characters. No split placing Shuichengella received BSML/NJ/MP ≥ 10. When the data matrix is reduced to only include Osmundaceae, Shuichengella groups with Osmundacaulis, and the corresponding split represents the best-supported alternative (BSML/NJ/MP = 42/32/22; all other alternatives BSML/NJ/MP < 10). Hence, if Shuichengella is accurately interpreted as belonging to Osmundaceae, it probably is part of the same lineage that gave rise to Osmundacaulis. If future studies should produce further support for this hypothesis, it may become appropriate to re-institute the subfamily Shuichengelloideae Z.M.Li, 1993 for these two genera.

2.1 (†) Subfamily Thamnopteroideae C.N.Mill., 1971

(Fig. 12)

Figure 12 Planar network (neighbour-net) for all operational units of subfamily Thamnopteroideae (Osmundaceae).

Note the clustering especially of species of the formerly separated Thamnopteris and Zalesskya in the centre of the network and the divergence of Chasmatopteris principalis. The six-letter labels are contractions of the taxon name of an operational unit formed from the first three letters of the genus name in bold followed by the first three letters of the specific epithet.

Diagnostic characters: Stems typically forming large, arborescent trunks. Stem core consisting primarily of tracheids. Peripheral metaxylem siphon typically entire and imperforate [most taxa] or with few leaf gaps [Chasmatopteris, some Thamnopteris spp.] or sparse perforations [Chasmatopteris]. Leaf trace protoxylem initiating subexarch in peripheral bulge, becoming endarch typically in cortex. Cortex of stems and stipes two-layered, differentiated into inner, primarily parenchymatous cylinder and outer sclerenchymatous cylinder. Stipes with a pair of stipular wings; stipe bundle with incurved tips (i.e., more or less horseshoe-shaped).

Status: Holophyletic or paraphyletic, extinct.

Known geochronologic range: Late Permian.

Comments: Species included in this subfamily are rather similar to each other; the group-limited matrix (including B. rhomboidea, see below) comprises only 20 partly defined and variable characters (sites). Species included in Thamnopteris (nine variable sites) are generally similar in their preserved traits, hence 10 out of 55 pairwise distances are zero, and little phylogenetic structuring is evident (Fig. 12; File S3) within the subfamily.

2.1.1 (†) Genus Thamnopteris Brongn., 1849

Synonyms here assigned:

Iegosigopteris Zalessky, 1935.

Petcheropteris Zalessky, 1931.

Zalesskya Kidst. & Gwynne-Vaughan, 1909.

Diagnostic characters: Stems typically forming large, arborescent trunks. Stem core consisting primarily of tracheids. Peripheral metaxylem siphon typically entire and imperforate. Leaf trace protoxylem initiating subexarch in peripheral bulge, becoming endarch typically in cortex. Cortex of stems and stipes two-layered, differentiated into inner, primarily parenchymatous cylinder and outer sclerenchymatous cylinder. Stipes with a pair of stipular wings; stipe bundle with incurved tips (i.e., more or less horseshoe-shaped).

Status: Putatively monophyletic, extinct.

Known geochronologic range: Late Permian.

Comments: Many species and several genera of large, “protostelic” osmundaceous trunks have been described from Upper Permian strata of the Ural Mountains (Eichwald, 1842, 1860; Kidston & Gwynne-Vaughan, 1908; Zalessky, 1924, 1931a, 1931b, 1935). Zalessky (1935) himself and later Miller (1971) remarked that the delimitation of several of these genera is problematic and probably artificial. The main feature distinguishing Petcheropteris from Thamnopteris, i.e., the more sinuous outlines of the stipe sections in the mantle, is “[…] due to compression prior to preservation” (Miller, 1971: 141). Iegosigopteris combines features of Thamnopteris (e.g., the more rhombic shapes of sclerenchyma rings in TS) and of Zalesskya (e.g., longer tracheids in the stem centre) (Zalessky, 1935; see Miller, 1971). Furthermore, the different tracheid length might result from a misinterpretation, since the original descriptions and illustrations (Zalessky, 1935: p. 2: fig. 1, pl. 3, fig. 2) show the stem core tracheids to be rather short and indeed more like those of Thamnopteris. Finally, we note that the main features distinguishing Zalesskya from Thamnopteris are a greater number of leaf traces in the cortex, a thicker inner cortex, incompletely preserved outer cortex, and missing information on mantle and petiole bases. Therefore, we see no reason to consider Zalesskya anything but a particularly large and incomplete Thamnopteris trunk. Consequently, the three genera listed above are treated here as junior synonyms of Thamnopteris.

Overall, the most distinct species of the genus (pairwise distances of 0.21–0.42 based on the Thamnopteroideae-matrix) are Thamnopteris javorskii (13 out of 20 characters unambiguously defined) and Thamnopteris splendida (15/20). Interspecies relationships are largely unclear; a limiting factor is the lack of overlap in the unambiguously defined characters. For instance, the 0.42 distance between Thamnopteris javorskii and Thamnopteris kidstonii (16/20 characters unambiguously defined) translates into six different out of 14 unambiguously defined characters for both species.

Included species: (†) T. diploxylon (Kidst. & Gwynne-Vaughan, 1908) comb. nov. (Late Permian: Russia).

(†) T. gracilis (Eichw., 1860) comb. nov. (Late Permian: Russia).

(†) T. gwynnevaughanii Zalessky, 1924 (Late Permian: Russia).

(†) T. javorskii (Zalessky, 1935) comb. nov. (Late Permian: Russia).

(†) T. kazanensis Zalessky, 1927 (Late Permian: Russia).

(†) T. kidstonii Zalessky, 1924 (Late Permian: Russia).

(†) T. schlechtendalii (Eichw.) Brongn., 1849 (Late Permian: Russia).

(†) T. splendida (Zalessky, 1931) comb. nov. (Late Permian: Russia).

(†) T. uralica (Zalessky, 1924) comb. nov. (Late Permian: Russia).

References: Bower (1926), Brongniart (1849), Eichwald (1842, 1860), Gould (1970), Kidston & Gwynne-Vaughan (1908, 1909), Miller (1971) and Zalessky (1924, 1927, 1931b, 1935).

2.1.2 (†) Genus Chasmatopteris Zalessky, 1931

Diagnostic characters: Stems typically forming large, arborescent trunks. Stem core consisting primarily of tracheids. Peripheral metaxylem siphon with leaf gaps and sparse perforations. Leaf trace protoxylem initiating subexarch in peripheral bulge, becoming endarch typically in cortex. Cortex of stems and stipes two-layered, differentiated into inner, primarily parenchymatous cylinder and outer sclerenchymatous cylinder. Stipes with a pair of stipular wings; stipe bundle with incurved tips (i.e., more or less horseshoe-shaped).

Status: Monotypic, extinct.

Known geochronologic range: Late Permian.

Comments: In light of our critical reappraisal of the significance of the degree of stele perforation for systematic classification (see also Vera, 2008), one can argue whether the presence of complete perforations in the xylem siphon merits the separate generic status of C. principalis. This is especially so given that we propose a rather broad definition of its closest relative Thamnopteris (see above), into which C. principalis would otherwise be merged. However, given that the entire subfamily Thamnopteroideae is otherwise composed of species with consistently imperforate xylem siphons, we consider the combination of having a stem core of more or less entirely tracheids (“protostelic”) but a truly perforated (“dictyoxylic”) stelar xylem siphon to be so unique as to warrant the separate generic status of C. principalis within Thamnopteroideae. Otherwise, the special significance of C. principalis as the single taxon having a stele construction somewhat intermediate between that of the remaining Thamnopteroideae and those of Osmundoideae might not be adequately emphasized in its systematic classification.

Chasmatopteris is the most distinct (derived) known Thamnopteroideae (Fig. 12; MD = 0.38–0.54 based on the Thamnopteroideae-matrix, see File S3; the relatively low distance to T. kazanensis, MD = 0.17, is unrepresentative and an artefact of missing data). Should Chasmatopteris represent an extinct genus in its own right, it probably evolved from a Thamnopteris-like ancestor, rendering the latter genus paraphyletic.

Included species: (†) C. principalis Zalessky, 1931[a] (Late Permian: Russia).

References: Zalessky (1931a), Gould (1970) and Miller (1971).

2.2 Subfamily Osmundoideae R.Br. ex Sweet, 1826

(Fig. 13)

Figure 13 Planar network (neighbour-net) for all operational units of subfamily Osmundoideae (Osmundaceae).

Note the divergence of the ‘branch’ comprising the closely similar subgenera Plenasium and Aurealcaulis and the separation of the ‘branch’ containing Todea and Leptopteris, including “Millerocaulis estipularis.” The six-letter labels are contractions of the taxon name of an operational unit formed from the first three letters of the genus name in bold followed by the first three letters of the specific epithet; Oum, Osmundastrum. A fully labelled raw version of this graph is provided in Fig. S5.

Diagnostic characters: Stem core a primarily parenchymatous pith. Stelar xylem siphon usually thin (up to ca 20 tracheids in radial thickness) and with prominent leaf gaps. Leaf protoxylem poles initiating in mesarch position in stele, becoming endarch in stele or (rarely) in cortex. Cortex of stems and stipes two-layered, differentiated into inner, primarily parenchymatous cylinder and outer sclerenchymatous cylinder. Inner cortex usually thinner than outer cortex, rarely about equally thick. Stipes with a pair of stipular wings; stipe bundle with incurved tips (i.e., more or less horseshoe-shaped).

Status: Holophyletic, extant with fossil representatives.

Known geochronologic range: Permian to present.

Comments: The signal of characters instrumental for the recognition of subfamily Osmundoideae is completely outcompeted by the signal from homoplastic characters and differentiation within its sublineages in the case of the all-inclusive Osmundales matrix and the Osmundaceae matrix (BSML/NJ/P < 10). Further taxon-reduction (81 taxa vs. 124 in the all-inclusive matrix), however, has the effect that the most distinct genera within the Osmundoideae (Palaeosmunda, Plenasium) not only receive higher support from BS (File S3) but also become apparent in the neighbour-net graphs. Notably, the Osmundoideae matrix includes ca 25% fewer characters than the all-inclusive matrix (33 vs. 45), but also has a lesser proportion of missing and ambiguous data cells (16% vs. 20%).

2.2.1 (†) Genus Palaeosmunda R.E.Gould, 1970

Diagnostic characters: Stems erect, forming arborescent trunks. Stem core a primarily parenchymatous pith. Stelar xylem siphon thin with variably developed leaf gaps. Leaf trace protoxylem poles initially single and in mesarch position in stele, becoming endarch and, in some cases, first bifurcating before departure from stele; leaf trace protoxylem further dividing repeatedly within stem cortex; leaf trace departing from stem with at least four (commonly more than 10) protoxylem strands. Cortex of stems and stipes two-layered, differentiated into inner, primarily parenchymatous cylinder and outer sclerenchymatous cylinder, containing few to many (reaching ca 45) leaf traces in a given stem transverse section; inner cortex about as thick as outer cortex. Stipes with a pair of stipular wings; stipe bundle with incurved tips (i.e., more or less horseshoe-shaped); stipe sclerenchyma cylinder somewhat rhombic to fusiform in cross-section, with lateral margins usually thinner than abaxial and adaxial portions, distally becoming extended laterally into flanges that partially or completely replace the stipules.

Status: Probably holophyletic, extinct.

Known geochronologic range: Late Permian, possibly extending to the Late Triassic.

Comments: The position of Palaeosmunda with respect to the other Osmundoideae genera and in particular the (putatively) paraphyletic Millerocaulis is not entirely clear (Fig. 13). Independent of the optimality criterion used, BS fails to recover any preferred placement of Palaeosmunda within alternative tree topologies (BSML/NJ/MP of all possible alternatives < 10). Nevertheless, the genus seems to represent an early-diverged, distinct sister lineage when analyzed in a larger taxonomic context (Figs. 9 and 11). Both their morphology (see Diagnostic characters; Fig. 6) and stratigraphic occurrence (Upper Permian to possibly Upper Triassic) can be taken as arguments against the alternative hypotheses that this genus represents a lineage that evolved from a Millerocaulis-type ancestor, or that it could represent the ancestral stock from which the remaining Osmundoideae evolved. Millerocaulis stipabonettiorum has several features that are unusual among Millerocaulis species and more reminiscent of Palaeosmunda (Herbst, 1995); the most conspicuous similarities are the rhombic to fusiform shape of the stipe sclerenchyma rings and the absence of sclerenchyma patches or fibers associated with either leaf traces or stipe vascular bundles (Herbst, 1995). Unfortunately, protoxylem—whose distribution and development should enable unambiguous assignment to either Millerocaulis or Palaeosmunda—is not preserved in the single available specimen. Nevertheless, since (1) we refrain from using Millerocaulis as a “waste-basket taxon” for left-over species that cannot be positively assigned to any of the other taxa of Osmundoideae; (2) all our subset analyses consistently place Millerocaulis stipabonettiorum closer to Palaeosmunda than to any other representative of Millerocaulis (the only splits involving Millerocaulis stipabonettiorum receiving BSML/NJ > 10 are those with Palaeosmunda or one of its species); and (3) generic assignment of this species should thus be considered provisional anyway, we tentatively group Millerocaulis stipabonettiorum with Palaeosmunda and exclude it from our analyses of Millerocaulis (Fig. 13).

Included species: (†) P. playfordii R.E.Gould, 1970 (Late Permian: Queensland, Australia).

(†) P. williamsii R.E.Gould, 1970 (Late Permian: Queensland, Australia).

Tentatively included:

(†) “Millerocaulis” (?Palaeosmunda) stipabonettiorum (Late Triassic: Argentina).

References: Gould (1970), Herbst (1995) and McLoughlin (1992); see also Li (1983).

2.2.2 (†) Genus Millerocaulis Erasmus ex Tidwell emend. E.I.Vera, 2008

(Fig. 14)

Figure 14 Planar network (neighbour-net) for all operational units of genus Millerocaulis (Osmundaceae, Osmundoideae).

Note the poorly resolved spider-web structure of the net and the conspicuous close placements of several taxa from similar-aged deposits in similar geographic realms (see text for details). The six-letter labels are contractions of the taxon name of an operational unit formed from the first three letters of the genus name in bold followed by the first three letters of the specific epithet.

Diagnostic characters: Stems usually rhizomatous to small (semi-)erect. Stem core a primarily parenchymatous pith. Stelar xylem siphon usually thin (up to ca 20 tracheids in radial thickness) and with prominent leaf gaps. Protoxylem poles in stele initially single and mesarch, becoming endarch in stele or (rarely) in cortex. Cortex of stems and stipes two-layered, differentiated into thin inner, primarily parenchymatous cylinder and thick outer sclerenchymatous cylinder. Stipes with a pair of stipular wings; stipe bundle with incurved tips (i.e., more or less horseshoe-shaped). Sclerenchyma ring in stipe base circular to elliptic in cross-section, homogenous to gradually or diffusely heterogeneous.

Status: Probably paraphyletic with respect to Osmundeae.

Known geochronologic range: Triassic to mid-Cretaceous.

Comments: Vera (2008) proposed merging Millerocaulis Erasmus ex Tidwell emend. Tidwell (1994; with imperforate steles) and Ashicaulis Tidwell (1994; with perforate steles) into a single genus, the broadly defined Millerocaulis Tidwell emend. E.I.Vera, which we follow here.

Members of Millerocaulis, explicitly circumscribed here as a paraphyletic taxon, can be either precursors or belong to sister lineages of the “modern” Osmundaceae (classified as Osmundeae). In the light of the notable size and disparity of the genus, it is tempting to subdivide it further. With the available information, however, a consistent further subdivision is difficult to achieve. The neighbour-net focussing on Millerocaulis is essentially a “spider-web” (Fig. 14), and the same holds true for support consensus networks (Grimm, Bomfleur & McLoughlin (2017) available from Dryad Digital Repository http://dx.doi.org/10.5061/dryad.270gs), illustrating the lack of consistent sorting signals in the underlying matrix. The traditional subdivision based on the degree of stele perforation (informal groups as in the listing below) is, therefore, not supported by any further evidence. Furthermore, the data compiled here show that imperforate and highly perforate steles are end members of a complete transformation series in Osmundoideae and, thus, provide a poor basis for classification above species level. A much deeper understanding of character conservation and evolution in the Osmundoideae would be necessary to propose a finer subdivision. Moreover, it is conspicuous that the network (Fig. 14) shows several boxes that are composed of species of similar age and similar geographic occurrence that plot closely together (e.g., Millerocaulis limewoodensis with Millerocaulis dunlopii and Millerocaulis aucklandicus all from the Middle Jurassic of eastern Australasia); we suspect that more complete knowledge of the natural variability of anatomical features of the constituent species might prove some of those to represent variants of the same natural species, as has been suggested for Millerocaulis dunlopii and Millerocaulis aucklandicus (Miller, 1971; Tidwell, 1986; zero pairwise distance here). Further potential sister species pairs including candidates for further taxonomic revision are listed in Table 1. The association of the Triassic Millerocaulis herbstii with the Cretaceous Millerocaulis kolbei is probably an artefact: both taxa differ from all other Millerocaulis as much as from each other.

Table 1 Potential sister species within Millerocaulis as inferred via non-parametric bootstrapping of the Millerocaulis-matrix (matrix dimensions 33 taxa, 23 variable characters).

Pair	Provenance	Pairwise distance	BS support	
Pair	AllOther	ML	LS	MP	
M. amarjolensis + M. beipiaoensis	IND (K) + CHI (J)	0.06	0.14–0.61	31	42	22	
M. broganii + M. woolfei	AUS (T) + ANT (T)	0	0.17–0.50	94	87	68	
M. dunlopii + M. aucklandicus	NZ (J) + NZ (J)	0	0.11–0.47	42	42	12	
M. indicus + M. donponii	IND (K) + AUS (J)	0.06	0.17–0.67	57	59	27	
M. hebeiensis + M. rajmahalensis	CHI (J) + IND (K)	0.15	0.13–0.60	30	37	28	
M. herbstii + M. kolbii	ARG (J) + SAF (K)	0.23	0.25–0.64	31	49	20	
M. spinksii + M. websteri	AUS (?T) + AUS (?T)	0.05	0.10–0.58	30	37	<10	
Notes:

Only pairs are shown, where the corresponding split received BS support >33 under at least one optimality criterion (i.e., approximately one-third or more of the scored characters support such a split).

Co-occurrence in bold.

AllOther, range with respect to any other taxon in the matrix (“Millerocaulis” stipabonettiorum, see above, not considered); IND, India; K, Cretaceous; CHI, China; J, Jurassic; AUS, Australia; T, Triassic; ANT, Antarctica; NZ, New Zealand; ARG, Argentina; SAF, South Africa.

Although Millerocaulis is defined mainly on the absence of traits diagnostic of other genera of Osmundoideae, it should not be considered a “waste-basket” or “form-taxon” for Osmundoideae of uncertain affinity. We suggest that if in a new fossil osmundoid rhizome critical characters, such as stelar xylem siphon or the stipe sclerenchyma ring, are not adequately preserved, or if the fossil in question is insufficiently comparable with other fossil osmundoid taxa, it should not automatically be described as a new Millerocaulis species. Instead, we suggest that such specimens should be described in open nomenclature and labeled with the lowest taxonomic rank to which they can be confidently assigned (e.g., “undetermined osmundoid rhizome” or “Osmundoideae gen. and sp. indet.”). The name Millerocaulis should be reserved for Osmundoideae rhizomes with a distinctly plesiomorphic character suite compared with the other genera of the subfamily and, in particular, Osmundeae.

Included species: “Millerocaulis s.str.” group (low degree of stele perforation; see Char. 8): (†) M. chubutensis (R.Herbst) Tidwell, 1994 (Late Jurassic: Argentina).

(†) M. donponii Tidwell & Clifford, 1995 (Middle Jurassic: Australia).

(†) M. dunlopii (Kidst. & Gwynne-Vaughan) Tidwell, 1994, including Osmundites aucklandicus P.Marshall, 1926 (Middle Jurassic: New Zealand).

(†) M. indicus (B.D.Sharma) Tidwell, 1994 (Early Cretaceous: India).

(†) M. juandahensis Tidwell & Clifford, 1995 (Middle Jurassic: Australia).

(†) M. limewoodensis Tidwell & Clifford, 1995 (Middle Jurassic: Australia).

“Ashicaulis” group (moderate degree of stele perforation; see Char. 8): (†) M. amarjolensis (B.D.Sharma) Tidwell, 1986 (Early Cretaceous: India).

(†) M. australis (E.I.Vera) E.I.Vera, 2008 (Early Cretaceous: West Antarctica).

(†) M. beipiaoensis (N.Tian, Y.D.Wang, Wu Zhang et al., 2013) comb. nov. (Middle Jurassic: Liaoning, China).

(†) M. broganii Tidwell, Munzing & M.R.Banks, 1991 (?Triassic: Tasmania, Australia).

(†) M. gibbianus (Kidst. & Gwynne-Vaughan) Tidwell, 1986 (Middle Jurassic: New Zealand).

(†) M. guptai (B.D.Sharma) Tidwell, 1986 (Early Cretaceous: India).

(†) M. hebeiensis (Ziquiang Wang) Tidwell, 1986 (Middle Jurassic: Hebei, China).

(†) M. herbstii (S.Archang. & de la Sota) Tidwell, 1986 (Late Triassic: Argentina).

(†) M. livingstonensis (Cantrill) E.I.Vera, 2008 (Late Cretaceous: West Antarctica).

(†) M. lutziae (R.Herbst) Herbst, 2006 (Late Triassic: Argentina).

(†) M. macromedullosus (M.Matsumoto, K.Saiki, Wu Zhang et al.) E.I.Vera, 2008 (Middle Jurassic: Hebei, China).

(†) M. patagonicus (S.Archang. & de la Sota) Tidwell, 1986 (Middle–Late Jurassic: Argentina).

(†) M. rajmahalensis (K.M.Gupta) Tidwell, 1986 (Early Cretaceous: India).

(†) M. richmondii Tidwell, 1992 (?Triassic: Tasmania, Australia).

(†) M. sahnii (Vishnu-Mittre) Tidwell, 1986 (Early Cretaceous: India).

(†) M. santaecrucis (R.Herbst) R.Herbst, 1995 (Middle–Late Jurassic: Argentina).

(†) M. spinksii Tidwell, Munzing & M.R.Banks, 1991 (?Triassic: Tasmania, Australia).

(†) M. swanensis Tidwell, Munzing & M.R.Banks, 1991 (?Triassic: Tasmania, Australia).

(†) M. wadei (Tidwell & S.R.Rushforth) Tidwell, 1986 (Late Jurassic: Utah, USA).

(†) M. websteri Tidwell, Munzing & M.R.Banks, 1991 (?Triassic: Tasmania, Australia).

(†) M. woolfei (G.W.Rothwell, Ed.L.Taylor & T.N.Taylor) E.I.Vera, 2008 (Middle Triassic: East Antarctica).

(†) M. wrightii Tidwell, Munzing & M.R.Banks, 1991 (?Early Jurassic: Tasmania, Australia).

“Millerocaulis kolbei” type (high degree of stele perforation; see Char. 8) (†) M. kolbei (Seward) Tidwell, 1986 (Cretaceous: South Africa).

References: Archangelsky & de la Sota (1962, 1963), Bower (1926), Cantrill (1997), Cheng (2011), Cheng & Li (2007), Cheng, Wang & Li (2007), Edwards (1933), Gupta (1970), Gwynne-Vaughan (1911), Herbst (1977, 1994, 1995, 2001, 2006, 2008), Kidston & Gwynne-Vaughan (1907, 1910), Marshall (1926), Matsumoto et al. (2006), Miller (1971), Prynada (1974), Rothwell, Taylor & Taylor (2002), Seward (1907), Sharma (1973), Sinnott (1914), Tian, Wang & Jiang (2008), Tian et al. (2013, 2016), Tidwell (1986, 1992, 1994, 2002), Tidwell & Ash (1994), Tidwell & Clifford (1995), Tidwell & Rushforth (1970), Tidwell, Munzing & Banks (1991), Vera (2007, 2008, 2010), Vishnu-Mittre (1955) and Wang (1983).

2.2.3 Tribus Osmundeae Hook. ex Duby, 1828 (“modern” Osmundaceae)

(Fig. 15)

Figure 15 Planar network (neighbour-net) for all operational units of tribe Osmundeae (containing those Osmundaceae genera with extant species).

Note the divergence of the ‘branch’ comprising the subgenera Plenasium and Aurealcaulis; the divergence of subgenus Aurealcaulis; the separation of a ‘branch’ containing Todea and Leptopteris, including “Millerocaulis estipularis;” and the consistent placement of the oldest (Triassic to Jurassic) fossils between the younger members of Osmundastrum and Claytosmunda. The six-letter labels are contractions of the taxon name of an operational unit formed from the first three letters of the genus name in bold followed by the first three letters of the specific epithet; Oum, Osmundastrum.

Diagnostic axis characters: Stem core a primarily parenchymatous pith. Stelar xylem siphon typically thin (up to 15, rarely up to 20 tracheids in radial thickness) and with prominent leaf gaps. Leaf protoxylem poles initiating in mesarch position in stele, becoming endarch in stele or (rarely) in cortex. Cortex of stems and stipes two-layered, differentiated into inner, primarily parenchymatous cylinder and outer sclerenchymatous cylinder. Inner cortex usually thinner than outer cortex. Stipes with a pair of stipular wings; stipe bundle with incurved tips (i.e., more or less horseshoe-shaped). Stipe sclerenchyma ring heterogeneous: differentiation beginning in the stipe base usually with the formation of an abaxial arch of particularly thick-walled fibers that, distally, may differentiate further into massive sclerotic rings, patches, or arches in characteristic configurations.

Status: Natural, holophyletic.

Known geochronological range: Triassic to present.

Comments: Traditionally, the occurrence of a heterogenous sclerenchymatic ring in the stipe bases has been used to differentiate modern Osmundaceae (genera with extant taxa: Claytosmunda, Osmunda, Osmundastrum, Plenasium, Leptopteris, Todea) from extinct, ancient Osmundoideae (Ashicaulis–Millerocaulis group; Osmundacaulis sensu Miller; Palaeosmunda). Under the widely shared assumption that all extant Osmundaceae are holophyletic (Metzgar et al., 2008; Bomfleur, Grimm & McLoughlin, 2015; PPG I, 2016; Testo & Sundue, 2016), the heterogenous ring represents the synapomorphy for this clade. Thus, all fossil taxa with heterogenous sclerenchyma rings showing similar differentiation from an abaxial arch of thick-walled fibers should be classified as members of one of the modern genera, or be accommodated in new genera if they have character suites that clearly distinguish them from extant Osmundeae. We recognize two mutually holophyletic subtribes (see Bomfleur, Grimm & McLoughlin, 2015; but see Yatabe, Nishida & Murakami, 1999; Metzgar et al., 2008): Todeinae, the lineage leading to and including Leptopteris and Todea, and Osmundinae (former genus Osmunda), the lineage including Claytosmunda, Osmunda, Osmundastrum, and Plenasium with two subgenera (Aurealcaulis and Plenasium).

2.2.3.1 Subtribus Todeinae Bomfleur, G.W.Grimm & McLoughlin subtrib. nov.

Diagnostic stem characters: Stem core a primarily parenchymatous pith. Stelar xylem siphon typically thin (up to 15 tracheids in radial thickness) and with prominent leaf gaps. Leaf protoxylem poles initiating in mesarch position in stele, becoming endarch in stele or (rarely) in cortex, first bifurcating as leaf trace departs from stem. Cortex of stems and stipes two-layered, differentiated into inner, primarily parenchymatous cylinder and outer sclerenchymatous cylinder; inner stem cortex usually thinner than outer stem cortex; outer stem cortex heterogeneous, with a distinct ring of fibers surrounding each leaf trace. Stipes with a pair of stipular wings; stipe bundle with incurved tips (i.e., more or less horseshoe-shaped); stipe sclerenchyma ring heterogeneous, differentiating upwards into a thin band of particularly thick-walled fibers forming the outer margin of the sclerenchyma ring.

Status: Holophyletic; extant with fossil representatives.

Known geochronologic range: Early Cretaceous to present.

Comments: Our concept of a Todeinae subtribe inferred solely on the basis of stem anatomy is equivalent to the informal “leptopteroid clade” of Escapa & Cúneo (2012) inferred from the frond compression fossil record. Stems of Todeinae are readily distinct from those of their potential sister clade Osmundinae (cf. Bomfleur, Grimm & McLoughlin, 2015; but see Yatabe, Nishida & Murakami, 1999; Metzgar et al., 2008). Detailed analysis of rhizome evolution and more fossils representing this lineage will be needed to decide whether the Todeinae stems (i) can be derived from the basic type represented in Osmundinae (“paraphyletic Osmunda” scenario discussed by Bomfleur, Grimm & McLoughlin, 2015) or (ii) represent their actual sister lineage (“monophyletic Osmunda” scenario of Bomfleur, Grimm & McLoughlin, 2015). An important question in this context is also whether Osmundaceae foliage (Todea-type vs. Claytosmunda/Osmundastrum-type) found from the Triassic onwards (see Escapa & Cúneo, 2012; Grimm et al., 2015), can be associated with either one of the subtribes. So far, only two relatively young stem fossils with characteristic Todeinae anatomy have been found. Todeinae may simply be greatly under-represented in the fossil rhizome record, or earliest (pre-Cretaceous) Todeinae had a less differentiated, Osmundinae-like stem.

2.2.3.1.1 Genus Todea Willd. ex Bernh., 1801

Diagnostic axis characters: Stem core a primarily parenchymatous pith. Stelar xylem siphon typically thin (generally less than 10, rarely reaching 15 cells in radial thickness) and with prominent leaf gaps. Leaf protoxylem poles initiating in mesarch position in stele, becoming endarch in stele or (rarely) in cortex, first bifurcating as leaf trace departs from stem. Cortex of stems and stipes two-layered, differentiated into inner, primarily parenchymatous cylinder and outer sclerenchymatous cylinder; inner stem cortex usually thinner than outer stem cortex, containing a patch of thick-walled fibers adaxial to each leaf trace; outer stem cortex heterogeneous, with a distinct ring of fibers surrounding each leaf trace. Stipes with a pair of stipular wings; stipe bundle with incurved tips (i.e., more or less horseshoe-shaped); stipe inner cortex with numerous small sclerenchyma strands scattered throughout; stipe sclerenchyma ring heterogeneous, differentiating upwards into a thin band of particularly thick-walled fibers forming the outer margin of the sclerenchyma ring.

Status: Holophyletic, extant with one putative fossil representative.

Known geochronologic range: Early Cretaceous to present.

Comments: Recent phylogenetic analyses support the inclusion of the fossil Todea tidwellii in the Todea–Leptopteris lineage, but are ambiguous regarding the question whether it represents a potential precursor or relative of modern Todea (making Todea as defined here holophyletic) or related to the common ancestor of Todea and Leptopteris (making Todea, as defined here, paraphyletic) (Bomfleur, Grimm & McLoughlin, 2015). The Early Cretaceous age of Todea tidwellii coincides with the estimated divergence of the two extant genera and would allow for both scenarios (Grimm et al., 2015). In the neighbour-nets used here, Todea tidwellii groups consistently with Todea barbara (see also Table 2), and a corresponding clade would receive moderate to high support for all matrices including these taxa (BSML = 64–73; BSLS = 66–75; BSMP = 43–52), which provides ample support for the conclusion of Jud, Rothwell & Stockey (2008) that Todea tidwellii is an early respresentative of the modern genus Todea.

Table 2 Tabulation of values regarding relationships within the Todeinae and the placement of the two fossil members of the subtribe.

Split (clade in an accordingly rooted tree)	Pairwise distance	BS support	
Internal	External	ML	LS	MP	
Todeinae	Other	
Todeinae	0–0.26	N/A	0.10–0.63	46	75	<10	
Leptopteris (incl. ?L. estipularis)	0–0.15	0.14–0.26	0.10–0.57	<10	11	13	
?L. estipularis + L. wilkesiana	0.05	0.03–0.26	0.10–0.57	34	25	20	
Extant Leptopteris + Todea	0–0.24	0.05–0.26	0.17–0.63	21	27	16	
Extant Leptopteris	0–0.10	0.14–0.24	0.10–0.57	<10	26	16	
L. fraseri + Todea*	0.10–0.23	0.07–0.26	0.19–0.63	44	25	13	
L. fraseri + T. barbara*	0.14	0.10–0.25	0.19–0.63	15	10	13	
T. tidwellii + T. barbara	0.10	0.19–0.26	0.24–0.63	72	75	52	
Notes:

All other competing splits received BSML/LS/MP < 10.

All values are based on the Osmundeae-only-matrix.

* Splits that are in conflict with molecular data.

Included species: T. barbara (L.) Moore, 1857 (Extant: South Africa, Australia).

T. papuana Hennipman, 1968 (Extant: Papua New Guinea).

(†) T. tidwellii Jud, G.W.Rothwell & Stockey, 2008 (Early Cretaceous: British Columbia, Canada).

References: Seward & Ford (1903), Bower (1926), Gwynne-Vaughan (1911), Hewitson (1962), Kidston & Gwynne-Vaughan (1907), Miller (1971) and Jud, Rothwell & Stockey (2008).

2.2.3.1.2 Genus Leptopteris C.Presl, 1845

Diagnostic axis characters: Stem core a primarily parenchymatous pith. Stelar xylem siphon typically thin (up to 15, rarely up to 20 tracheids in radial thickness) and with prominent leaf gaps. Leaf protoxylem poles initiating in mesarch position in stele, becoming endarch in stele or (rarely) in cortex, first bifurcating as leaf trace departs from stem. Cortex of stems and stipes two-layered, differentiated into inner parenchymatous cylinder and outer sclerenchymatous cylinder; inner stem cortex usually thinner than outer cortex, lacking sclerenchyma patches; outer stem cortex heterogeneous, with a distinct ring of fibers surrounding each leaf trace. Stipes with a pair of stipular wings; stipe bundle with incurved tips (i.e., more or less horseshoe-shaped); stipe inner cortex lacking sclerenchyma strands; stipe sclerenchyma ring heterogeneous, differentiating distally into a thin band of particularly thick-walled fibers forming the outer margin of the sclerenchyma ring.

Status: Holophyletic, extant with one equivocal fossil record.

Known geochronologic range: Possibly Early Cretaceous to present.

Comments: A single, rather poorly preserved specimen of a fossil osmundaceous stem from Lower Cretaceous strata of the Rajmahal Hills, India, was described under the name Osmundacaulis estipularis B.D.Sharma, D.Bohra & R.Singh, 1979. The species was supposed to be distinct from most other Osmundaceae in lacking stipular wings (hence the epithet estipularis = lat. “without stipules”). We disagree with this assumption, and point out several features of the specimen based on which we propose an alternative interpretation: (1) preservation of the holotype is imperfect, and parenchyma is mostly lacking throughout the specimen; (2) there are abundant transverse sections of unusually thick roots visible between the sclerenchyma cylinders of stipes throughout the mantle of the trunk; and (3) the composition of the outer stem cortex appears heterogeneous with clearly distinct rings surrounding each leaf trace already in the outer cortex. Based on these observations we argue that the type and only specimen of Osmundacaulis estipularis represents a poorly preserved fossil of an arborescent osmundoid trunk in which the strong vertical roots penetrating downwards through the mantle, together with the imperfect preservation of parenchyma, hamper the identification of stipular wings. Furthermore, the outer stem cortex is heterogeneous in a manner typical of Todea and Leptopteris, and what is visible of the inner cortex shows no evidence of sclerenchyma patches that would be diagnostic of Todea; in fact, the entire aspect of the specimen is very similar to the basal stem section of the extant arborescent Leptopteris wilkesiana reproduced by Miller (1971: compare figs. 3 and 4 of Plate II), in which the roots have removed most of the parenchymatic stipular wings. We hypothesize that this specimen represents the only known fossil trunk of a Leptopteris yet discovered, but definite assignment should await a re-investigation of the type material.

In contrast to Todea, character suites of Leptopteris rhizomes provide a highly ambiguous phylogenetic signal as reflected in the much lower supports for a potential Leptopteris clade (BSML/LS/MP ≤ 16 including the fossil described as Osmundacaulis estipularis; BSML < 10, BSLS = 26–34; BSMP = 16–18 for a clade comprising all extant species). Under ML, a competing, presumably inaccurate split (see, e.g., Metzgar et al., 2008; Bomfleur, Grimm & McLoughlin, 2015; see also Fig. 15) associates the extant Leptopteris fraseri with the Todea spp. (BSML = 39–47). Mutual holophyly of Leptopteris and Todea, however, is well-supported by distance data (Fig. 15; Table 2).

Included species: L. fraseri (Hook. & Grev.) C.Presl, 1845 (Extant: Australia).

L. hymenophylloides (A.Rich.) C.Presl, 1845 (Extant: Australia, New Zealand).

L. superba (Colenso) C. Presl, 1848 (Extant: Australia, New Zealand).

L. wilkesiana (Brack.) Christ, 1897 (Extant: tropical Pacific islands including Fiji, Samoa, Vanuatu, New Caledonia, and possibly others).

Tentatively included: (†)?Leptopteris estipularis (B.D.Sharma, D.Bohra & R.Singh, 1979) comb. nov. (Early Cretaceous: India).

References: Bower (1926), Hewitson (1962), Kidston & Gwynne-Vaughan (1907) and Miller (1971); see also Sharma, Bohra & Singh (1979).

2.2.3.2 Subtribus Osmundinae Bomfleur, G.W.Grimm & McLoughlin subtrib. nov.

(Fig. 16)

Figure 16 Planar network (neighbour-net) for all operational units of subtribe Osmundinae (Osmundaceae, Osmundoideae, Osmundeae).

Note the divergence of the ‘branch’ comprising the closely similar subgenera Plenasium and Aurealcaulis out from the centre of the box comprising Osmunda; the divergence of subgenus Aurealcaulis; the separation of a ‘branch’ containing Todea and Leptopteris, including “Millerocaulis estipularis;” and the consistent placement of all Mesozoic fossils between the younger members of Osmundastrum and Claytosmunda. The six-letter labels are contractions of the taxon name of an operational unit formed from the first three letters of the genus name in bold followed by the first three letters of the specific epithet; Oum, Osmundastrum.

Diagnostic stem characters: Stem core a primarily parenchymatous pith. Stelar xylem siphon typically thin (up to ca 20, rarely up to 25 tracheids in radial thickness) and with prominent leaf gaps. Leaf protoxylem poles initiating in mesarch position in stele, becoming endarch in stele or (rarely) in cortex. Cortex of stems and stipes two-layered, differentiated into inner, primarily parenchymatous cylinder and outer sclerenchymatous cylinder; outer stem cortex homogeneous, thicker than inner stem cortex. Stipes with a pair of stipular wings commonly containing strands of thick-walled fibers of various shapes and sizes; stipe bundle with incurved tips (i.e., more or less horseshoe-shaped). Stipe sclerenchyma ring heterogeneous: differentiation typically initiating in the stipe base with the formation of an abaxial arch of particularly thick-walled fibers.

Status: Holophyletic (cf. Bomfleur, Grimm & McLoughlin, 2015), extant with fossil representatives.

Known geochronologic range: Triassic to present.

Comments: In analyses of molecular datasets that are based also on conserved coding chloroplast and mitochondrial genes, the outgroup-inferred root is consistently placed between Osmundastrum and a clade comprising all remaining extant Osmundaceae (Yatabe, Nishida & Murakami, 1999; Metzgar et al., 2008). Combining several lines of evidence, such as detailed analysis of the signal from molecular data, fossilized birth-death dating using frond fossils, rhizome anatomical evidence, and hybridization capacity, Bomfleur, Grimm & McLoughlin (2015) argued that this is an outgroup-ingroup (long) branching artefact. In a recent comprehensive pteridophyte analysis aiming to capture rate shifts in the evolution of fern lineages (Testo & Sundue, 2016), the distances between the hypothetical most recent common ancestor (MRCA) of living Osmundaceae to the extant species is much smaller than the distances between the Osmundaceae-MRCA and the Osmundaceae root and between the Osmundaceae-MRCA and -tips and the roots/tips of Marattiales (having diverged earlier) and Hymenophyllaceae (having diverged later; PPG I, 2016; Testo & Sundue, 2016). In such a case, ingroup–outgroup (long) branching artefacts and accordingly misplaced ingroup roots may be inevitable. Thus, we see no reason to not recognize Osmundinae as holophyletic (Bomfleur, Grimm & McLoughlin, 2015; Grimm et al., 2015). With respect to axis anatomy, the signals from the few but consistent traits that differentiate members of Millerocaulis, Todeinae and Osmundinae appear to become outcompeted by signals from convergent traits. Moreover, some early members of Osmundinae (especially of Claytosmunda) lack the derived features typical of its modern relatives, and have many presumably primitive traits shared with species of Millerocaulis. Consequently, pairwise distances between (early) members of the Osmundinae and of Millerocaulis can be smaller than those within each lineage, which may explain the poor support for an Osmundinae clade and the Osmundastrum- and Osmunda-lineages (Table 3).

Table 3 Tabulation of values regarding potential relationships of pre-Cretaceous fossils treated here as members of the Osmundastrum-lineage.

Split; (alternative) clade in an accordingly rooted tree	Pairwise distance (matrix S08)	BS support	
					Matrix S08	Matrix S09	
		Within (pot.) clade	To extant O. cinnamomeum	To extant C. claytoniana	ML	LS	MP	ML	LS	MP	
Osmundastrum cinnamomeum	Cretaceous onwards	0.03–0.10	0.03–0.10	0.15–0.30	<10	16	<10	<10	18	<10	
O. cinnamomeum (Cretaceous, Canada) + O. precinnamomeum		0.07	0.10/0.17	0.23/0.23	60	59	33	65	59	38	
O. cinnamomeum + O. precinnamomeum	Cretaceous onwards	0.03–0.17	0.03–0.17	0.15–0.30	57	51	16	56	65	17	
O. pulchellum + O. (pre)cinnamomeum	Jurassic onwards	0.03–0.17	0.03–0.17	0.10–0.30	26	22	<10	22	27	<10	
Osmundastrum (incl. O. indentatum)	Triassic onwards	0.03–0.28	0.03–0.31	0.10–0.37	<10	<10	<10	14*	<10*	<10*	
O. indentatum + Claytosmunda embreeii	Triassic–Jurassic	0.14	0.28/0.25	0.37/0.21	42	57	33	51	62	37	
O. indentatum + C. embreeii + C. sinica	Triassic–Jurassic	0.14–0.25	0.25–0.29	0.21–0.37	14	10	<10	29	13	<10	
O. indentatum + C. embreeii + C. sinica + C. preosmunda	Triassic–Jurassic	0.11–0.29	0.25–0.29	0.21–0.37	27	22	<10	35	23	<10	
Note:

* A modern Osmundastrum-O. indentatum clade would receive BSML/LS/MP = 11/11/<10; and including C. embreeii BSML/LS/MP = <10/18/<10.

In the case of Osmundinae and its genera, the practical advantage of an “evolutionary” classification (Figs. 8G and 8H) in contrast to a strict “cladistic” classification (Figs. 8D and 8E) becomes vital. Even if unambiguous fossil evidence should prove that Todeinae evolved from an Osmundinae-like ancestor and confirm the outgroup-defined root, Osmundinae can remain valid and usable as a well-defined paraphyletic taxon. A cladistic classification, on the other hand, would require major taxonomic changes: in addition to rejecting the subtribe subdivision, many early Osmundinae axes would need to remain nameless (see comments below for each genus).

We follow PPG I (2016) in recognising four genera in the Osmundinae, which have mostly been treated as subgenera of Osmunda. Even though this subdivision is supported also by vegetative and reproductive features and by molecular data (see Bomfleur, Grimm & McLoughlin, 2015), all four genera can be diagnosed and well distinguished solely on the basis of axis anatomy. Three of the genera are (likely) holophyletic (Osmundastrum, Osmunda, Plenasium), whereas Claytosmunda, which probably represents the most primitive suite of axis anatomical characters (see Bomfleur, Grimm & McLoughlin, 2015), is monotypic today but paraphyletic when fossils are included.

2.2.3.2.1 Genus Claytosmunda (Y.Yatabe, N.Murak. & K.Iwats.) Metzgar & Rouhan, 2016

Diagnostic axis characters: Stem core a primarily parenchymatous pith. Stelar xylem siphon typically thin (up to 15, rarely up to 20 tracheids in radial thickness) and with prominent leaf gaps. Leaf protoxylem poles initiating in mesarch position in stele, becoming endarch in stele or (rarely) in cortex, first bifurcating in outermost cortex or outside the stem. Cortex of stems and stipes two-layered, differentiated into inner, primarily parenchymatous cylinder and outer sclerenchymatous cylinder; outer stem cortex homogeneous, thicker than inner stem cortex. Stipes with a pair of stipular wings commonly containing strands of thick-walled fibers of various shapes and sizes; stipe bundle with incurved tips (i.e., more or less horseshoe-shaped). Stipe sclerenchyma ring heterogeneous with an abaxial arch of particularly thick-walled fibers that may further develop into two opposite, separate or thinly connected lateral masses.

Status: Partly ancestral (paraphyletic per definition) to Osmunda, Plenasium, and possibly Osmundastrum; extant with fossil representatives.

Known geochronologic range: Middle Triassic to present.

Comments: In a previous phylogenetic analysis with fewer characters and fewer operational units, we concluded that “[…] a subdivision into two putatively monophyletic subgenera Osmunda sensu Yatabe et al. and Claytosmunda generates two taxa without discriminating anatomical and morphological features” (Bomfleur, Grimm & McLoughlin, 2015: 15f). Based on the more comprehensive analysis of structural and morphological characters in Osmundales presented here, however, we agree that species of Claytosmunda are in fact distinguishable from the remaining Osmunda species based on the combination of (italicized) anatomical characters as outlined above. The plesiomorphic features of Claytosmunda include the protoxylem division occurring only outside the stem (shared with Osmundastrum) and the sporadic occurrence of mesarch leaf traces in the inner or rarely outer cortex (the one prominent difference between Claytosmunda beardmorensis and Claytosmunda claytoniana, as far as the two species can be compared). The main diagnostic rhizome trait is the composition of the heterogenous sclerenchymatic ring in the leaf trace.

Several fossils currently included in Millerocaulis are here formally transferred to Claytosmunda. In most cases, the similarity and inferred relationship of each of these species to Osmundinae in general and to genus Claytosmunda in particular (i.e., including also some species of Osmunda sensu Miller, 1971) has already been suggested by the original authors in the respective comparative discussion. For example, having noted the remarkable similarity of Osmunda plumites to Claytosmunda claytoniana, Tian and colleagues noted that the species “[…] should also be a member of this paraphyly” (Tian et al., 2014a: 217). Also Claytosmunda chengii, Claytosmunda preosmunda, Claytosmunda sinica, Claytosmunda tekelili, and Claytosmunda wangii were immediately interpreted to be close relatives and possible precursors of Claytosmunda claytoniana, the only extant representative of the genus (Cheng & Li, 2007: 258; Cheng, Wang & Li, 2007: 1357f.; Cheng, 2011: 102; Vera, 2012: 43; Tian et al., 2014b). Precursors of the remaining lineages Osmunda, Plenasium, and, possibly, Osmundastrum would probably be diagnosed as members of Claytosmunda. The paraphyletic nature of species with Claytosmunda-rhizome anatomy is well represented in the neighbour-net, where species included in this genus occupy positions between the better defined (morphologically more distinct) genera of the Osmundinae (Fig. 16; or Osmundeae: Figs. 11, 13 and 15). Claytosmunda is more or less equally similar to all other members of Osmundeae and shows no apparent preference toward any extant species. Accordingly, no member of Claytosmunda, including the extant Claytosmunda claytoniana, would be supported as a member of a clearly defined clade (BSML/LS/MP ≤ 10; File S3; see the BS support networks in Grimm, Bomfleur & McLoughlin (2017) available from Dryad Digital Repository http://dx.doi.org/10.5061/dryad.270gs).

Although Claytosmunda nathorstii from the Paleogene of Svalbard is known only from isolated stipes and was thus excluded from the phylogenetic analyses owing to it lacking too many characters, the species is clearly allied with Claytosmunda based on its characteristic stipe structure with two lateral masses of thick-walled fibers in the sclerenchyma cylinder and with an elongate mass of fibers in each stipular wing (Miller, 1967).

Included species: C. claytoniana (L., 1753) Metzgar & Rouhan, 2016 (Extant: East Asia and eastern North America).

(†) C. beardmorensis (J.M.Schopf, 1978) comb. nov. (Middle Triassic: East Antarctica).

(†) C. chengii nom. nov. (Middle Jurassic: Liaoning, China).

(†) C. johnstonii (Tidwell, Munzing & M.R.Banks, 1991) comb. nov. (?Early Jurassic: Tasmania, Australia).

(†) C. embreii (Stockey & S.Y.Sm., 2000) comb. nov. (Early Cretaceous: California, USA).

(†) C. liaoningensis (Wu Zhang & Shao-Lin Zheng, 1991) comb. nov. (Middle Jurassic: Liaoning, China).

(†) C. nathorstii (C.N.Mill., 1967) comb. nov. (Palaeogene: Svalbard).

(†) C. plumites (N.Tian & Y.D.Wang 2014[a]) comb. nov. (Middle Jurassic: Liaoning, China).

(†) C. preosmunda (Y.M.Cheng, Yu F.Wang & C.S.Li, 2007) comb. nov. (Middle Jurassic: Liaoning, China).

(†) C. sinica (Y.M.Cheng & C.S.Li, 2007) comb. nov. (Middle Jurassic: Liaoning, China).

(†) C. tekelili (E.I.Vera, 2012) comb. nov. (Early Cretaceous: West Antarctica).

(†) C. wangii (N.Tian & Y.D.Wang, 2014[b]) comb. nov. (Middle Jurassic: Liaoning, China).

(†) C. wehrii (C.N.Mill., 1982) comb. nov. (Miocene: Washington, USA).

References: Bomfleur, Grimm & McLoughlin (2015), Bower (1926), Cheng (2011), Cheng & Li (2007), Cheng, Wang & Li (2007), Hewitson (1962), Kidston & Gwynne-Vaughan (1907), Miller (1967, 1971, 1982), Schopf (1978), Stockey & Smith (2000), Tian et al. (2014a, 2014b), Tidwell, Munzing & Banks (1991), Vera (2012), Yatabe, Murakami & Iwatsuki (2005) Refe and Zhang & Zheng (1991).

2.2.3.2.2 Genus Osmundastrum C.Presl, 1847

Diagnostic stem characters: Stem core a primarily parenchymatous pith. Stelar xylem siphon thin and with prominent leaf gaps. Leaf protoxylem poles initiating in mesarch position in stele, becoming endarch in stele, first bifurcating in outermost cortex or outside the stem. Cortex of stems and stipes two-layered, differentiated into inner, primarily parenchymatous cylinder and outer sclerenchymatous cylinder; outer stem cortex homogeneous, thicker than inner stem cortex. Stipes with a pair of stipular wings commonly containing strands of thick-walled fibers of various shapes and sizes; stipe bundle with incurved tips (i.e., more or less horseshoe-shaped). Stipe sclerenchyma ring heterogeneous with an abaxial arch or mass and two opposite, separate or thinly connected lateral masses of particularly thick-walled fibers.

Status: Holophyletic; extant with fossil representatives.

Known geochronologic range: Triassic to present.

Comments: The recently described Osmunda pulchella Bomfleur, G.W.Grimm & McLoughlin from the Lower Jurassic of Sweden is sufficiently similar to extant and other fossil Osmundastrum species to warrant assignment to this genus (Bomfleur, Grimm & McLoughlin, 2015; see also Grimm et al., 2015; Fig. 16), even though it lacks the fiber patch adaxial to each leaf trace that is characteristic of the other species. We also include another pre-Cretaceous species in Osmundastrum, Osmundastrum indentatum, which is less derived and, as a consequence, overall more similar to the basic type within Osmundinae as seen in members of Claytosmunda and Osmunda (to some degree), and accordingly placed in the neighbour-nets.

Osmundastrum precinnamomeum from the Paleocene of North America (Miller, 1967) is more distinct from extant Osmundastrum cinnamomeum than other specimens that have been included in the latter species, thus we accept is as a separate species. Whether (some) fossil Osmundastrum cinnamomeum are better treated as distinct species requires a better understanding of the extant diversity in this species. We note that the diversity evident in the fossils is as high or higher than between species of other (extinct and extant) genera (see also File S3); a Osmundastrum cinnamomeum-only clade would receive low or diminishing support. Overall these modern-type Osmundastrum rhizomes (Osmundastrum cinnamomeum, Osmundastrum precinnamomeum) nevertheless form a coherent group distinct from other Osmundinae or Osmundoideae; a corresponding clade would receive moderate support under ML and LS (low under MP; Table 3). Osmundastrum pulchellum (Bomfleur, G.W.Grimm & McLoughlin) comb. nov. from the Lower Jurassic of Sweden is a probable precursor of modern Osmundastrum, which ranges from the Late Cretaceous onwards (Bomfleur, Grimm & McLoughlin, 2015). Hence, we include it in this holophyletic genus, even though it lacks the fiber patch adaxial to each leaf trace that is characteristic of the modern members. The intermediate, bridging nature of this fossil is also apparent in the neighbour-nets provided here, where it consistently occupies a position between modern Osmundastrum and early representatives with putatively primitive axis anatomy as seen in early Claytosmunda. This placement underscores its great similarity to its more derived congeners (MD ≥ 0.08) and to early members of the Claytosmunda–Osmunda(–Plenasium) lineage (MD ≥ 0.10). If forced into a tree, Osmundastrum pulchellum would be placed close to modern Osmundastrum (Table 3; see also BS support networks in Grimm, Bomfleur & McLoughlin (2017) available from Dryad Digital Repository http://dx.doi.org/10.5061/dryad.270gs), as expected for an early member of this lineage.

Molecular dating using the frond fossil record confirms a Triassic split between Osmundastrum and the remainder of the Osmundinae (Bomfleur, Grimm & McLoughlin, 2015; Grimm et al., 2015). Triassic Osmundaceae fossils include Osmundinae rhizomes with a heterogenous sclerenchyma ring in the leaf traces, usually exhibiting the pattern typical of many Jurassic Osmundinae and the extant Claytosmunda claytoniana. An exception is Australosmunda indentata (Hill, Forsyth & Green, 1989), which has the typical Osmundastrum configuration. In other aspects (e.g., stele dissection), this species is most similar to the Jurassic Claytosmunda embreii, and to a lesser degree, Claytosmunda sinica and Claytosmunda preosmunda; corresponding clades would receive low but consistent support based on our matrices (Table 3). Given the high conservatism and diagnostic value of the organization of the heterogenous sclerenchyma ring in all extant members of Osmundaceae (Osmundeae; see, e.g., Hewitson, 1962; Miller, 1967, 1971; Bomfleur, Grimm & McLoughlin, 2015; this study), we nevertheless regard Millerocaulis indentata as an early member of the Osmundastrum lineage. We consider its association with early Claytosmunda spp. in the graphs to be a methodological artefact due to the dominance of shared primitive (such as a nearly imperforate stele and commonly mesarch basal leaf traces) or generally homoplastic features encoded in our matrix among the earliest members of both lineages within Osmundinae (compare also the neighbour-nets with potential tree, BS-best supported topologies). This example cautions that classification of fossil taxa should always consider the actual traits shared or not by two taxa resolved as potential sisters in trees or grouped in networks. A further hint to the existence of two already differentiated, modern lineages in the Triassic is the higher diversity between the Triassic Osmundinae species (MD = 0.14 between the purported sister species Osmundastrum indentatum and Claytosmunda embreii; MD = 0.42 between Claytosmunda beardmorensis and Osmundastrum indentatum) in comparison to the situation from the Jurassic onwards (e.g., MD ≥ 0.08 vs. ≥ 0.10 between Osmundastrum pulchellum and other Osmundastrum or Claytosmunda/Osmunda).

Included species: O. cinnamomeum (L.) C.Presl, 1847 (Extant: East Asia, eastern North America to eastern South America; Miocene: Hokkaido, Japan and Washington, USA; Cretaceous: Alberta, Canada).

(†) O. precinnamomeum (C.N.Mill., 1967), comb. nov. (Paleocene: North Dakota, USA).

(†) O. pulchellum (Bomfleur, G.W.Grimm & McLoughlin, 2015) comb. nov. (Early Jurassic: Sweden).

(†) O. indentatum (R.S.Hill, S.M.Forsyth & F.Green, 1989) comb. nov. (Triassic: Tasmania, Australia).

References: Bomfleur, McLoughlin & Vajda (2014), Bomfleur, Grimm & McLoughlin (2015), Bower (1926), Faull (1901, 1910), Grimm et al. (2015), Gwynne-Vaughan (1911), Hewitson (1962), Hill, Forsyth & Green (1989), Kidston & Gwynne-Vaughan (1907), Matsumoto & Nishida (2003), Miller (1967, 1971), Serbet & Rothwell (1999) and Wardlaw (1946).

2.2.3.2.3 Osmunda L., 1753

Diagnostic axis characters: Stem core a primarily parenchymatous pith. Stelar xylem siphon thin and with prominent leaf gaps. Leaf protoxylem poles initiating in mesarch position in stele, becoming endarch in stele or rarely in cortex; leaf trace either arising with single protoxylem strand that bifurcates in inner or outer cortex or rarely arising from stele already with two protoxylem strands. Cortex of stems and stipes two-layered, differentiated into inner parenchymatous cylinder and outer sclerenchymatous cylinder; outer stem cortex homogeneous, thicker than inner stem cortex. Stipes with a pair of stipular wings commonly containing strands of thick-walled fibers of various shapes and sizes; stipe bundle with incurved tips (i.e., more or less horseshoe-shaped). Stipe sclerenchyma ring heterogeneous developing two opposite lateral masses that may extend and conjoin into an adaxial arch of particularly thick-walled fibers.

Status: Holophyletic; extant with fossil representatives.

Known geochronologic range: Paleocene to present.

Comments: Based only on axis anatomy, it is difficult to distinguish clearly between Osmunda and the paraphyletic Claytosmunda, which comprises stem group members of the Osmundinae and members of the actual lineage leading to the extant Claytosmunda claytoniana (note the placement of the extant Osmunda lancea in neighbour-nets). According to a recent molecular dating that makes use of the entire rhizome and frond fossil record of modern Osmundaceae, the Claytosmunda claytoniana-lineage diverged from the Osmunda–Plenasium lineage sometime in the Jurassic (Bomfleur, Grimm & McLoughlin, 2015; Grimm et al., 2015), with Osmunda and Plenasium separating in the Cretaceous. Thus, Jurassic and younger Osmundinae fossils likely represent members of either the Claytosmunda claytoniana lineage or the Osmunda–Plenasium lineage. However, Paleogene fossils (Osmunda oregonensis, Osmunda pluma) still appear intermediate between extant members of Osmunda and Claytosmunda in their development of leaf trace protoxylem and development of fibre masses in the sclerenchyma ring. This is also well reflected by their intermediate position in the neighbour-nets. Some fossil axes show the modern Osmunda feature of the leaf trace protoxylem dividing already in the inner cortex, but also the retention of just two separate lateral fibre masses in the sclerenchyma ring instead of an adaxial arch, as is typical of Claytosmunda. Hence, we restrict Osmunda to only those species that have the adaxial arch in the sclerenchyma ring, and retain other early potential members of the Osmunda-lineage in Claytosmunda until more explicit phylogenetic hypotheses can be presented.

Included species: O. japonica Thunb., 1780 (Extant: East Asia).

O. lancea Thunb., 1784 (Extant: East Asia).

O. regalis L., 1753 (Extant: eastern North to eastern South America, Europe, Asia, Southern Africa).

(†) O. ilianensis C.N.Mill., 1967 (Miocene: Hungary, Austria).

(†) O. oregonensis (C.A.Arnold) C.N.Mill., 1967 (Eocene: Oregon, USA).

(†) O. pluma C.N.Mill., 1967 (Paleocene: North Dakota, USA).

(†) O. shimokawaensis M.Matsumoto & H.Nishida, 2003 (Miocene: Hokkaido, Japan).

References: Arnold (1945, 1952), Bower (1926), Faull (1901), Gwynne-Vaughan (1911), Hewitson (1962), Kidston & Gwynne-Vaughan (1907, 1910), Matsumoto & Nishida (2003), Miller (1967, 1971) and Unger (1854).

2.2.3.2.4 Plenasium C.Presl, 1836

Diagnostic stem characters: Stem core a primarily parenchymatous pith. Stelar xylem siphon with prominent leaf gaps. Leaf protoxylem poles initiating in mesarch position in stele, becoming endarch in stele; leaf trace adaxially curved upon departure from stele, usually arising with two protoxylem strands from two adjacent stelar xylem segments. Cortex of stems and stipes two-layered, differentiated into inner parenchymatous cylinder and outer sclerenchymatous cylinder; outer stem cortex homogeneous, thicker than inner stem cortex. Stipes with a pair of stipular wings typically containing numerous, small scattered strands of thick-walled fibres; stipe bundle with incurved tips (i.e., more or less horseshoe-shaped). Stipe sclerenchyma ring heterogeneous, differentiating distally into a thin band of particularly thick-walled fibres forming the outer margin of the sclerenchyma ring.

Status: Holophyletic; extant with fossil representatives.

Known geochronologic range: Early Cretaceous to present.

Comments: Plenasium species differ from all remaining Osmundoideae in that their leaf traces originate typically from two independent protoxylem poles in two adjacent xylem segments.

2.2.3.2.4.1 (†) Plenasium subgenus Aurealcaulis (Tidwell & L.R.Parker), 1987, comb. et stat. nov.

Diagnostic characters: Stems reaching great size; steles highly perforate and dissected, comparatively thick (up to ca 25 tracheids in radial thickness); leaf traces usually originating from two independent protoxylem poles from two adjacent stelar xylem segments (similar to Plenasium), departing from stele usually in the form of two separate bundles that fuse into a single C-shaped strand in their course through the cortex. Cortex of stems and stipes two-layered, differentiated into inner parenchymatous cylinder and outer sclerenchymatous cylinder; outer stem cortex homogeneous, thicker than inner stem cortex. Stipes with a pair of stipular wings typically containing a single or numerous distinct masses, commonly in elongate shape or arrangement; stipe bundle with incurved tips (i.e., more or less horseshoe-shaped). Stipe sclerenchyma ring heterogeneous, differentiating distally into a thin band of particularly thick-walled fibres forming the outer margin of the sclerenchyma ring.

Status: Possibly ancestral (paraphyletic per definition) to subgenus Plenasium; extinct.

Known geochronologic range: Early Cretaceous to ?mid-Eocene.

Comments: Since Aurealcaulis was established (Tidwell & Parker, 1987), it has been considered fundamentally different from other Osmundaceae because of its allegedly exarch leaf trace protoxylem and its leaf traces arising in the form of two separate masses. The first feature is plainly a misidentification: the putative “protoxylem cells” that would render trace formation exarch (see arrow in fig. 17 of Tidwell & Parker, 1987) are only smaller metaxylem cells at the stele periphery, just as in the stelar xylem of any other osmundaceous fern. The centre-left of the same figure shows a stelar xylem segment with an actual protoxylem pole in mesarch position.

The second feature is that a given leaf trace usually arises in the form of two separate segments—each with one inward-facing protoxylem pole—that fuse into one C-shaped segment inside the cortex only at some distance from the stele. This seemingly unique feature, however, is merely an extreme form of the mode of leaf trace formation represented in Plenasium. In extant Plenasium species, the tips of the two meristele segments first fuse with each other and then separate from the stele in the form of a single, deeply C-shaped trace (Miller, 1971). The Eocene Plenasium arnoldii and Plenasium dowkeri are somewhat intermediate in that curvature of the basal leaf trace is so strong and the thickness of the strand so uneven that they adopt a curved-dumbbell shape, with two thick xylem segments connected only by a thin band of tracheids (see, e.g., Chandler, 1965: figs. 3, 4, 6, 7 and 17; Arnold, 1952: figs. 13, 14 and 16). In fact, a few leaf traces appear to depart from the stele in the typical Aurealcaulis manner, i.e., in the form of two completely separate segments (see, e.g., Chandler, 1965: figs. 6 and 7). Conversely, basal leaf traces of Aurealcaulis arise sporadically in the form of a deeply indented, but nonetheless intact C-shaped strand just like those typical of Plenasium chandleri and Plenasium arnoldii. This type of leaf trace formation is indeed common enough that Tidwell and Parker referred to it as “the other type of trace formation” in the original description of Aurealcaulis (Tidwell & Parker, 1987: 807). Hence, the supposedly unique mode of trace formation in Aurealcaulis is one end member of a spectrum that also encompasses forms typical of extinct and, to a lesser degree, extant Plenasium species.

Another problematic diagnostic feature relates to the heterogeneity of cells in the stipe sclerenchyma ring—the distinctive feature of all “modern” Osmundoideae. Most recent taxonomic treatments of fossil Osmundaceae have adopted characters from the original publication without critical appraisal: e.g., that the sclerotic ring in Aurealcaulis petioles were homogeneous (Tidwell & Parker, 1987: 805; see, e.g., Tian, Wang & Jiang, 2008; Wang et al., 2014a, 2014b). However, Cheng and Li (2007) already noted that the stipe sclerenchyma ring of Aurealcaulis has a heterogeneous composition. Tidwell & Parker (1987: 809) wrote in their original description “the cells comprising this ring are generally uniform in size (50–70 μm in diam.) and wall thickness (10–25 μm). However, a layer of fibres near the edge of the sclerotic ring is much thicker-walled and appears to be more resistant to decay than the remainder of the ring” (Tidwell & Parker, 1987: 809). We consider this to represent just different phraseology for describing the type of heterogeneous sclerenchyma ring that occurs in Plenasium: “While the sclerenchyma ring of the petiole base appears homogeneous in species of Plenasium, there is a very thin abaxial arch of thick-walled fibres visible in transverse sections near the stem. In sections more distant from the stem, the ring is completely surrounded by a narrow layer of these cells, and this situation persists throughout the stipular region” (Miller, 1967: 181)—or, alternatively: “thick-walled fibres in sclerenchyma ring thin near attachment to stem extending to surround ring as a narrow band in lower one-third of stipular region” (Miller, 1971: 31).

Finally, Aurealcaulis has been described to differ from all other Osmundaceae in that its roots arise directly from the trace and not from the stele (Tidwell & Parker, 1987: 811). However, this is the regular mode of root formation in other Osmundaceae (Bower, 1926; Hewitson, 1962; Miller, 1967, 1971).

With these supposed differences revised, we identify Aurealcaulis as being: (1) a typical member of Osmundaceae with mesarch protoxylem in the stem and endarch protoxylem in the leaf traces; (2) a member of Osmundeae (modern Osmundoideae) with a heterogeneous stipe sclerenchyma ring similar to those of Todea, Leptopteris, and Plenasium; (3) an extinct member of Osmundinae owing to its homogeneous outer cortex (unlike Todeinae; Tidwell & Medlyn, 1991); and (4) a close relative and possible ancestor of extant species of Plenasium owing to their peculiar similarities of having two protoxylem initials per leaf trace, leaf traces arising in the form of two separate or thinly connected deeply indented masses, and a similar type of development of the sclerenchyma ring.

Included species: (†) P. bransonii (Tidwell & Medlyn, 1991) comb. nov. (?Eocene: New Mexico, USA).

(†) P. burgii (Tidwell & J.E.Skog, 2002) comb. nov. (Early Cretaceous: Nebraska, USA).

(†) P. crossii (Tidwell & L.R.Parker, 1987) comb. nov. (Paleocene: Wyoming, USA).

(†) P. dakotense (Tidwell & J.E.Skog, 2002) comb. nov. (Early Cretaceous: South Dakota, USA).

(†) P. moorei (Tidwell & Medlyn, 1991) comb. nov. (?Eocene: New Mexico, USA).

(†) P. nebraskense (Tidwell & J.E.Skog, 2002) comb. nov. (Early Cretaceous: Nebraska, USA).

References: Arnold (1945, 1952), Chandler (1965), Hewitson (1962), Miller (1967, 1971), Tidwell & Medlyn (1991), Tidwell & Parker (1987) and Tidwell & Skog (2002).

2.2.3.2.4.2 Plenasium subgenus Plenasium

Diagnostic stem characters: Stem core a primarily parenchymatous pith. Stelar xylem siphon thin (usually up to ca 15 tracheids in radial thickness) and with prominent leaf gaps; leaf protoxylem poles initiating in mesarch position in stele, becoming endarch in stele; leaf trace strongly curved adaxially upon departure from stele, usually arising with two protoxylem strands from two adjacent stelar xylem segments. Cortex of stems and stipes two-layered, differentiated into inner parenchymatous cylinder and outer sclerenchymatous cylinder; outer stem cortex homogeneous, thicker than inner stem cortex. Stipes with a pair of stipular wings typically containing numerous, small scattered strands of thick-walled fibres; stipe bundle with incurved tips (i.e., more or less horseshoe-shaped). Stipe sclerenchyma ring heterogeneous, differentiating distally into a thin band of particularly thick-walled fibres forming the outer margin of the sclerenchyma ring.

Status: Holophyletic; extant with fossil representatives.

Known geochronologic range: Paleocene to present.

Included species: P. banksiifolium (C.Presl) C.Presl, 1836 (Extant: East and Southeast Asia).

P. bromeliifolium (C.Presl) C.Presl, 1836 (Extant: East and Southeast Asia).

P. javanicum (Blume) C.Presl, 1848 (Extant: Southeast Asia).

P. vachelii C.Presl, 1848 (Extant: Southeast Asia).

(†) P. arnoldii (C.N.Mill., 1967), comb. nov. (Paleocene: North Dakota, USA).

References: Chandler (1965), Hewitson (1962), Kidston & Gwynne-Vaughan (1907) and Miller (1967, 1971).

2.2.3.2.4.? Plenasium subgenus indet.

Comments: Plenasium dowkeri and Plenasium chandleri from the Palaeogene of North America and Europe occupy intermediate positions between species of subgenus Plenasium and species of subgenus Aureacaulis in the neighbour-nets (Figs. 11, 13, 15 and 16). Information on axis anatomy is at present inadequate to determine whether these two species are better assigned to subgenus Plenasium or subgenus Aurealcaulis. Therefore, we retain these species simply as Plenasium subgenus indet.

(†) P. chandleri (C.A.Arnold, 1952) comb. nov. (Eocene: Oregon, USA).

(†) P. dowkeri (Carruth., 1870) comb. nov. (Paleocene: North Dakota, USA; UK).

2.? Family Osmundaceae, subfamily unknown 2.?.1 (†) Genus Osmundacaulis C.N.Mill., 1967 emend. Tidwell, 1986

(Fig. 17)

Figure 17 Planar network (neighbour-net) for all operational units of genus Osmundacaulis (subfamily incertae sedis, Osmundaceae).

The three-letter labels indicate the first three letters of the specific epithet of the relevant taxon.

Diagnostic characters: Axes usually arborescent or erect, rarely rhizomatous. Cortex of stems and stipes two-layered, differentiated into inner, primarily parenchymatous cylinder and outer sclerenchymatous cylinder; inner stem cortex thicker than outer stem cortex. Peripheral xylem siphon with prominent leaf gaps, very thick (typically >30 tracheids in radial thickness), commonly highly perforated (except Osmundacaulis lemonii, Osmundacaulis nerii, Osmundacaulis tidwellii, and Osmundacaulis whittlesii); phloem external and internal and sporadically connecting through leaf gaps (i.e., dictyostelic). Leaf trace protoxylem initially single and in mesarch position in stele, in most cases dividing repeatedly in stele and inner cortex (except Osmundacaulis bamfordae); leaf trace strongly curved already upon departure from stele. Stipes with a pair of stipular wings; stipe bundle with incurved tips (i.e., horseshoe-shaped).

Status: Putatively holophyletic, extinct.

Known geochronologic range: Jurassic to Cretaceous.

Comments: Osmundacaulis is clearly distinct from other Osmundaceae in having an inner cortex that is thicker than the outer cortex, in having very thick and typically highly perforated stelar metaxylem cylinders, and in having leaf traces that are already strongly curved at the point of departure from the stele. These features served to distinguish first the species Osmundites skidegatensis Penh. (see Kidston & Gwynne-Vaughan, 1908) and later the “Osmundacaulis skidegatensis group” (Miller, 1971) that subsequently formed Osmundacaulis Miller emend. Tidwell as currently understood. Osmundacaulis contains a disparate array of morphologically distinct species that, analogous to Millerocaulis, are transitional from having highly perforated steles (e.g., Osmundacaulis skidegatensis) to only a few (e.g., Osmundacaulis nerii, Osmundacaulis tidwellii), or no perforations at all (Osmundacaulis lemonii).

Included species: (†) O. andrewii Tidwell & Pigg, 1993 (?Early Jurassic: Tasmania, Australia).

(†) O. atherstonei (Schelpe) C.N.Mill., 1971 (Early Cretaceous: South Africa).

(†) O. bamfordae R.Herbst, 2015 (Early Cretaceous: South Africa).

(†) O. griggsii Tidwell & Pigg, 1993 (Early Jurassic: Tasmania, Australia).

(†) O. hoskingii R.E.Gould, 1973 (Middle Jurassic: Queensland, Australia).

(†) O. janae Tidwell & Pigg, 1993 (Early Jurassic: Tasmania, Australia).

(†) O. jonesii Tidwell, 1987 (Early Jurassic: Tasmania, Australia).

(†) O. lemonii Tidwell, 1990 (Late Jurassic: Utah, USA).

(†) O. natalensis (Schelpe) C.N.Mill., 1971 (Early Cretaceous: South Africa).

(†) O. nerii Tidwell & Ross Jones, 1987 (Early Jurassic: Tasmania, Australia).

(†) O. pruchnickii Tidwell & Pigg, 1993 (Early Jurassic: Tasmania, Australia).

(†) O. richmondii Tidwell & Pigg, 1993 (Early Jurassic: Tasmania, Australia).

(†) O. skidegatensis (Penh.) C.N.Mill., 1967 (Early Cretaceous: British Columbia, Canada).

(†) O. tasmanensis Tidwell & Pigg, 1993 (Early Jurassic: Tasmania, Australia).

(†) O. tehuelchensis R.Herbst, 2003 (Middle Jurassic: Argentina).

(†) O. tidwellii R.Herbst, 2015 (Early Cretaceous: South Africa).

(†) O. whittlesii McKenzie, A.Smith, G.W.Rothwell et al., 2015 (Early Cretaceous: British Columbia, Canada).

(†) O. zululandensis R.Herbst, 2015 (Early Cretaceous: South Africa).

References: Bower (1926), Gould (1973), Herbst (2003, 2008, 2015), Kidston & Gwynne-Vaughan (1907), Miller (1971), Penhallow (1902a, 1902b), Schelpe (1955, 1956), Smith, Rothwell & Stockey (2015), Tidwell (1986, 1987, 1990, 2002), Tidwell & Jones (1987) and Tidwell & Pigg (1993).

2.?.2 (†) Genus Shuichengella Z.M.Li, 1993

Diagnostic characters: Axes large, erect to arborescent. Cortex of stems and stipes two-layered, differentiated into inner, primarily parenchymatous cylinder and outer sclerenchymatous cylinder; inner stem cortex much thicker than outer stem cortex, containing numerous (up to 60) leaf traces in a given transverse section. Peripheral xylem siphon with sporadic narrow, complete leaf gaps, thin (typically ca 10 cells in radial thickness). Leaf trace strongly curved upon departure from stele, containing three protoxylem clusters in inner cortex.

Status: Monotypic, extinct.

Known geochronologic range: Late Permian.

Comments: The precise systematic affinities of Shuichengella primitiva within Osmundaceae are unknown. However, it is notable that the genus shares some rather unusual characters with Osmundacaulis, e.g., an inner cortex that is much thicker than the outer cortex, and the strong curvature and multiple protoxylem clusters of leaf traces already differentiated upon departure from the stele. Li (1993) instituted the subfamily Shuichengelloideae for the monotypic genus; however, considering that, for instance, important information on stipe anatomy is lacking (see Tidwell & Ash, 1994) and that insufficient comparison can be made with other taxa, we consider Shuichengella to be too poorly known to warrant erection of a separate subfamily.

Included species: (†) S. primitiva (Z.M.Li) Z.M.Li, 1993 (Late Permian: Guizhou, China).

References: Li (1983, 1993) and Tidwell & Ash (1994).

2.?.3 (†) Genus Anomorrhoea Eichw., 1860

Status: Nomen dubium, extinct.

Known geochronologic range: Late Permian.

Comments: The holotype is a fragment of the mantle of roots and stipe bases with a small portion of outer cortex attached; stele, inner cortex, and most of the outer cortex is missing. It is, thus, too fragmentary to be identified with certainty (Kidston & Gwynne-Vaughan, 1909; Zalessky, 1927; Miller, 1971). The sclerotic outer cortex and prominent stipular wings indicate affinities with Osmundaceae, but the lack of information on the stem core composition makes it impossible to determine whether, for example, the genus belongs to Osmundoideae or to Thamnopteroideae. We propose that use of the name should be abandoned until better and more completely preserved material enables more detailed comparison.

Included species: (†) A. fischeri Eichw., 1860 (Late Permian: Russia; here considered Osmundaceae gen. indet.).

References. Kidston & Gwynne-Vaughan (1909), Zalessky (1927) and Miller (1971).

2.?.4 (†) Genus Bathypteris Eichw., 1860

Diagnostic characters: Stems forming large, arborescent trunks. Stem core consisting of tracheids with scalariform thickenings. Peripheral metaxylem siphon entire and imperforate. Cortex of stems and stipes two-layered, differentiated into inner parenchymatous cylinder and outer sclerenchymatous cylinder. Leaf trace protoxylem single. Stipes lacking stipular wings but bearing multicellular spines; stipe bundle with incurved tips (i.e., more or less horseshoe-shaped), surrounded by sclerotic cells that form a narrow lining band. Sclerenchyma ring in stipe base circular to elliptic in cross-section, homogenous.

Status: Monotypic, extinct.

Known geochronologic range: Late Permian.

Comments: The affinities of Bathypteris cannot be determined precisely; it has a two-layered cortex and horseshoe-shaped stipe bundle typical of Osmundaceae, but its petiole surfaces bear multicellular spines instead of the characteristic stipular wings—a feature otherwise known only from Itopsidema among Osmundales (Daugherty, 1960).

Included species: (†) B. rhomboidalis (S.Kutorga) Eichw., 1860 (Late Permian: Russia).

References: Kidston & Gwynne-Vaughan (1909), Zalessky (1927) and Miller (1971).

Conclusion and Outlook

This study provides a comprehensive comparative dataset and the methodological tool-kit to describe, identify and classify fossil osmundalean axes. A phylogenetic framework assigns explicit biological meaning to the names of fossil and extant taxa. The design of our character matrix is optimized for these purposes. Information on the anatomy of forthcoming finds of fossil osmundalean axes can easily be incorporated into the matrix in order to facilitate systematic placement, comparisons, and character analysis. We propose natural groups that can be identified based on diagnostic character suites, and we aim to reconcile the extensive fossil record of the group in the form of anatomically preserved axes with modern phylogenies and systematic treatments of extant taxa based on molecular data. In light of the wealth of information about the axis anatomy in extant and fossil Osmundales, the use of artificial form-genera of osmundalean axes is, in our opinion, no longer needed; overall, axis anatomy has proven so informative for the systematic interpretation of fossil Osmundales that in case an anatomically preserved axis cannot be identified with certainty, it should be described in open nomenclature instead of being formally described. With the artifice of accepting also paraphyletic taxa as valid taxonomic units, we expect our systematic framework to be particularly stable and widely applicable, yet adaptable in case new data warrant erection of new groups. Furthermore, we hope to have set a new basis for in-depth analyses of evolutionary trends in Osmundales. The many homoplasious characters in our matrix make it difficult to infer explicit scenarios for the evolution of the order; a first necessary step will, therefore, be to assess which characters or character suites are compatible with the molecular tree. Ideally, this will make it possible to estimate probabilities for character changes, which can then be included in subsequent tree inferences (e.g., as character weights in a ML or Bayesian framework). Total evidence (TE) analysis would benefit from using approaches that treat fossils not as terminal taxa, but place them according to their age (Ronquist et al., 2012), but see the result of TE dating for the modern Osmundoideae lineage provided in the supplement to Grimm et al., 2015). Much of the diversity and disparity of osmundalean axes is, however, concentrated in lineages with no extant representatives, and the evolutionary history of Osmundales spans at least 255 Ma. Our revision indicates that various lineages of Osmundales gave rise to similar adaptations during different time intervals (“temporal convergences”). Thus, it is important to assess diversity and phylogenetic relationships independently for each time slice, and compare the results to the preceding or subsequent time slice. A promising experiment may be to add also hypothetical ancestors of the extant genera placed in the corresponding time period, defined by a set of characters reconstructed using probabilistic ancestral state reconstruction methods (Felsenstein, 2012; Revell, 2012). A sound evolutionary hypothesis for Osmundales should include horizontal contemporaneous diversity patterns as well as vertical (along phylogenetic lineages) evolutionary trends.

Appendix A: Formal Taxonomic Treatment of Nomenclatural Novelties

Guaireaceae subfam. Itopsidemoideae subfam. nov.

Name-bringing genus: Itopsidema Daugherty (in American Journal of Botany 47: 775. 1960).

Diagnosis: Stems radially symmetrical with spiral phyllotaxis. Stem core a parenchymatic pith with variable amounts of interspersed sclereids and tracheids; stele with a distinct peripheral metaxylem siphon lacking discrete leaf gaps and consisting of a spongy admixture of metaxylem and more or less diffusely interspersed patches of parenchyma. Stem cortex primarily parenchymatous and not differentiated into distinct layers; stipe without stipular wings and sclerenchyma ring; stipe vascular bundle with recurved tips (i.e., more or less inverse-omega-shaped). Roots commonly arising from abaxial side of leaf trace within the stem cortex.

Tribus Osmundeae subtribus Osmundinae subtrib. nov.

Name-bringing genus: Osmunda L. (in Species Plantarum 2: 1063. 1753)

Diagnosis: Stem core a primarily parenchymatous pith. Stelar xylem siphon comparatively thin and with prominent leaf gaps. Leaf protoxylem poles initiating in mesarch position in stele, becoming endarch in stele or (rarely) in cortex. Cortex of stems and stipes two-layered, differentiated into inner, primarily parenchymatous cylinder and outer sclerenchymatous cylinder; outer stem cortex homogeneous, thicker than inner stem cortex. Stipes with a pair of stipular wings commonly containing strands of thick-walled fibres of various shapes and sizes; stipe bundle with incurved tips (i.e., more or less horseshoe-shaped). Stipe sclerenchyma ring heterogeneous: differentiation typically initiating in the stipe base with the formation of an abaxial arch of particularly thick-walled fibres.

Tribus Osmundeae subtribus Todeinae subtrib. nov.

Name-bringing genus: Todea Willd. ex Bernh. (in Schraders Jahrbuch der Botanik 1800(2): 126. 1801)

Diagnosis: Stem core a primarily parenchymatous pith. Stelar xylem siphon comparatively thin and with prominent leaf gaps. Leaf protoxylem poles initiating in mesarch position in stele, becoming endarch in stele or (rarely) in cortex, first bifurcating as leaf trace departs from stem. Cortex of stems and stipes two-layered, differentiated into inner, primarily parenchymatous cylinder and outer sclerenchymatous cylinder; inner stem cortex usually thinner than outer stem cortex; outer stem cortex heterogeneous, with a distinct ring of fibres surrounding each leaf trace. Stipes with a pair of stipular wings; stipe bundle with incurved tips (i.e., more or less horseshoe-shaped); stipe sclerenchyma ring heterogeneous, differentiating upwards into a thin band of particularly thick-walled fibres forming the outer margin of the sclerenchyma ring.

Millerocaulis tuhajkulensis (Pryn. in Gorskii, 1944 ex Pryn., 1974) comb. nov.

Basionym: Osmundites tuhajkulensis Pryn. in Gorskii (in “Geology of the USSR” [in Russian] XII: 252. 1944; nomen nudum) ex Pryn. (in “Proceedings of the All-Union Scientific Research Geological Institute” [in Russian] 182: 254. 1974).

Synonym: Osmundites jelkinensis Pryn. in Gorskii (in “Geology of the USSR” [in Russian] XII: 252. 1944; nomen nudum).

Comments: Presumably due to its publication in Russian, the species has been largely overlooked in the international literature (but see Gould, 1970).

Claytosmunda beardmorensis (J.M.Schopf, 1978) comb. nov.

Basionym: Osmundacaulis beardmorensis J.M.Schopf (in Canadian Journal of Botany 56: 3034. 1978).

Synonyms: Millerocaulis beardmorensis (J.M.Schopf) Tidwell (in SIDA 11: 402. 1986)

Ashicaulis beardmorensis (J.M.Schopf) Tidwell (in SIDA 16: 256. 1994).

Comments: This species is known from large, silicified root mounds from the Middle Triassic Fremouw Formation of the Transantarctic Mountains (Schopf, 1978). Root growth removed much of the parenchyma of the rhizomes inside the mounds; therefore, knowledge about the anatomy, especially of the stipes, is still incomplete. However, those features available demonstrate very close similarity to extant Claytosmunda claytoniana. This is even more noteworthy because large, almost complete frond compression fossils identical to modern Claytosmunda claytoniana occur also in the Triassic of East Antarctica (Taylor et al., 1990; Phipps et al., 1998).

One remarkable difference is that leaf traces of Osmundacaulis beardmorensis commonly arise with protoxylem still in mesarch position (Schopf, 1978; B. Bomfleur, 2012, personal observation). However, this feature also occurs in other species of Osmundoideae and might reflect aberrant development of an ancestral state.

Claytosmunda chengii nom. nov.

Basionym: Ashicaulis claytoniites Y.-M.Cheng (in Review of Palaeobotany and Palynology 156: 98. 2011).

Synonyms: Claytosmunda claytoniites (Y.-M.Cheng, 2011) comb. nov. [to be replaced; a junior homonym of Claytosmunda claytoniites (C.J.Phipps, T.N.Taylor, E.L.Taylor et al., 1998) comb. nov. based on (basionym) Osmunda claytoniites C.J.Phipps, T.N.Taylor, E.L.Taylor et al. (in American Journal of Botany 85: 889. 1998)].

Ashicaulis advencensis Y.-M. Cheng (nomen nudum in Review of Palaeobotany and Palynology 156: 101. 2011).

Claytosmunda embreei (Stockey & S.Y.Sm., 2000) comb. nov.

Basionym: Millerocaulis embreei Stockey & S.Y.Sm. (in International Journal of Plant Sciences 161: 160. 2000).

Claytosmunda johnstonii (Tidwell, Munzing & M.R.Banks, 1991) comb. nov.

Basionym: Millerocaulis johnstonii Tidwell, Munzing & M.R.Banks (in Palaeontographica Abteilung B 223: 94. 1991).

Synonyms: Ashicaulis johnstonii (Tidwell, Munzing & M.R.Banks) Tidwell (in SIDA 16: 256. 1994).

Claytosmunda liaoningensis (Wu Zhang & Shao-Lin Zheng, 1991) comb. nov.

Basionym: Millerocaulis liaoningensis Wu Zhang & Shao-Lin Zheng (in Acta Palaeontologica Sinica 30: 717. 1991).

Synonym: Ashicaulis liaoningensis (Wu Zhang & Shao-Lin Zheng) Tidwell (in SIDA 16: 256. 1994).

Claytosmunda plumites (N.Tian & Y.D.Wang 2014) comb. nov.

Basionym: Ashicaulis plumites N.Tian & Y.D.Wang (in Journal of Plant Research 127: 210. 2014a).

Claytosmunda preosmunda (Y.M.Cheng, Yu F.Wang & C.S.Li, 2007) comb. nov.

Basionym: Millerocaulis preosmunda Y.M.Cheng, Yu F. Wang & C.S.Li (in International Journal of Plant Sciences 168: 1352. 2007).

Claytosmunda sinica (Y.M.Cheng & C.S.Li, 2007) comb. nov.

Basionym: Millerocaulis sinica Y.M.Cheng & C.S.Li (in Review of Palaeobotany and Palynology 144: 253. 2007).

Claytosmunda tekelili (E.I.Vera, 2012) comb. nov.

Basionym: Millerocaulis tekelili E.I.Vera (in Alcheringa 36: 37. 2012).

Claytosmunda wangii (N.Tian & Y.D.Wang, 2014) comb. nov.

Basionym: Ashicaulis wangii N.Tian & Y.D.Wang (in Science China Earth Sciences 57: 673. 2014b).

Claytosmunda wehrii (C.N.Mill., 1982) comb. nov.

Basionym: Osmunda wehrii C.N.Mill. (in American Journal of Botany 69: 116. 1982).

Osmunda kidstonii (Stopes, 1921) comb. nov.

Basionym: Osmundites kidstonii Stopes (in Annals of Botany 35: 59. 1921).

Synonyms: Osmundacaulis kidstoni (Stopes) C.N.Mill. ex C.N.Mill. (in Contributions from the Museum of Paleontology University of Michigan 23: 137. 1971).

Millerocaulis kidstonii (Stopes) Tidwell (in SIDA 11: 403. 1986).

Ashicaulis kidstonii (Stopes) Tidwell (in SIDA 16: 256. 1994).

Comments: The species is incompletely known; information on the structure of the pith, stele, and cortex is lacking. However, the anatomy of the stipe bases is so clear and so similar to modern Osmunda that we have no hesitation in assigning the fossil to this genus.

Osmundastrum indentatum (R.S.Hill, S.M.Forsyth & F.Green, 1989) comb. nov.

Basionym: Australosmunda indentata R.S.Hill, S.M.Forsyth & F.Green (in Palaeontology 32: 292. 1989).

Synonyms: Millerocaulis indentata (R.S.Hill, S.M.Forsyth & F.Green) Tidwell (in SIDA 16: 255. 1994).

Osmundastrum precinnamomeum (C.N.Mill., 1967) comb. nov.

Basionym: Osmunda precinnamomea C.N.Mill. (in Contributions from the Museum of Paleontology 21: 171. 1967).

Osmundastrum pulchellum (Bomfleur, G.W.Grimm & McLoughlin, 2015) comb. nov.

Basionym: Osmunda pulchella Bomfleur, G.W.Grimm & McLoughlin (in BMC Evolutionary Biology 15: 126. 2015).

Plenasium subgenus Aurealcaulis stat. et comb. nov.

Original name and status: Genus Aurealcaulis Tidwell & L.R.Parker (in American Journal of Botany 74: 805. 1987).

Plenasium (Aurealcaulis) bransonii (Tidwell & Medlyn, 1991) comb. nov.

Basionym: Aurealcaulis bransonii Tidwell & Medlyn (in Great Basin Naturalist 51: 331. 1991).

Plenasium (Aurealcaulis) burgii (Tidwell & J.E.Skog, 2002) comb. nov.

Basionym: Aurealcaulis burgii Tidwell & J.E.Skog (in Palaeontographica Abteilung B 262: 28. 2002).

Plenasium (Aurealcaulis) crossii (Tidwell & L.R.Parker, 1987) comb. nov.

Synonym: Plenasium (Aurealcaulis) crossii (Tidwell & L.R.Parker) comb. nov.

Basionym: Aurealcaulis crossii Tidwell & L.R.Parker (in American Journal of Botany 74: 805. 1987).

Plenasium (Aurealcaulis) dakotense (Tidwell & J.E.Skog, 2002) comb. nov.

Basionym: Aurealcaulis dakotensis Tidwell & J.E.Skog (in Palaeontographica Abteilung B 262: 32. 2002).

Plenasium (Aurealcaulis) moorei (Tidwell & Medlyn, 1991) comb. nov.

Basionym: Aurealcaulis moorei Tidwell & Medlyn (in Great Basin Naturalist 51: 326. 1991).

Plenasium (Aurealcaulis) nebraskense (Tidwell & J.E.Skog, 2002) comb. nov.

Basionym: Aurealcaulis nebraskensis Tidwell & J.E.Skog (in Palaeontographica Abteilung B 262: 31. 2002).

Plenasium (Plenasium) arnoldii (C.N.Mill., 1967) comb. nov.

Basionym: Osmunda arnoldii C.N.Mill. (in Contributions from the Museum of Palaeontology 21: 181. 1967).

Plenasium chandleri (C.A.Arnold, 1952) comb. nov.

Basionym: Osmundites chandleri Arnold (in Palaeontographica Abteilung B 92: 68. 1952).

Comments: In her detailed treatment of Osmunda dowkeri, Chandler (1965: 158) expressed the opinion that “[…] it seems reasonably certain that Osmundites chandleri should be transferred to the living Osmunda and to the sub-genus Plenasium within it.” We concur with this transferal.

Plenasium dowkeri (Carruth., 1870) comb. nov.

Basionym: Osmundites dowkeri Carruth. (in Quarterly Journal of the Geological Society 26: 352. 1870).

Synonym: Osmunda dowkeri (Carruth.) Chandler (in Bulletin of the British Museum (Natural History), Geology 10: 142. 1965).

Thamnopteris diploxylon (Kidst. & Gwynne-Vaughan, 1909) comb. nov.

Basionym: Zalesskya diploxylon Kidst. & Gwynne-Vaughan (in Transactions of the Royal Society of Edinburgh 46: 226. 1909).

Thamnopteris gracilis (Eichw., 1860) comb. nov.

Basionym: Chelepteris gracilis Eichw. (in Lethaea Rossica I: 98. 1860).

Synonym: Zalesskya gracilis (Eichw.) Kidst. & Gwynne-Vaughan (in Transactions of the Royal Society of Edinburgh 46: 220. 1909).

Thamnopteris javorskii (Zalessky, 1935) comb. nov.

Basionym: Iegosigopteris javorskii Zalessky (in Bulletin de l’Académie des Sciences de l’URSS: 747. 1935).

Thamnopteris splendida (Zalessky, 1931) comb. nov.

Basionym: Petcheropteris splendida Zalessky (in Bulletin de l’Académie des Sciences de l’URSS: 705. 1931b).

Thamnopteris uralica (Zalessky, 1924) comb. nov.

Basionym: Zalesskya uralica Zalessky (in Journal of the Linnean Society, Botany 46: 356. 1924).

Appendix B: Glossary of Terms and Abbreviations as they are used here

Axis: The most general term applied to the axial organs of Osmundales; includes the stem and its surrounding mantle of roots and persistent stipe bases; may be, for instance, arborescent (trunk) or non-arborescent (rhizome).

Complete gap: A leaf gap that completely perforates the stelar xylem siphon; note that gap formation may be complete even though the complete aspect of the perforation may not be visible in any single TS, e.g., in cases where gaps are small and transect the siphon obliquely at an acute angle.

Delayed gap: Mode of formation of a complete leaf gap in which the gap breaks through to the pith only after the trace has departed from the stele.

Dictyostele: Type of stele with external and internal phloem layers connecting through (at least some) leaf gaps, thus completely enveloping stelar xylem segments (or groups thereof) in TS of axis. Example: Osmundacaulis skidegatensis.

Dictyostelic: Having a dictyostele.

Dictyoxylic: Having a type of reticulate stele in which leaf gaps completely perforate the stelar metaxylem siphon such that in TS the siphon will appear like an interrupted ring of two or more separate stelar xylem segments.

Dissected stele: Type of stele with external and internal endodermis layers that connect through (at least some) leaf gaps, thus completely enveloping stelar xylem segments (or groups thereof) in some TS of axis. Example: Guairea milleri.

Heterogeneous sclerenchyma ring: A sclerenchyma ring in the stipe that, in its distal course away from the stem, develops distinct patches of particularly thick-walled fibres with characteristic arrangements beginning usually with an abaxial arch; proposed here as diagnostic of modern Osmundoideae.

Immediate gap: Mode of formation of a complete leaf gap in which the gap breaks through to the pith immediately upon the departure of the trace from the stele; usually indicated already further below in the stele in the form of a notch or incision along the internal surface of the siphon.

Incomplete gap: A leaf gap that does not perforate the stelar xylem siphon; formation of only incomplete gaps may be assumed in cases where the internal surface of the siphon is perfectly smooth and shows no notches, incisions, or embayments in a given TS.

“Hewitson method”: Method introduced by Hewitson (1962) for counting the number of stelar xylem segments in a given TS, in which only those xylem masses are recognized as individual segments that are completely separated from adjacent masses (i.e., not even connected by just a single tracheid); alternative approaches to determine the number of xylem segments might include, for instance, counting instead the number of protoxylem clusters in a given TS of the stelar xylem siphon.

Homogeneous sclerenchyma cylinder/ring: Stipe sclerenchyma cylinder that does not show a clear differentiation into regions composed of particularly thick-walled or completely occluded sclerenchyma cells (e.g., arches, masses, patches).

Leaf trace: Within stem portion of a vascular bundle supplying a leaf from the point where, in TS of axis, it appears completely separated from the stele up to the point where, in TS of axis, it departs from the stem to enter the stipe base and become the stipe vascular bundle.

Leaf gap: Notch, incision, or perforation left in the external surface of the stelar xylem cylinder above a departed leaf trace; may be, for example, complete versus incomplete or immediate versus delayed.

Mantle: Outer part of the axis enveloping the actual stem, composed usually of roots and persistent stipe bases; trunks of arborescent forms usually characterized by thick mantle with more or less vertical roots (i.e., roots mostly sectioned transversely in axis TS); rhizomes of non-arborescent Osmundales usually characterized by mantle with roots radiating out- and downwards in sinuous course (i.e., usually sectioned obliquely or longitudinally, but rarely transversely in axis TS), in some cases forming conspicuous mounds.

Petiole: Stipe.

Pith: Type of stem core consisting primarily of parenchyma, in some cases with varying amounts of sclerenchyma or interspersed tracheids.

Plenasoid: Term initially introduced to illustrate one representative example for a mode of leaf trace formation in which each trace is formed from two protoxylem poles originating from two adjacent stelar xylem segments; then adopted as one of a few “types” of leaf trace formation in Osmundales; here considered obsolete as a nominal category.

Rhizome: The creeping to (semi-)erect axis characterizing a non-arborescent osmundalean plant.

Root trace: Within stem portion of a vascular bundle supplying a root, from the point where, in TS of axis, it appears completely separated from the stele or leaf trace up to the point where, in TS of axis, it has departed from the stem and become the root vascular bundle.

RS: Radial section.

Sclerenchyma cylinder: a cylinder of sclerenchyma in the outer stipe cortex that derives from the outer stem cortex; may be homogeneous or heterogeneous; absent in Guaireaceae.

Sclerenchyma ring: aspect of a sclerenchyma cylinder in a given TS.

Semi-plenasoid (here considered obsolete): A mode of leaf trace formation in which each trace originates from a single protoxylem pole that begins to divide already in the stele, i.e., before the leaf trace separates from the stele.

Stele: Central part of the stem, excluding stem cortex; including all tissues contained inside the (external) stelar endodermis, such as stem core and stelar metaxylem siphon.

Stem: Central part of the axis consisting of stele and cortex, excluding the mantle of roots and stipe bases.

Stem core: The central part of the stele that is enveloped by the stelar metaxylem siphon; may consist primarily of tracheids or of a primarily parenchymatic pith with varying amounts of tracheids and/or sclerenchyma.

Stem cortex: The outer part of the stem surrounding the stele, extending from external endodermis out to stem epidermis; traversed by root traces and leaf traces.

Stipe (petiole): Leafless portion of a frond connecting the frond blade (lamina) to the stem; may bear stipular wings or spines.

Trunk: The tall, upright form of axis characterizing an arborescent plant.

TS: Transverse section.

Xylem segment, stelar x.s.: Anastomosing portion of the reticulate peripheral xylem siphon of a dictyoxylic stele that, in a given TS, appears as an isolated element separated from adjacent elements at each side by a complete (“Hewitson method”) leaf gap.

Xylem siphon, stelar x.s.: Distinct tube of xylem at the stele periphery; encloses the stem core; may show variable development of leaf gaps; shared by all Osmundales.

Appendix C: Identification and Classification Key

A polytomous key for identification and classification of fossil osmundalean axes is provided in Fig. 18. The concluding Fig. 19 gives an overview of the taxic diversity and structural disparity of osmundalean axes grouped following the revised systematic treatment proposed in this paper.

Figure 18 Polytomous key illustrating possible ways for straightforward identification and classification of structurally preserved axes of Osmundales according to diagnostic anatomical features.

Figure 19 Overview diagram illustrating the diversity and disparity of osmundalean stems arranged according to the taxonomic revision proposed here.

Each circular diagram represents a simplified stem cross-section of the relevant species that gives basic information on the structure and proportional sizes of the stem core, xylem siphon, cortex or cortical layers, and the entire stem excluding the mantle, all to the same scale. Taxon names are given in the form of six-letter labels that are contractions formed from the three first letters of the genus name in bold followed by the first three letters of the specific epithet; Oum, Osmundastrum. Primarily parenchymatic tissues in pale yellow; stelar metaxylem in dark brown; sclerenchyma in light brown; note that individual fibres or fibre patches in pith or cortical tissues are omitted. A large-format, high-resolution version of this image is provided in Fig. S6.

Supplemental Information

Supplemental Information 1 Nomenclatural and taxonomic history and taxic diversity of extinct and extant Osmundales, based on selected comprehensive reviews since the 1970s.

Click here for additional data file.

Supplemental Information 2 Diagrammatic representation of a cutout stem portion of a Millerocaulis-type osmundoidalean rhizome (Osmundoideae, Osmundaceae) showing a selection of general anatomical features.

Petiolar parenchyma in yellow-green; outer and inner surface of sclerenchymatic outer cortex in light grey; xylem in brown; parenchyma of pith and inner cortices left transparent and most roots omitted for clarity reasons (unlabelled version of Fig. 1).

Click here for additional data file.

Supplemental Information 3 Planar phylogenetic network (neighbour-net) for all operational units of order Osmundales except those with >60% characters missing.

Fully labelled version of Fig. 9.

Click here for additional data file.

Supplemental Information 4 Planar phylogenetic network (neighbour-net) for all operational units of family Osmundaceae.

Fully labelled raw version of Fig. 11.

Click here for additional data file.

Supplemental Information 5 Planar phylogenetic network (neighbour-net) for all operational units of subfamily Osmundoideae (Osmundaceae).

Fully labelled raw version of Fig. 13.

Click here for additional data file.

Supplemental Information 6 Overview diagram illustrating the diversity and disparity of osmundalean stems arranged according to the taxonomic revision proposed here.

Each circular diagram represents a simplified stem cross-section of the relevant species that gives basic information on the structure and proportional sizes of the stem core, xylem siphon, cortex or cortical layers and the entire stem excluding the mantle, all to the same scale. Taxon names are given in the form of six-letter labels that are contractions formed from the three first letters of the genus name in bold followed by the first three letters of the specific epithet; Oum, Osmundastrum. Primarily parenchymatic tissues in pale yellow; stelar metaxylem in dark brown; sclerenchyma in light brown; note that individual fibres or fibre patches in pith or cortical tissues are omitted.

Click here for additional data file.

Supplemental Information 7 Matrix in nexus format.

Click here for additional data file.

Supplemental Information 8 Matrix in simplified nexus format.

Click here for additional data file.

Supplemental Information 9 Spreadsheet compilation showing results of various statistical analyses of the matrix.

Click here for additional data file.

Supplemental Information 10 Simple pairwise (Hamming) distances inferred from the matrix.

Click here for additional data file.

Supplemental Information 11 Tutorial for modifying matrix and analyses, using the recently described Millerocaulis zamunerae Sagasti, Garcia Massini, Escapa et al. as an example.

Click here for additional data file.

Supplemental Information 12 Annotated spreadsheet file listing various morphological features and character codings for all operational units in the matrix.

Click here for additional data file.

Supplemental Information 13 Comparison of dimensions and delta values for a selection of comprehensive morphological character matrices.

Click here for additional data file.

Supplemental Information 14 Heatmap comparing pairwise (Hamming) distance values for all operational units used in the matrix.

Click here for additional data file.

We thank the editor William DiMichele (Washington, D.C., USA) and two anonymous reviewers for detailed and constructive criticism on an earlier version of this manuscript; Harald Schneider (London/Xishuangbanna) for helpful discussion on the recently revised classification of modern Osmundaceae (PPG I, 2016); Weston Testo (Burlington) for providing complete documentation of the phylogenetic analyses in Testo & Sundue (2016); and Ignacio H. Escapa (Trelew), Juan Garcia Massini (La Rioja), Ning Tian (Shenyang), Ezequiel I. Vera (Buenos Aires), and Yongdong Wang (Nanjing) for kind help in obtaining literature.

Additional Information and Declarations

Competing Interests

Author Contributions

Data Availability

New Species Registration

The authors declare that they have no competing interests.

Benjamin Bomfleur conceived and designed the experiments, performed the experiments, analyzed the data, contributed reagents/materials/analysis tools, wrote the paper, prepared figures and/or tables, reviewed drafts of the paper.

Guido W. Grimm conceived and designed the experiments, performed the experiments, analyzed the data, contributed reagents/materials/analysis tools, wrote the paper, prepared figures and/or tables, reviewed drafts of the paper.

Stephen McLoughlin analyzed the data, contributed reagents/materials/analysis tools, wrote the paper, reviewed drafts of the paper.

The Supplemental Data Archive (SDA) is available from the Dryad Digital Repository (http://dx.doi.org/10.5061/dryad.270gs) and is mirrored at www.palaeogrimm.org/data/Bfr17_SDA.zip.

The following information was supplied regarding the registration of a newly described species: New names and combinations of fossil plant taxa (not governed by IPNI; fossils receive no LSIDs)

Itopsidemoideae subfam. nov.

Osmundinae subtrib. nov.

Todeinae subtrib. nov.

Millerocaulis tuhajkulensis (Gorskii ex Pryn., 1944) comb. nov.

Claytosmunda beardmorensis (J.M.Schopf, 1978) comb. nov.

Claytosmunda embreei (Stockey & S.Y.Sm., 2000) comb. nov.

Claytosmunda chengii nom. nov.

Claytosmunda johnstonii (Tidwell, Munzing & M.R.Banks, 1991) comb. nov.

Claytosmunda kidstonii (Stopes, 1921) comb. nov.

Claytosmunda liaoningensis (Wu Zhang & Shao-Lin Zheng, 1991) comb. nov.

Claytosmunda plumites (N.Tian & Y.D.Wang, 2014) comb. nov.

Claytosmunda preosmunda (Y.M.Cheng, Yu F.Wang & C.S.Li, 2007) comb. nov.

Claytosmunda sinica (Y.M.Cheng & C.S.Li, 2007) comb. nov.

Claytosmunda tekelili (E.I.Vera, 2012) comb. nov.

Claytosmunda wangii (N.Tian & Y.D.Wang, 2014) comb. nov.

Claytosmunda wehrii (C.N.Mill., 1982) comb. nov.

Osmunda kidstonii (Stopes, 1921) comb. nov.

Osmundastrum indentatum (R.S.Hill, S.M.Forsyth & F.Green, 1989) comb. nov.

Osmundastrum precinnamomeum (C.N.Mill., 1967) comb. nov.

Osmundastrum pulchellum (Bomfleur, G.W.Grimm & McLoughlin, 2015) comb. nov.

Plenasium subgenus Aurealcaulis stat. et comb. nov.

Plenasium (Aurealcaulis) bransonii (Tidwell & Medlyn, 1991) comb. nov.

Plenasium (Aurealcaulis) burgii (Tidwell & J.E.Skog, 2002) comb. nov.

Plenasium (Aurealcaulis) chandleri (Arnold, 1952) Chandler ex Bomfleur, G.W.Grimm & McLoughlin

Plenasium (Aurealcaulis) crossii (Tidwell & L.R.Parker, 1987) comb. nov.

Plenasium (Aurealcaulis) dakotense (Tidwell & J.E.Skog, 2002) comb. nov.

Plenasium (Aurealcaulis) moorei (Tidwell & Medlyn, 1991) comb. nov.

Plenasium (Aurealcaulis) nebraskense (Tidwell & J.E.Skog, 2002) comb. nov.

Plenasium (Plenasium) arnoldii (C.N.Mill., 1967) comb. nov.

Plenasium chandleri (Arnold, 1952) comb. nov.

Plenasium dowkeri (Carruthers, 1870) comb. nov.

Thamnopteris diploxylon (Kidst. & Gwynne-Vaughan, 1909) comb. nov.

Thamnopteris gracilis (Eichw., 1860) comb. nov.

Thamnopteris javorskii (Zalessky, 1935) comb. nov.

Thamnopteris splendida (Zalessky, 1931) comb. nov.

Thamnopteris uralica (Zalessky, 1924) comb. nov.

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
