# Peer review of "The fossil Osmundales (Royal Ferns)—a phylogenetic network analysis, revised taxonomy, and evolutionary classification of anatomically preserved trunks and rhizomes"

_PeerJ, doi:10.7717/peerj.3433_

## Round 0.1 · original submission · Major Revisions

Overall, I believe this paper is worthy of publication in PeerJ, but only after what might amount to a major revision. I stress that the paper should not be rejected. This represents a great deal of work and is an ambitious and potentially high profile study. Refinement of the manuscript should greatly increase its reach.

The paper provides valuable information on the stem anatomy of the ferns in question, and undertakes a phylogenetic analysis that will be of interest to a large number of plant scientists.

However, both reviewers provide excellent suggestions for ways in which this paper can be improved and have significantly higher impact. The comments overlap in a number of areas but most particularly the need for better organization of the presentation of the data and results, and clearer presentation of the methods used. It would appear that both reviewers had some difficulty ferreting out data, and following the results and discussion due to the manner in which the manuscript is organized. Both also caution about making strong statements that are potentially not supported (or may appear unsupported, perhaps simply due to the organization of the paper) by the analyses or that may be unnecessarily inflammatory.

Reviewer 1 raises a number of important questions about the analysis, and the presentation of the data. In addition, the reviewer raises the matter of environmental effects on the anatomical characters that form the basis of the study. This should be addressed, if possible, in a revision.

Reviewer 2 again makes significant, and I believe helpful, comments about the presentation and organization of the data, and the way the manuscript is structured. This reviewer also used the authors' data matrix to run a parsimony analysis and, from the results of that, suggests that the authors do likewise, which would (or would not) support some of their assertions about that approach.

Reviewer 1 ·

Basic reporting

The authors have conducted a needed analysis for stem anatomy in these ferns and provided some standardization of the terms to be applied to allow for improved comparison. This was a large task and one that can be valuable for future studies.
Overall the study seems to have been exhaustive in finding all the species for which stem anatomy is known.
The main weakness I see in this paper is that there are really no methods explained in such a way that the work is repeatable.

Experimental design

Methods: This may be the journal style, but I think that putting the description of all these character states into the methods section dilutes the understanding of what methods were used for this study. To me, it would be best to describe the methods for obtaining the characters – i.e. how did the authors evaluate these characters used in the analysis? Was it just from the literature, or reexamination of actual specimens in some cases, how there were examined, etc. and then describe the methods of analysis to create the new classification and expose the relationships of the taxa. If only literature was used for this study, what was used to determine the characters (light microscope photos, SEM photos, drawings, the text descriptions by the original authors, etc.)? Were all the specimens at the same stage of development? Were these all from similar environments or were some from wet and others from dry? Would this influence the anatomical characters used – such as more or less schlerencyma vs parenchyma? More could be provided for justification on the analyses done. What was the rationale for selecting 60% undefined characters to eliminate taxa – why not 50% or 70%? What is magical about 60%? I did not see an explanation for the choice or a reference cited to indicate that this was a standard procedure. Were the characters weighted in any way? Did not say in text or I missed it. NeighborNet is a solid analysis as the authors note and generates networks, but it depends on the reliability of the data put into it and it is a planar result. Therefore it would be helpful to know how many reticulations were examined, or if any appeared that would warrant additional analyses before accepting them. And so forth – to adequately provide the information for someone else to replicate this study. That is what methods should do.
The actual description of the characters for the taxa should be in a description section outlining each of the characters and their various states, followed by the analysis of where they appear in the various taxa examined. I found myself jumping around in this paper to follow the reasoning from section to section and then to the figures. So I suggest a separate section for description to provide clarity and logical flow for a reader.
Discussion – this section could have the characters discussed if no separate description section is provided. This is probably the place for the rationale for use of NeighborNet as the method. The discussion does not really address certain aspects of this artificial classification. Use of only stem anatomy will bias the study in various ways and should be addressed. If the developmental stages were not all the same for the specimens examined, how might that influence the characters used? Is there an environmental effect on stem growth – these stems are pretty much below or close to ground level, so water amount would influence the types of cells produced. Would this change the results of the analysis in any way? Since the authors are using an alternate artificial system of relationships (stem anatomy of fossil and modern vs. DNA phylogeny of only modern) their criticisms of some of the previous studies are somewhat overstated (e.g. “argument to exclude Osmunda cinnamomea from the genus should be obsolete”) as some of these classifications might prove to be correct if all characters of the plant including frond characters, fertile parts, etc. were to be considered in some future analysis and another researcher decided to break up the complex of genera in a different analysis. Any hypothesis of relationships can vary depending on the taxa used and the characters being emphasized, and I’m not convinced that using only the stem anatomy as the determining criteria for relationships has the final authority based on their lack of discussion about factors that can influence the structure of the stem and its anatomy. Therefore I suggest presenting previous classifications and then presenting the data from their analysis which suggests these other classifications should be revised for this particular data in this paper. This is particularly important here, because as noted above, the methods are not well stated or clear. And their title states that they are only considering the anatomically preserved stems and no other information from either modern or fossil material.

Validity of the findings

Mainly noted in the section above - the validity cannot be fully judged when the methods are not clearly stated, how much material was examined, or whether this paper is completely based on a review of previously published material. The parameters for the data analysis need to also be presented. Once that is clear, then the validity can be determined. The findings are certainly a valid opinion for a classification, as all classifications are based on the characters used by the authors.

Additional comments

Additional line by line comments are below for corrections.
Line 41 Royal Ferns is the subject of the sentence, should be ‘Royal Ferns are’ or use the ordinal name as the subject for ‘Osmundales (Royal Ferns) is’ as stated in the abstract.
Line 47 – Couper is cited as 1953 here but is 1954 in the Literature Cited. Correct one of the references.
Line 130. Why not use 0 for absent in all cases? This character 1, character 28 and character 29 use 2 for absent, character 33 uses 0 for present and 1 for absent, while most of the other characters that have absence as a state use 0. It would be easier for readers to understand the analysis and the rest of the statements about characters if there was consistency in the use of absence as 0. Or if you prefer keep them all 2, but be consistent for clarity. Does using a different number bias the analysis in any way? By selecting a 2 rather than a 0, are you implying that you think one is more advanced than another? All that should be clearly stated if different numbers are being applied for an absent character depending on the character being examined.
Line 143. Delete ‘that’.
Line 210. Delete ‘it’ – unclear reference to antecedent. Is ‘it’ the position, the plant, or the stele? Remove ‘it’ and the sentence seems clearer that position of section is the reference.
Lines 734 and 748 – refers to a 2013 reference with Wang, but only 2014 a and b are listed in Lit cited. Where is this one? Or should it be one of the 2014s?
Line 926 Zalessky is this a or b for 1931?
Line 1153 – ref for Rothwell and Stockey 2008 not in lit cit
Line 1384 – no date after Chandler for citation
Line 1821 - Bernhardi needs date of pub. 1801?
Line 1822 – Blume no date for pub 1928?
Line 1866 – no date for pub. 1907?
Line 1908 – Gorskii is out of alphabetical order and should precede the Gould reference
Line 1990 - no Kidston and Gwynne-Vaughn 1914 is cited anywhere in the paper - delete
Line 2413 – “disparaty” is probably disparity, right?

Reviewer 2 ·

Basic reporting

Most of the results are presented in figure captions rather than the body of the text. Please add a section explaining the results of the analyses that stands on its own. Similarly, although there is ample discussion under each taxon in the systematics section, I would like to see a general discussion about how these results challenge or agree with current classifications

Experimental design

no comment

Validity of the findings

no comment.

Additional comments

Recent papers by Bomfleur and associates have produced a series of excellent papers integrating fossil and extant Osmundaceae. The work is of the highest standard and have contributed enormously to our knowledge of these plants. This paper, however, I think would benefit from some heavy revision and perhaps additional analyses. The authors might feel that my criticisms stem from a philosophical disagreement about the value of neighbour-networks, and that is a fair response. However, I would like to stress here that what I really take issue with is the process of using networks to make decisions, not the networks themselves. I provide more specific comments organized by line number.

Subgenus Claytosmunda was recently elevated to genus by Metzgar & Rouhan in PPG 1, a new fern classification. Schneider, H., Smith, A.R., Hovenkamp, P., Prado, J., Rouhan, G., Salino, A., Sundue, M., Almeida, T.E., Parris, B., Sessa, E.B. and Field, A.R., 2016. A community-derived classification for extant lycophytes and ferns. Journal of Systematics and Evolution. Please revise the text to include this.

39: Uses of primitive and derived states in the text are fine, but the general application of “primitive” to the clade as a whole in this sentence seems is at risk of conflating the extant plants with their extinct ancestors. I would recommend changing this to something like “contains a high proportion of primitive character states” or something along those lines.

568-609: I am not convinced by the author’s arguments in favor of an “evolutionary classification”. Their reasoning mostly stems from the inconvenience of a cladistics-based classification. I consider it a great strength that a newly scored trait can change a topology and transform synapomorphies in plesiomorphies. Many of the “weaknesses” presented can also be viewed as strengths. I suggest dialing back this critique and focusing on justification of the evolutionary classification. I don’t agree that its better, but I do feel that classifications involve many subjective decisions and the authors are free to make theirs.

611: Nomenclatural remarks. It seems pretty evident to me that these names will be effectively published. Is this statement necessary?

Results and Discussion: Is the editorial standard of PeerJ to include all of your results and discussion in the figure captions? It would be nice to have a paragraph explaining your major findings that stands on its own and not have to read about the major results in the figure caption.
The authors make a compelling argument as to why a bifurcating phylogenetic tree might not be the best way to analyze their data; it would suffer badly from conflicting topologies and the strict consensus would include large polytomies. However, I disagree that this is reason not to perform the analysis. I analyzed the included dataset under parsimony using TNT and, although the strict consensus includes large polytomies, it also includes some nicely resolved clades. I find these results useful because they provide a way to distill the dataset down to the most robust relationships. Single most parsimonious trees and Measures of clade support can also be evaluated to get more out of the data. These analyses also have the benefit of directly using the character data that was generated, and not relying in a distance matrix. I would suggest that the authors conduct some type of bifurcating tree analyses [they probably already have] to be included as a supplementary file. I understand that the authors might disagree here, the title of the paper mentions “network analysis” after all. This brings me to the larger question. Are neighbor-net analyses alone appropriate for drawing the conclusions that are made here? Frankly, I don’t think they are. I don’t have any problem with the analyses themselves, I find them to be a useful way of examining the dataset. However, I am concerned with the interpretation of the results and how they were used to make taxonomic decisions. Most of the statement to this effect are in the figure captions where the authors make statements emphasizing the shape of their networks, such as “note the clear divergence…”. But many of these “clear divergences” could be interpreted differently. The “clarity” of divergence also differs based on the taxa included. In figure 10, Itopsidemoideae is “clearly” distinguished from Guaireoideae, but in Fig. 9 that is not the case; they both belong to the same undifferentiated arm of the network. This example makes me wonder if any analysis including as few taxa as Fig. 10 would give us some sort of “differentiated” pattern. Unless some other criteria by which taxonomic decisions are based can be introduced, these choices seem particularly subjective to me. I don’t doubt that the authors are defining “good” taxa, but this will be lost on any reader who is not an expert in the group. I suspect that since the authors have a strong grasp of the distribution of character states, they are seeing patterns in the results are not apparent to the rest of us. This relates to another problem, networks don’t tell us anything about the distribution of character states, homoplasy, or synapomorphies. Can you state whether your ingroup is monophyletic based on these results? It appears that any knowledge of character data is drawn from the descriptions of taxa. Would it be more straightforward to simply circumscribe taxa using a traditional morphological taxonomic approach?

I appreciate that you have italicized diagnostic characters in the descriptions. It’s not clear to me however at what level they are diagnostic. Does that mean that they have no homoplasy? Or no homoplasy within their clade? Can you provide a bit more explanation?

Although there is ample discussion under each taxon in the systematics section, I would like to see a general discussion about how these results challenge or agree with current classifications.
1301: You state “As long as the inclusion of the Plenasium species group in Osmunda remains universally accepted, then any argument to exclude Osmunda cinnamomea from the genus should be obsolete”. I disagree because your logic is essentially phenetic and ignores the fact that Osmunda and Osmundastrum can exhibit plesiomorphic traits, and Plenasium (which is nested in them) has derived traits.

Here, you also mention the “potentially misinformative molecular data" the results in the paraphyly of Osmunda s.s. and cite Bomfleur, 2015. This is an important topic and I am glad you raised the issue here. In Bomfleur 2015 you cite conflicting signal from markers and insufficient outgroup selection as reasons why the molecular data might be giving an inaccurate result. The outgroup problem has been addressed in the recent phylogeny of Testo & Sundue 2016 [Testo, W. and Sundue, M., 2016. A 4000-species dataset provides new insight into the evolution of ferns. Molecular Phylogenetics and Evolution, 105, pp.200-211], but those authors find the same result as Metzgar. The marker selection is different from Metzgar, but overlapping for a couple of markers. Thus it seems to weigh in on the issue.

1145: You state that this is “a plain example of the failure of strict-cladistic systematics". This is a gratuitous comment that doesn't need to be here since you don't actually perform these analyses or use a strict cladistic classification. Also, isn't another solution simply to re-circumscribe Todea? Furthermore, you’re admission that its status is "ambiguous" seems to deflate it as a good example of a "failure".
Supplement: I am confused by the inclusion of molecular sequence data and tree in the data file since these are not discussed in the text at all.
2077: probably is misspelled

---

## Round 0.2 · Minor Revisions

Overall, I believe this paper is very close to ready for acceptance. The authors have carefully revised their paper and have satisfied the earlier concerns of the reviewers regarding the initial submission. Reviewer 2 has made a few suggestions for minor revisions. I suggest the authors review these, revise the manuscript if they feel it is warranted, or respond to the reviewer's comments, and then resubmit the final revision. I do not imagine any further review will be necessary - so this final round is really simply a matter of clarification of these last bits, in order to give the authors a chance to consider the reviewer's suggestions.

Reviewer 1 ·

Basic reporting

The paper seems fine now and the authors have addressed previous comments and suggestions. The new organization helps to improve the flow of the paper and allows improved comprehension of their points.

Experimental design

No commen

Validity of the findings

no comment

Reviewer 2 ·

Basic reporting

no comment

Experimental design

no comment

Validity of the findings

no comment

Additional comments

The authors have addressed all of my major concerns described in the first round of reviews. I congratulate them on producing another fine manuscript. I note below a few sentences that I found confusing and which might benefit from revision.

Line 546: I think the use of the word ‘root’ in this sentence makes it hard to follow. The second sentence does not logically follow the first sentence, or if it does, I cannot follow it. The third sentence appears to be incomplete or missing a couple of words. Do you mean “but which have substantial…?

Figure 11 caption: I cannot follow the first phrase in the 2nd sentence. I do not see two distinct branches Shuichenglella since it only has one taxon. I do see that Osumdacaulis and Thamnopteridoideae have two branches each, but are these “distinct”? Am I missing something?

Figure 15 caption: I am confused by the phrase “overlap between Aurealcaulis and Plenasium with the species P. dowkeri and P. chandleri.” Do you mean that those species are less derived members of Aurealcaulis? They don’t exactly “overlap”.

---

## Round 0.3 · accepted · Accept

Thanks very much for the careful revisions of this paper. The paper is a very important contribution to our understanding of fern evolution, linking the fossil record to the record of extant forms. It is beautifully illustrated and should generate considerable interest from pteridologists and the broader botanical community. It is great to have this in PeerJ. Thanks for submitting it here.